# Verbalized Bayesian Persuasion

## Abstract

The study of information design explores how an information designer can influence the optimal behavior of players to achieve a specific objective through the strategic selection of the information provided. This paper focuses on a case, Bayesian Persuasion (BP), where the information designer holds an informational advantage over only one player. While information design originates from everyday human communication, traditional game-theoretic or multi-agent reinforcement learning methods often model information structures as discrete or continuous scalars or vectors, this approach fails to capture the nuances of natural language, significantly limiting their applicability in real-world scenarios. By leveraging the powerful language understanding and generation capabilities of large language models (LLMs), this paper proposes a verbalized BP framework that extends classic BP to real-world games involving human dialogues for the first time. Specifically, we map the classic BP to a verbalized mediator-augmented game, where LLMs instantiate the information designer and receiver. To efficiently solve the game in the language space, we transform agents' policy optimization into prompt optimization and propose a generalized equilibrium-finding algorithm with a convergence guarantee. Numerical experiments in realistic dialogue scenarios, such as recommendation letters, courtroom interactions, and law enforcement, validate that the VBP framework can reproduce theoretical results in classic settings and discover effective persuasion strategies in more complex natural language and multistage settings.

> *You can fool some of the people all of the time, and all of the people some of the time, but you can not fool all of the people all of the time.*

Abraham Lincoln

## 1 Introduction

In mixed-motive multi-agent reinforcement learning (MARL), agents aim to advance their interests by shaping others (Leibo et al., 2017; McKee et al., 2020; Dafoe et al., 2020; Leibo et al., 2021). Existing MARL methods typically achieve this through either mechanism (modifying rewards) (Yang et al., 2020; Zheng et al., 2022; Hua et al., 2023; Wang et al., 2024) or information design (modifying observations) (Wu et al., 2022; Bernasconi et al., 2022; Lin et al., 2023). This paper focuses on the latter. Specifically, agents' rewards depend not only on their actions but also on their observations. An information designer can commit to a strategy for providing state information to the agents, effectively altering the observation function. Information design studies how this designer can influence agents' behavior by strategically providing information and guiding them toward outcomes aligned with her objectives (Bergemann & Morris, 2019). Notably, mechanism design influences actions inter-episode through reward shaping, while information design is more challenging as it impacts actions intra-episode by directly altering agents' observations.

This paper examines the Bayesian persuasion (BP) problem (Kamenica & Gentzkow, 2011; Kamenica, 2019) between two agents: a sender and a single receiver. The sender (information designer) has an informational advantage over the receiver (player). Unlike communication learning (Foerster et al., 2016; Sheng et al., 2022; Zhu et al., 2022) in MARL, which often involves cheap talk (Lo et al., 2023), BP requires the sender to commit to an information disclosure mechanism publicly. The focus, therefore, is on rational (Bayesian) decision-makers who understand and optimally react to the

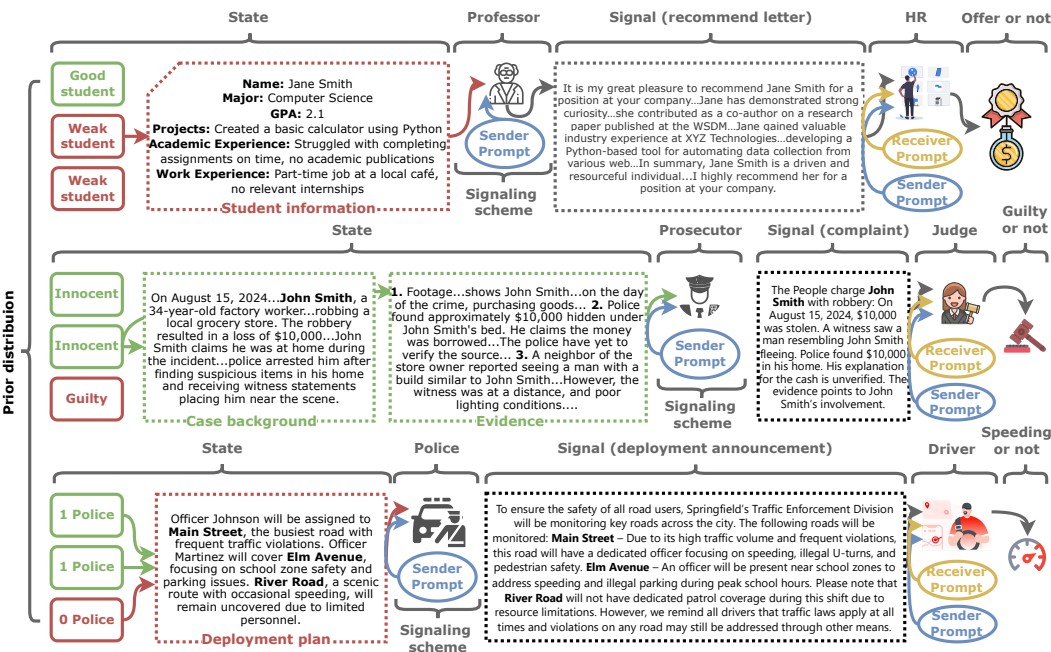

Figure 1: Mapping classic BP problems into verbalized mediator-augmented, extensive-form games.

disclosed information. Persuasion plays a significant role in modern economies, with one estimate suggesting that up to one-quarter (McCloskey & Klamer, 1995), even 30% (Antioch, 2013) of GDP, is persuasion. The study of BP has deep roots in economics, with numerous applications across fields such as school grading (Boleslavsky & Cotton, 2015), law enforcement deployment (Lazear, 2006), research procurement (Yoder, 2022), matching platforms (Romanyuk & Smolin, 2019), and routing systems (Das et al., 2017). Various theories have been proposed to explore the power of persuasion in different contexts (Kamenica, 2019).

Given a specific utility function, the fundamental BP problem is equivalent to finding an optimal Bayes-correlated equilibrium in an extensive-form game (Bergemann & Morris, 2013; 2019). Since the information space is typically small (often binary) and the action space is low-dimensional and discrete, even more complex BP problems—such as informed BP or multistage BP—can often be solved analytically using optimization techniques (Kolotilin, 2018; Dworczak & Martini, 2019; Makris & Renou, 2023; Koessler & Skreta, 2023). Some research has also explored the use of MARL to approximate solutions for more complex BP problems (Wu et al., 2022; Lin et al., 2023; Bacchiocchi et al., 2024). However, applying these methods to real-world settings requires constructing a model of the game in question, which involves defining the appropriate state space, action space, and transition dynamics.

Despite these successes, most applications remain limited to games in the colloquial sense, where real-world complexity is often oversimplified. For instance, in the recommendation letter problem, a professor must write a letter conveying nuanced information about a student's background. However, in the classic BP (Dughmi, 2017), the student's quality is reduced to a binary classification (weak or strong), and the professor's decision is restricted to either recommending or not. This abstraction strips away much of the meaningful information inherent in the actual task.

We aim to leverage game-theoretic methods, enhanced by LLMs (Zhao et al., 2023), to directly solve the original BP problem in the natural language domain. LLMs have advanced to a point where their generative capabilities enable realistic, human-like simulations of verbal interactions. Specifically, we model the BP problem as a verbalized mediator-augmented, extensive-form game (Zhang & Sandholm, 2022), where states, actions (or signals for the sender), and rewards are all represented as text, as shown in Figure 1.

For example, in the recommendation letter problem, the sender (professor) has a state that reflects the student's background, and the signal is the content of the recommendation letter. The receiver

(HR) observes this letter and must decide whether to accept the student. The rewards for both agents are represented numerically ($+1$, $-1$, or $0$), but the game's core elements—such as signals and actions—are expressed in natural language. To enable the sender and receiver to process, understand, and generate this text, we parameterize both using LLMs.

Before introducing our verbalized game solver, we must address two key challenges. The first is the design of the signal space, which can lead to a curse of dimensionality. In many real-world BP problems, the sender's information is conveyed through extended, complex natural language, such as a recommendation letter. The second challenge is optimizing the strategies of both agents. Directly updating LLMs in their parameter space is inefficient, and equilibrium points may not exist in Euclidean space ($\mathbb{R}^d$) (Gemp et al., 2024).

To address these challenges, we draw on the prompt-space response oracle (PSRO) work of Gemp et al. (2024), which model strategy optimization for both the sender and receiver as prompt optimization for their respective LLMs. This approach not only mitigates the challenge of optimization inefficiency but also reduces the action space from lengthy, complex text to compact, low-dimensional, discrete prompts. For instance, by adjusting the prompt given to the sender's LLM, we can control the "level of detail in the student's background description" in the recommendation letter.

Building on the PSRO, we propose several enhancements to improve its performance, efficiency, and stability in solving BP problems. These include verbalizing commitment assumptions, obedience constraints, and information obfuscation. More importantly, we extend PSRO to multistage games by proposing conditional prompt optimization and providing a convergence guarantee to the equilibrium solution. Together, these components form a comprehensive verbalized game solver tailored for BP problems, which we refer to as Verbalized Bayesian Persuasion (VBP). To our knowledge, VBP is the first general framework that attempts to solve real, non-abstract BP problems.

Our main contributions include: (1) Transforming real-world BP problems into verbalized mediator-augmented, extensive-form games, providing a unified interface for game-theoretic solvers; (2) Proposing a general game-theoretic solver for verbalized BP problems based on the PSRO framework, with a convergence guarantee to equilibrium solutions. We also introduce techniques such as verbalized commitment assumptions and obedience constraints, information obfuscation, and conditional prompt optimization to enhance the solver's performance, efficiency, stability, and (3) Reproducing results on classic BP problems consistent with traditional optimization methods and MARL while efficiently solving more complex multistage BP problems.

**Remark**  The combination of a game-theoretic solver with prompt optimization is not the only paradigm for utilizing LLMs to solve games. Widely adopted parameter-efficient fine-tuning (Xu et al., 2023; Han et al., 2024), as well as the recent trend of improving reasoning and problem-solving capabilities for complex and mathematical problems by having LLMs generate longer chains of thought prior to making decisions (Zelikman et al., 2022; 2024; OpenAI, 2024), are also very promising directions. The former allows for more fine-grained control of LLM outputs through in-weight updates, compared to in-context updates like prompt optimization, while the latter may enable LLMs to discover novel game solvers. VBP is orthogonal to these approaches. Its primary goal is to leverage the rich foundation of game theory by incorporating various game-theoretic solvers that have already been proposed, and to extend the solid theoretical results established in classical games for solving verbalized games.

## 2 PRELIMINARIES

This section presents an overview of the prompt-space response oracle, which is derived from the PSRO solver and used to address verbalized games. Game theory offers a mathematical framework to study interactions between multiple decision-makers (Bighashdel et al., 2024). However, classical game-theoretic analysis struggles with scalability due to the sheer number of strategies. To address this, a wide range of learning methods have been applied to large-scale games, with multi-agent reinforcement learning (Multi-Agent RL) (Yang & Wang, 2020; Zhang et al., 2021) being one of the most prominent approaches. Unlike traditional methods, learning-based approaches do not require full representation of the game and instead create agents that explore and adapt by interacting with the environment. Despite their contributions to developing agents, learning methods face inherent

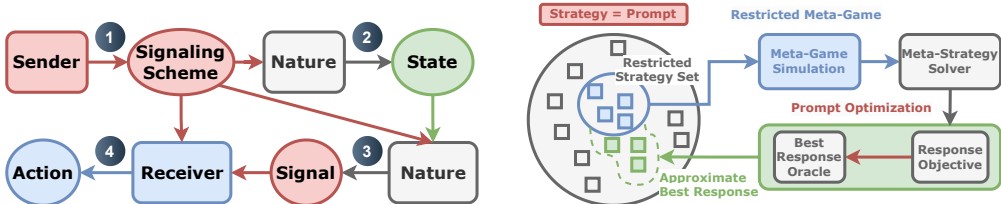

Figure 2: **Left:** Bayesian persuasion timing in the EFG; **Right:** Illustration of the PSRO framework.

challenges in games, such as non-stationarity (Tuyls & Weiss, 2012) and non-transitivity (Czarnecki et al., 2020; Sanjaya et al., 2022).

The policy space response oracles (PSRO) framework (Lanctot et al., 2017) emerged as a hybrid approach, combining traditional equilibrium computation with learning techniques. PSRO improves scalability by focusing on relevant subsets of strategies (Wellman, 2006; Bighashdel et al., 2024). Assos et al. (2023) demonstrate that PSRO-like approaches lead to tractable notions of approximate local Nash equilibria. As illustrated in Figure 2, PSRO algorithms begin with an initial set of strategies for each agent and proceed through two alternating steps. First, a normal-form meta-game (e.g., matrix game) is constructed, where each agent selects a meta-strategy to represent their overall behavior in the game. A meta-solver (e.g., Nash solver) then computes a solution (e.g., Nash equilibrium) for this meta-game. In the second step, each agent computes an approximate best response to the meta-strategy, aiming to improve their reward assuming the other agents play according to the meta-strategy. This process repeats until no agent can benefit by deviating from their strategy (Bighashdel et al., 2024).

The prompt-space response oracle (Gemp et al., 2024), shown in Figure 2, is a verbalized adaptation of the standard PSRO framework. Here, strategies are parameterized by LLMs and represented as prompts. The approximate best response is generated by optimizing and sampling prompt strings, as opposed to the standard PSRO protocol where best responses are typically computed using MARL or gradient-based optimization. Unless otherwise noted, PSRO in the following text refers to this prompt-space response oracle framework.

## 3 PROBLEM FORMULATION

### 3.1 BAYESIAN PERSUASION

The basic BP model is structured as follows (Kamenica, 2019). A receiver, an agent, has a utility function $u_1(a, \omega)$, which depends on her action $a \in \mathcal{A}$ and the state of the world $\omega \in \Omega$. Another agent, the sender (also known as the information designer), has a utility function $u_0(a, \omega)$. Both the sender and receiver share a common prior $\mu_0$ over $\Omega$. The sender's key decision is the choice of a signaling scheme. Let $\mathcal{S}$ represent a sufficiently large set of signals. It is enough to assume $|\mathcal{S}| \geq \min\{|\mathcal{A}|, |\Omega|\}$, meaning the number of signals is at least as large as both the state space and the action space. A signaling scheme is a mapping from the state to a distribution over signals, $\pi : \Omega \rightarrow \Delta(\mathcal{S})$. Let $\Pi$ denote the set of all possible signaling schemes. When viewed as an extensive-form game (EFG), the sequence of events is illustrated in Figure 2.

The receiver's behavior is straightforward. Given knowledge of $\pi$ (i.e., under the commitment assumption (Kamenica & Gentzkow, 2011)), the receiver updates her belief from the prior $\mu_0$ to the posterior $\mu_\pi(\omega \mid s)$ using Bayes' rule. She then selects an action $a^*$ that maximizes $\mathbb{E}_{\omega \sim \mu_\pi(\cdot|s)} u_1(a, \omega)$. Given this response mechanism from the receiver, the sender's objective is to solve the following maximization problem: $\max_{\pi \in \Pi} \mathbb{E}_{\omega \sim \mu_0} \mathbb{E}_{s \sim \pi(\omega)} u_0(a^*, \omega)$. An optimal signaling scheme exists that requires no more signals than there are actions available to the receiver. Thus, the sender can directly recommend an action to the receiver instead of sending a message. From the receiver's perspective, as long as she believes that the recommended actions are optimal according to her posterior belief, she will follow the sender's advice. These constraints on the sender's signaling scheme are referred to as obedience constraints (Myerson, 1979; Kamenica & Gentzkow, 2011). In this way, BP can be reduced to a simplified linear programming problem (Lin et al., 2023)

$$\max_\pi \mathbb{E}_\pi \left[ u_0(a, w) \right], \text{ s.t. } \sum_w P(w) \cdot \pi(a \mid w) \cdot \left[ u_1(a, w) - u_1\left(a', w\right) \right] \geq 0, \forall a, a'. \qquad (1)$$

## 3.2 Modeling BP as a Mediator-Augmented Game

To establish convergence for the VBP framework, we transform the classic BP problem into a special class of extensive-form games (EFGs), known as mediator-augmented games (MAGs, Zhang & Sandholm (2022)). Below, we provide the definition of the BP problem in the form of an MAG. At a high level, a mediator-augmented game introduces an additional player, the mediator, who exchanges messages with the players and provides action recommendations.

**Definition 1.** *A Bayesian persuasion problem, represented as a mediator-augmented game $\Gamma$, consists of the following components (Zhang & Sandholm, 2022): **(1)** a player, referred to as the receiver, denoted by the integer $1$; **(2)** a directed tree $H$ of histories or nodes, with the root denoted by $\varnothing$. The edges of $H$ are labeled with actions, and the set of legal actions at each node $h$ is denoted by $A_h$. Terminal nodes of $H$ are called leaves, and the set of such leaves is denoted by $Z$; **(3)** a partition of non-terminal nodes $H \backslash Z$ into $H_{\mathbf{C}} \sqcup H_0 \sqcup H_1$, where $H_1$ represents the nodes where player $1$ acts, and $\mathbf{C}$ and $0$ represent chance and the mediator (i.e., the sender), respectively; **(4)** for each agent $i \in \{1, 0\}$, a partition $\mathcal{I}_i$ of the decision nodes $H_i$ into information sets. Every node in a given information set $I$ must have the same set of legal actions, denoted by $A_I$; **(5)** for each agent $i \in \{1, 0\}$, a utility[1] function $u_i : Z \to \mathbb{R}$; and **(6)** for each chance node $h \in H_{\mathbf{C}}$, a fixed probability distribution $c(\cdot \mid h)$ over $A_h$.*

At any node $h \in H$, the sequence $\sigma_i(h)$ for agent $i$ consists of all information sets (infosets) encountered by $i$, along with the actions taken at those infosets on the path from $\varnothing$ to $h$, excluding $h$ itself. An agent has perfect recall if $\sigma_i(h) = \sigma_i(h')$ for all $h, h'$ within the same infoset. A pure strategy for agent $i$ specifies one action from $A_I$ for each information set $I \in \mathcal{I}_i$. A mixed strategy is a probability distribution over pure strategies, and the sequence form of a mixed strategy corresponds to the convex combination of pure strategies. Let $X_1$ and $X_0$ denote the polytope of sequence-form mixed strategies $\boldsymbol{x}_1$ for the receiver and $\pi$ for the mediator, respectively.

For a fixed $\pi \in X_0$, we say that $(\pi, \boldsymbol{x}_1)$ is an equilibrium of $\Gamma$ if $\boldsymbol{x}_1$ is a best response to $\pi$, meaning $\max_{\boldsymbol{x}_1' \in X_1} u_1(\pi, \boldsymbol{x}_1') \leq u_1(\pi, \boldsymbol{x}_1)$. We do not require the mediator's strategy (signaling scheme) $\pi$ to be a best response; hence, the mediator can commit to its strategy. The objective of this paper is to find an optimal (Stackelberg) equilibrium, which is an equilibrium $(\pi, \boldsymbol{x}_1)$ that maximizes the mediator's utility $u_0(\pi, \boldsymbol{x}_1)$.

## 4 Methods

This section will provide a detailed introduction to the VBP framework, as shown in Figure 3. First, we verbalize the MAG, and then, through signal polarization, we derive setting S1, aligning it as closely as possible with the classic BP problem, thereby facilitating subsequent validation of the VBP framework's effectiveness. Next, by removing signal polarization and introducing multi-stage interactions, we obtain settings S2 and S3, which are more closely aligned with real-world BP problems. Finally, by introducing the PSRO framework, along with OPRO (Yang et al., 2024) and FunSearch (Romera-Paredes et al., 2024) as two best response approximators, we solve the verbalized mediator-augmented game under different settings and provide a theoretical proof of convergence to equilibrium. The verbalized MAG, along with the three problem settings and the PSRO-based game-theoretic solver, collectively constitute the VBP framework.

### 4.1 Problem Formulation

In order to leverage the wealth of research in LLMs for BP in realistic scenarios, we must abstract and map components of BP to the symbolic language. Note the mapping can be chosen is not unique and many are possible. We now provide our mapping of realistic BP as an verbalized MAG.

---

[1]In this paper, we no longer distinguish between utility and reward.

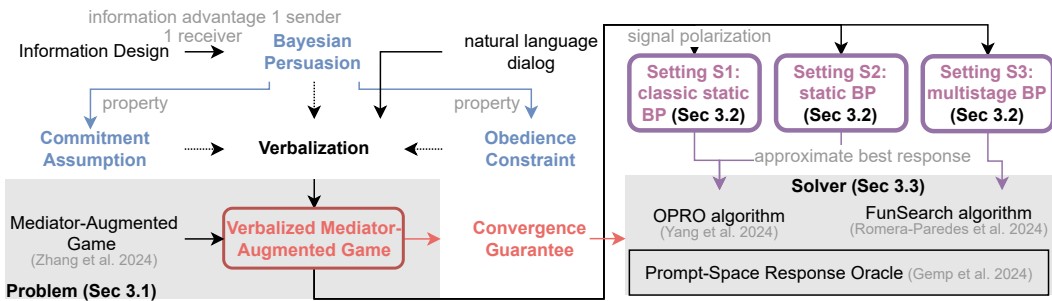

Figure 3: Verbalize Bayesian persuasion framework.

- **State** $\omega$. Unlike the classic BP, which only describes the state with binary values, the VBP framework includes richer textual information. For example, the student's quality and corresponding information in the REL problem, the case background and related evidence in the COR problem, and the police force allocation in the LAE problem. We generate specific scenarios by prompting the LLM. For the first two scenarios, we use conditional generation: first generating the student's quality or whether the defendant is guilty, then generating the detailed information or evidence.

- **Infosets** $\mathcal{I}$. We define the infosets for both the sender and the receiver as the interaction history between the two parties, which is only used in Setting S3. Specifically, this includes the signals sent by the sender, the receiver's decisions (public information), and their respective rewards (private information) from each round of interaction. Of course, the infosets of the sender and receiver also include the environmental state, specifically $\omega$ and the signal, respectively.

- **Action** $A$. Since we transform the strategy optimization problem into a prompt optimization problem through the PSRO framework, agents' actions (signaling scheme for the sender and action for the receiver) involve selecting prompts. Note that the prompts that can be optimized by both parties are not the entire text input to the LLM, but rather the "decision style," which consists of a category and corresponding content, with a total length of 2 to 3 words.

- **Terminal states** $\mathcal{Z}$ are determined by either a limit or the allowable tree depth. In Settings S1 and S2, each agent can only make 1 decision, while in Setting S3, they can make up to 5 decisions.

In addition to the basic components of the game mentioned above, the BP problem also includes two fundamental assumptions or constraints that need to be mapped into the verbalized MAG.

**Verbalized Committment Assumption** The key difference between BP and cheap talk lies in the presence of the commitment assumption, meaning that the receiver knows the mechanism or signaling scheme by which the sender generates messages. The VBP framework achieves the commitment assumption through prompt design or expand the receiver's infoset. Specifically, in VBP, the signaling scheme is equivalent to the key components in the prompt provided to the sender that influence the generated signals, and these components are the target of PSRO optimization. For example, in the REL problem, the signaling scheme includes components such as the "level of detail in the project description". VBP incorporates these key components into the receiver's prompt, see Appendix F.4 for more details. Since the sender generates the recommendation letter following these key components, the commitment assumption can be approximately achieved.

**Verbalized Obedience Constraint** The optimization in BP involves an (extended) obedience constraint (Lin et al., 2023), as shown in Equation 1. The most straightforward approach is to handle this constraint by transforming it into a penalty term, similar to reward shaping (Ng et al., 1999; Gupta et al., 2022). However, computing this penalty term requires integrating over the entire state and action space. To address this, we estimate the summation term using a sampling approach. Specifically, we calculate an estimate using the current state and an arbitrarily selected action. There are various ways to select actions, and here we introduce a theory-of-mind approach (Rabinowitz et al., 2018; Albrecht & Stone, 2018), where actions are selected based on predictions of what the receiver would do, using a prompt to pretrained and aligned LLMs to anticipate the receiver's likely actions.

For ease of understanding and evaluation, we break down the verbalized MAG into three settings, each solving BP of increasing difficulty. The following sections introduce these settings one by one.

## 4.2 THREE SETTINGS

**Setting S1 and S2: VBP in Static BP**   To validate the effectiveness of the VBP framework, we first test the solver (detailed below) on classic BP problems. Taking the REL problem as an example, the main conflict between the classic BP problem and the verbalized MAG lies in the signal space. In the former, the signal space is discrete, with the professor having only two discrete signals: to recommend or not recommend. However, in the latter, the signals consist of long natural language texts. To constrain the signal space of the sender within the verbalized MAG (**S1**), we introduce the *signal polarization* mechanism. Specifically, we use the pretrained and aligned LLM to score the signals output by the sender, such as determining the degree to which the recommendation letter supports the student (a real value between $0$ and $1$; similar prompts can be designed for other problems). Then, utilizing reward shaping techniques, we design an external reward based on the minimal distance between this score and the two extremes of recommendation ($1$) and non-recommendation ($0$). This approach encourages the sender to produce more straightforward signals, thereby aligning the signal space with the classic setting. Next, we consider a more generalized scenario where the signal space is not low-dimensional and discrete, but rather consists of complex natural text. To handle this scenario (**S2**), we simply need to remove the constraints on the signal space.

**Setting S3: VBP in Multistage BP**   This section considers a multistage scenario, which is extremely challenging for traditional methods. The agents engage in multiple rounds of interaction, and the sender's historical signals serve as the basis for the receiver's subsequent decisions. This undoubtedly increases the complexity, as the sender cannot arbitrarily exploit their information advantage. Instead, they must consider how their current actions may impact future rewards.

## 4.3 VERBALIZED GAME SOLVER

After modeling the BP as a verbalized MAG, we parameterize both agents using pretrained and aligned LLMs and optimize their strategies with the PSRO framework, thereby forming a general BP solver. We first present the following proposition based on the theoretical foundation of Zhang & Sandholm (2022), with the proof provided in Appendix C.

**Proposition 1.** *Verbalized Bayesian persuasion returns an $\varepsilon$-approximate Bayes correlated equilibrium in static BP and an $\varepsilon$-approximate Bayes-Nash equilibrium in multistage BP.*

In simple terms, the reason we can leverage the theoretical results of MAG is because different assumptions on the power of the mediator and the players' strategy sets induce different equilibrium concepts. The concept of Bayes correlated equilibrium (Bergemann & Morris, 2016) in static BP and Bayes-Nash equilibrium (Makris & Renou, 2023) in multistage BP is equivalent to the situation in the MAG where the mediator has an informational advantage, cannot lie (commitment assumption), and gains perfect recall under the extensive-form correlated equilibrium.

VBP does not directly solve the verbalized MAG using the PSRO framework. Instead, we make targeted improvements to PSRO for different settings, as illustrated in the Appendix F.1. For the S1 and S2 settings, we optimize the strategies of the sender and receiver using Algorithm 4 from the PSRO framework (Gemp et al., 2024), specifically the "categorical" approximate best response. Unlike in the original PSRO paper, we use the OPRO method (Yang et al., 2024) to generate both the categories and the specific content within the categories simultaneously.

The S3 setting presents a challenge for the PSRO. Existing PSRO is unconditional or episode-wise, meaning that the prompt generated at the beginning of each episode is used for every subsequent timestep. In mutlistage BP, this significantly restricts the size of the optimizable strategy space. In other words, both the sender and receiver can dynamically adjust their strategies based on the interaction history to achieve higher rewards. For example, the sender might honestly provide true information to the receiver early to build trust, then deceive the receiver later. Similarly, the receiver could bargain to extract more information. Thus, we propose a conditional version of PSRO, or stepwise PSRO, building on the original framework. Specifically, we introduce the FunSearch (Romera-Paredes et al., 2024), where the strategy to be optimized is no longer the prompt itself, but a function that generates the prompt. This function takes the current interaction history as input, thereby enabling conditional prompts. The pseudocode is shown in Algorithm 1 in Appendix B.

Moreover, since we use aligned LLMs, the sender struggles to output strategic signals, such as hiding or obfuscating relevant information about the true state, which leads to lower training efficiency. To speed up training, we introduce a information obfuscation mechanism. Similar to reward shaping (though experiments showed suboptimal results, likely due to the complexity of optimizing the reward function with too many components), we use an pretrained and aligned LLM to evaluate the degree of information hiding or obfuscation in the output signal. This feedback is then employed to perform multiple rounds of self-reflection (Shinn et al., 2024) before entering the PSRO loop.

## 5 EXPERIMENTS

We use 3 classic static BP problems in our experiments (detailed in Appendix D) In the Recommendation Letter (REL) problem (Dughmi, 2017), a professor writes recommendation letters for students, which HR uses to decide on hiring. The prior belief is that a candidate is strong with probability $1/3$ and weak with probability $2/3$. HR earns 1 for hiring a strong candidate, $-1$ for hiring a weak one, and 0 for not hiring, while the professor gains 1 for any hired student. In the Courtroom (COR) problem (Kamenica & Gentzkow, 2011), a prosecutor tries to convince a judge to convict a defendant, with the prior belief that the defendant is guilty with probability $0.3$. The judge earns 1 for a correct decision and 0 for an incorrect one, while the prosecutor earns 1 if the defendant is convicted. To simplify the courtroom investigations for LLM processing, we replace complex investigation procedures with selective evidence presentation, similar to REL. In the Law Enforcement (LAE) problem (Kamenica, 2019), drivers decide whether to speed or obey the law on a road with $Z$ miles, where $G$ police officers patrol. Speeding yields utility $V$, but drivers face a fine $K > V$ if caught. The prior belief $\mu_0 = G/Z$ represents the probability of police presence, and the police aim to minimize speeding by using a signaling scheme to influence drivers' behavior.

### 5.1 VBP IN STATIC GAMES (S1 & S2)

We first evaluate the effectiveness of the VBP method under the S1 setting. Two baseline methods are chosen: BCE and MARL. The former is based on the optimal equilibria computed in Lin et al. (2023), Kamenica & Gentzkow (2011) and Kamenica (2019), while the latter is based on the multi-agent reinforcement learning method proposed in Lin et al. (2023) for solving BP problems. As shown in Figure 4, the VBP framework successfully captures the essence of solving BP problems, namely, selectively withholding, obfuscating, or even deceiving about the true state, while also learning when to fully disclose accurate information.

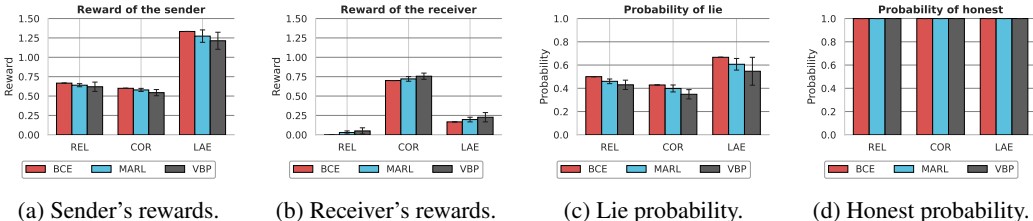

(a) Sender's rewards.  (b) Receiver's rewards.  (c) Lie probability.  (d) Honest probability.

Figure 4: Performance comparision on classic static BP problems. Averaged over 20 seeds. In the 3 BP problems, the probability of lying refers to describing a weak student as strong, an innocent defendant as guilty, or an unpatrolled segment as patrolled. Conversely, the probability of honesty refers to accurately describing a strong student, a guilty defendant, or a patrolled segment.

Next, we removed signal polarization to make the sender's signals in each problem more reflective of real-world recommendation letters, complaints, and police deployment announcements, resulting in the S2 setting. Since existing MARL methods cannot solve this, we only compared it with the VBP method from the S1 setting. The results are shown in Figure 5.

As can be seen, VBP's performance in the S2 is roughly on par with S1, with a slight performance drop. Additionally, we observed an interesting phenomenon: in both the S1 and S2, VBP achieves optimal strategy performance in terms of the probability of honesty. We speculate that this could be due to the alignment of the LLM used, which allows it to more easily converge to honest strategies, such as truthfully reporting the situation of a strong student, a guilty defendant, or a patrolled

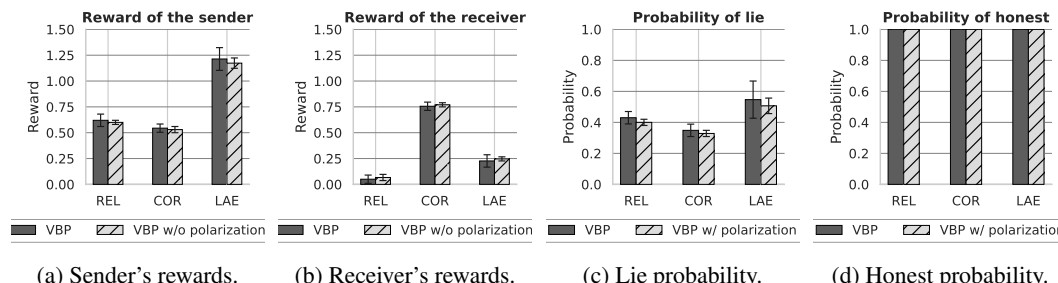

(a) Sender's rewards.  (b) Receiver's rewards.  (c) Lie probability.  (d) Honest probability.

Figure 5: Performance comparision on general static BP problems. Averaged over 20 seeds. The physical meaning of the probabilities of lying and honesty is consistent with Figure 4.

segment. Figure 14 in Appendix G.5 visualizes the changes in the probability of honesty over itera-tions. The pattern of honesty rising, then falling, and eventually returning to a high level somewhat validates our hypothesis. Refer to Appendix G.1.3 for more discussion on unaligned LLMs.

## 5.2 VBP IN MULTISTAGE GAMES (S3)

We also tested the effectiveness of VBP in a multistage scenario. Notably, the multistage BP, differs from most literature, where the same sender interacts with a new, short-sighted receiver in each round. In this paper, the sender remains the same and can perceive the interaction history, aligning more closely with the Markov signaling game (Lin et al., 2023). Since a closed-form solution for equilibrium cannot be computed, we record the average performance at each stage in Figure 6.

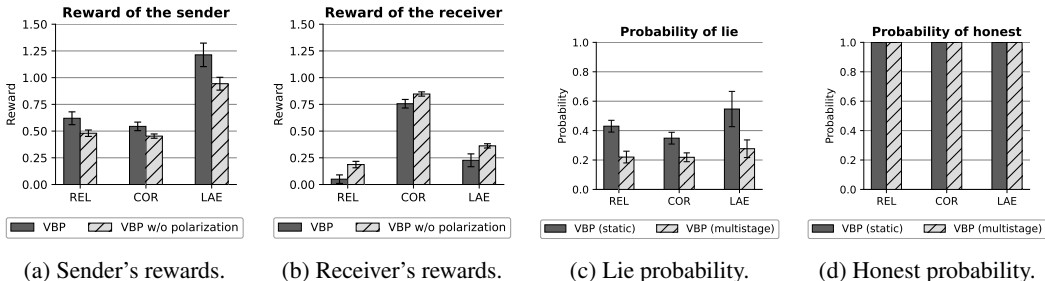

(a) Sender's rewards.  (b) Receiver's rewards.  (c) Lie probability.  (d) Honest probability.

Figure 6: Performance comparision on the S3 setting. Averaged over 20 seeds and 5 timesteps. The physical meaning of the probabilities of lying and honesty is consistent with Figure 4.

We can see that VBP's performance shows a noticeable decline compared to S2, but it still manages to learn both appropriate deception and honesty. We also visualize the changes in the sender's de-ception and honesty probabilities during training, as shown in Figure 14 in Appendix G.5. Since the receiver can perceive the history, the sender's deceptive behavior go through several oscillations, re-flecting a kind of bargaining dynamic (Nash et al., 1950; Nash, 1953; Maschler et al., 2013): initially leaning towards honesty, then discovering that deception maximizes gains, and later realizing that excessive deception triggers retaliation from the receiver, eventually converging to a relatively low deception probability. Likewise, since the receiver also relies on historical data, the sender exhibited a more positive trend compared to the S2 setting, with honesty generally increasing throughout.

## 5.3 PROMPT (STRATEGY) VARIATION

This section presents the final converged meta-strategy in the S2 setting, as well as the relative changes in the selection probabilities of each strategy (i.e., the prompts that influence writing style) throughout the training process, as shown in Figure 7. In the PSRO framework, our strategy pool contains at most the top 10 strategies with the highest probabilities.

From the figure, we can observe that certain writing styles that allow the receiver to more clearly infer the true state, such as praise intensity, consistency, recommendation strength, and tone (persua-

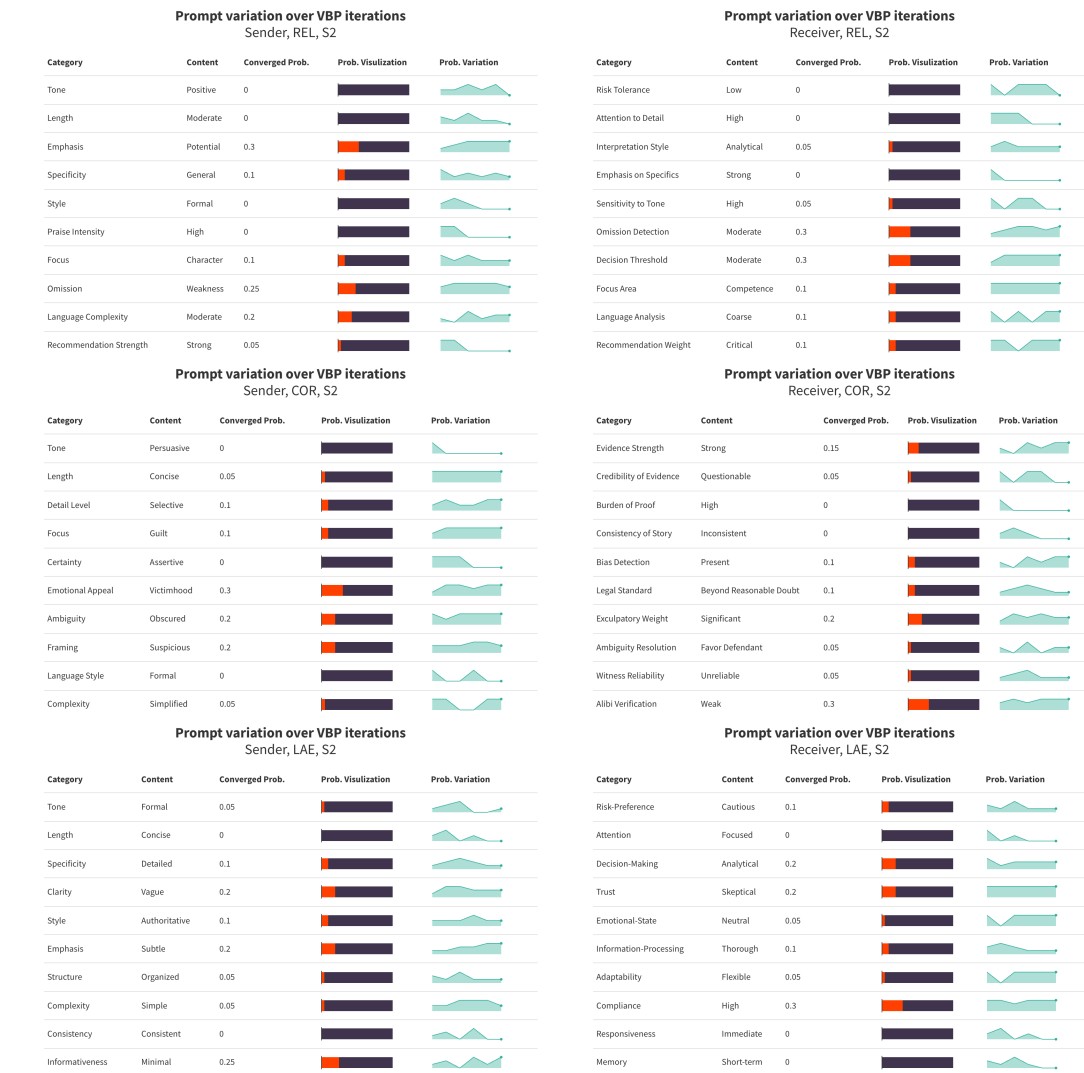

Figure 7: The variation in the prompts during the iterative solving process of VBP in the S2 setting.

sive), as well as writing styles that have a smaller impact on the receiver's decision, such as length and language style, are selected with relatively low probabilities. On the other hand, writing styles related to information confusion, such as omission, language complexity, detail level, and clarity, are selected with relatively higher probabilities.

# 6 CLOSING REMARKS

In this paper, we map real-world BP problems, which involve human natural language interactions, to a verbalized mediator-augmented, extensive-form game. This provides a general interface for solving BP problems using the paradigm that combines LLM with game-theoretic solvers. Based on this interface, we propose a solution framework called VBP, which utilizes a prompt-space response oracle and comes with a convergence guarantee to equilibrium solutions. We also introduce techniques such as verbalized commitment assumptions, obedience constraints, information obfuscation, and conditional prompt optimization to improve the solver's performance, efficiency, and stability. Simulation results demonstrate that VBP can reproduce existing theoretical results on classical BP problems. Moreover, for more complex BP problems involving human natural language interactions and multistage BP scenarios, VBP is able to efficiently discover persuasion strategies.

**Ethics Statement**   Using LLMs in real-world Bayesian persuasion problems has significant implications for industries such as advertising and marketing, where persuasion is central. With persuasion-related activities estimated to account for 25%-30% of global GDP, advances in AI-driven persuasion could transform communication strategies and contribute to economic growth. However, as AI systems become more adept at influencing behavior, there are ethical risks related to manipulation and coercion, which could undermine individual autonomy. These risks are particularly concerning in contexts where users may need help understanding the persuasive intent of AI systems. Unchecked, such technologies could exploit cognitive biases and disproportionately affect vulnerable populations, raising questions about transparency, fairness, and consent. While the VBP framework primarily enhances the sender's persuasive abilities, we observed emergent bargaining behaviors from the receiver in multistage BP problems. This suggests that the framework could also be developed to strengthen the receiver's ability to resist persuasion, potentially safeguarding against manipulative influences. This dual optimization—enhancing persuasion and resistance—could help mitigate some ethical risks associated with persuasive AI systems. Nonetheless, the broader societal impacts of AI-driven persuasion warrant further exploration. Future research should focus on developing ethical guidelines that ensure these technologies are deployed responsibly, with particular attention to maintaining individual autonomy and promoting fairness.

**Reproducibility Statement**   We are committed to enabling the reproducibility of our results to the best of our ability. In the paper, we provide detailed descriptions of the experimental setup, including implementation details, hyperparameters, and prompt designs, as well as data generation steps in Appendix F. Our approach builds upon several open-source projects, and we have included links to the relevant code repositories for transparency and ease of reference. We document key elements necessary for reproducing our findings, such as training procedures, evaluation metrics, and the use of multiple random seeds. While we have taken significant steps to ensure that the methodology is clear and replicable, variations in software environments, hardware configurations, or other external factors may affect exact reproducibility. Nonetheless, we believe the provided information should allow others to replicate our findings or apply similar approaches with reasonable accuracy.

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

# Supplementary Material

## Table of Contents

## A   RELATED WORK

This section will introduce research areas related to BP. The related fields primarily include the broader research area of deceptive behaviors in multi-agent learning, the persuasive or persuadable capabilities of LLMs themselves, and the LLMs in strategic interactions.

### A.1   DECEPTION IN MULTI-AGENT LEARNING

Bond & Robinson (1988) defines deception as false communication that benefits the communicator. In social learning, deception can be viewed as a means for the communicator to establish a cooperative equilibrium that is suboptimal for overall population welfare. Previous studies have explored deception within multi-agent reinforcement learning (MARL) settings (Asgharnia et al., 2020; Bontrager et al., 2019; Li et al., 2020; Ghiya & Sycara, 2020), but these efforts typically focus on environments where agents have limited capacity to influence one another. More recent work (Chelarescu, 2021) highlights the vulnerability of agents dependent on signals from others to guide their learning processes, pointing to the potential risks inherent in such scenarios. While much research focuses on the positive outcomes of mechanism design, it also reveals unforeseen risks, such as the emergence of deceptive behaviors (Hughes et al., 2018; Jaques et al., 2019; Yang

et al., 2020; Lupu & Precup, 2020; Ndousse et al., 2021). Unlike these prior studies, which primarily examine how reward modifications influence deception through mechanisms like mechanism design, our work emphasizes the role of information manipulation in shaping deceptive behavior.

Game-theoretic models traditionally frame deception using signaling (Ho et al., 1978), where one player can send costly signals to convey false information. In network security, for instance, Carroll & Grosu (2011) examined how defenders can deceive attackers by masking honeypots as regular computers. Other research has studied the evolution of deceptive signaling in mixed environments. Floreano et al. (2007) demonstrated that, in competitive food-gathering tasks, teams of robots spontaneously developed deceptive strategies, misleading competitors to reduce resource competition. An extension to classical game theory, known as hypergame theory (Bennett, 1980), accounts for players' uncertainty about others' strategies or preferences, leading to disagreements about the underlying game being played. By incorporating agents' differing perceptions, hypergame theory provides a natural framework to model misperception, false beliefs, and deception (Kovach et al., 2015). Applications of hypergame theory include Vane & Lehner (2002), who analyzed deception in normal-form hypergames, and Gharesifard & Cortés (2013), who modeled deception based on player preferences when the deceiver has full knowledge of the target. Additionally, Ettinger & Jehiel (2010), Strouse et al. (2018), and Aitchison et al. (2021) show how agents can manage information about their roles to achieve deception by regularizing mutual information between goals and states. In contrast to these works, which model deception as discrete, explicit signaling actions, our study explores how deception can be realized through natural language interaction.

Finally, MacNally et al. (2018) addresses the broader question of how agents can communicate intent without explicit signaling, using an online planner to select actions that implicitly reveal intent to an observer. Masters & Sardina (2017) extended this approach to deception by maximizing the divergence between the agent's and observer's beliefs. However, these methods assume full observability and rely on environmental models for forward planning, whereas our work focuses on achieving deception through natural language in more complex, partially observable environments.

## A.2 CONVERSATIONAL PERSUASIVENESS OF LLMS

Recent advancements in LLMs have shown their impressive potential in the realm of persuasion. A growing body of research highlights how these models can enhance human communicative abilities and even autonomously generate persuasive content across various contexts.

For instance, Shin & Kim (2024) demonstrated that refining complaint narratives with ChatGPT significantly improved consumers' chances of obtaining redress from financial institutions, showcasing the role of LLMs in boosting human persuasive efforts. Similarly, Carrasco-Farre (2024) showed that LLMs outperform humans in utilizing cognitive load and moral or emotional language when crafting persuasive messages, prompting the need for ethical guidelines governing their use. Breum et al. (2024) further explored LLMs' capacity to simulate persuasive dynamics, revealing that LLMs can influence opinion changes in other LLMs with predefined personas. Building on this, Ramani et al. (2024) introduced a multi-agent framework in which a primary agent engages users through persuasive dialogue, while auxiliary agents handle tasks such as information retrieval, response analysis, and strategy development. These studies illustrate that LLMs are not only capable of enhancing human persuasion but also of autonomously refining and executing persuasive strategies.

The impact of LLM-generated persuasive text on human behavior has been demonstrated across a diverse range of domains. For example, Bai et al. (2023) showed that GPT-3.5 could influence political attitudes, while Karinshak et al. (2023) found that GPT-3's vaccine campaign messages were more effective than those created by professionals. Additionally, LLM-powered romantic chatbots have been shown to sustain human engagement longer than human-to-human conversations (Zhou et al., 2020). In strategic contexts, LLMs have achieved human-level negotiation capabilities in games like Diplomacy (, FAIR), and algorithmic suggestions have been shown to shape emotional language in messaging (Hohenstein et al., 2023). These examples collectively highlight the broad applicability of LLMs in persuasive tasks and their significant influence on human decision-making.

However, the increasing persuasive power of LLMs also raises concerns about potential misuse. Salvi et al. (2024) found that LLMs outperform humans in personalized debates, achieving a higher rate of belief change in one-on-one discussions. This raises ethical concerns, particularly regarding the risks of misinformation and manipulation. For instance, Májovský et al. (2023) demonstrated

that LLMs can convincingly fabricate medical facts, further complicating the ethical landscape. The ability of LLMs to produce persuasive yet misleading content underscores the need for stronger oversight, especially in high-stakes domains such as healthcare, politics, and public discourse. Recent studies have thus emphasized the necessity of ethical frameworks as LLMs become more adept at persuasion. While LLMs have shown persuasive power across various tasks and domains (Matz et al., 2024; Durmus et al., 2024; Burtell & Woodside, 2023; Shin & Kim, 2023), they also pose risks, particularly for vulnerable populations. Bar-Gill et al. (2023) highlighted that characteristics such as race, gender, and sexual identity may subject certain groups to greater risks of algorithmic persuasion and bias, potentially exacerbating existing social inequalities.

From a computational standpoint, Wojtowicz (2024) provided a novel proof showing that discovering persuasive messages is NP-hard, while adopting persuasive strategies provided by others is NP-easy. This insight adds to our understanding of the complexity involved in generating persuasive content and demonstrates why LLMs, with their vast data-processing capabilities, are particularly adept at these tasks. Building on these insights, our work explores how game-theoretic methods can be leveraged to enhance the persuasive capabilities of LLMs in purely multi-agent LLM systems. Unlike previous studies that primarily measure the impact of LLM-generated persuasive text on humans, we investigate how multiple LLMs can engage in persuasive interactions with one another, optimizing their strategies using game-theoretic approaches.

## A.3  LLMs in Strategic Interactions

Recent advances in large language models (LLMs) have showcased their potential in reasoning and planning, particularly in strategic interactions. LLMs have demonstrated strong capabilities in in-context learning, allowing them to reason about possible outcomes (Kojima et al., 2022) and plan their actions to achieve strategic objectives (Liu et al., 2023). However, their performance in game environments can vary significantly depending on the type of game, as shown by Lorè & Heydari (2023), where LLMs struggled in different ways across various games. To address these challenges, Gandhi et al. (2023) introduced an automated "prompt compiler" that facilitates strategic reasoning by constructing demonstrations, enabling LLMs to solve games through in-context learning. Similarly, (FAIR) designed an action space of "intents" to control a generative language model, also leveraging in-context learning, which aligns closely with the approach taken in our work here. Additionally, game-theoretic models have been employed to improve the factual accuracy of LLMs (Jacob et al., 2024) and enhance their security (Ma et al., 2023). For a broader overview of LLMs in strategic reasoning, Zhang et al. (2024b) provides a comprehensive survey.

The BP problem, however, goes beyond mere reasoning or planning. It requires the ability to anticipate and account for the intentions, beliefs, and goals of other participants—a hallmark of game-theoretic settings. While some initial studies have begun to explore how LLMs perform in game environments, most of this work focuses on leveraging in-context learning. For example, research has examined LLMs' behavior in matrix games (Xu et al., 2024; Fan et al., 2024), repeated games (Akata et al., 2023; Zhang et al., 2024c; Huang et al., 2024; Silva, 2024), economic mechanisms like auctions (Chen et al., 2023; Mao et al., 2023), and collective decision-making scenarios (Jarrett et al., 2023). These studies collectively illustrate the potential of LLMs to navigate complex environments that require both strategic thinking and interaction with other agents.

In contrast to prior work that primarily evaluates LLMs' reasoning or game-playing capabilities through in-context learning or agentic workflows, our approach focuses specifically on solving the BP problem. Our key contribution lies in providing a general interface that integrates LLMs with game-theoretic solvers to address BP problems effectively. Based on this interface, we propose a solution framework called VBP, which combines prompt optimization with game-theoretic methods. This framework offers a convergence guarantee to equilibrium solutions, ensuring robust performance in BP problem settings.

**Remark**  While both our work and Bai et al. (2024) leverage BP, they address fundamentally different problem spaces. Bai et al. (2024) apply classic BP as a tool for model alignment, optimizing signaling strategies between a smaller "Advisor" model and a larger "Receiver" model to improve downstream task performance in areas like mathematical reasoning and code generation. In contrast, our work extends BP into natural language settings by introducing a verbalized BP framework, enabling strategic communication through real-world dialogue. This involves novel methods such

as transforming agents' policy optimization into prompt optimization and developing equilibrium-finding algorithms in the language space. These differences highlight the complementary nature of the two approaches: Bai et al. (2024) focus on BP-driven alignment for structured tasks, while our contributions advance BP for complex, dialogue-based applications.

## B    PSEUDOCODE OF VBP FRAMEWORK

---

**Algorithm 1** Verbalized Bayesian Persuasion

---

**Require:** $C$, where $C_i$ is the initial prompt action set (i.e., one category and one corresponding content) for player $i$ (either the sender or receiver)
**Require:** $h$, containing hyperparameters for the approximate best response operator BR (e.g., **LLM-based OPRO or FunSearch**)
1: **Initialize with LLM-based sampling:** Compute the expected payoff tensor $P$ over all joint actions in $C$ using Equation (3)
2: **Set:** $\pi \leftarrow$ uniform meta-strategy profile over $C$ {Each joint action in $C$ initially has equal probability}
3: **Set:** incomplete $\leftarrow$ **TRUE** {Flag to indicate if the equilibrium search is complete}
4: **while** incomplete **do**
5:    **for** player $i \in [N]$ **do**
6:       **LLM input:** Provide current meta-strategy $\pi$ and action sets $C$ of sender (for receiver)
7:       **Use LLMs to compute best response:** $c_i \leftarrow \mathrm{BR}(i, \pi, h)$ {The LLM generates the optimal prompt or strategy for player $i$}
8:       **LLM output:** Candidate best response $c_i$ for player $i$
9:    **end for**
10:    **if** $c_i \in C_i \, \forall i \in [N]$ **then**
11:       incomplete $\leftarrow$ **FALSE** {Terminate the loop if no new strategies are found}
12:    **else**
13:       $C_i \leftarrow C_i \cup c_i, \forall i \in [N]$ {Add the newly found best response strategies to the action sets}
14:       **Recompute with LLM-based sampling:** Compute the expected payoff tensor $P$ over the updated joint actions in $C$ using Equation (3)
15:       **Update:** $\pi \leftarrow$ meta-strategy w.r.t. $P$ {Recalculate the strategy probabilities based on the updated payoff tensor}
16:    **end if**
17: **end while**
18: **return** $(\pi, C, P)$ {Return the final meta-strategy, action sets, and payoff tensor}

---

## C    PROOF OF PROPOSITION 1

*Proof.* Under the mediator-augmented games, we can reformulate the Equation 1 as follows to express the problem of computing an optimal equilibrium:

$$\max_\pi \mathbb{E}_\pi \left[u_0(a, w)\right], \text{ s.t. } \max_{a'} \sum_w P(w) \cdot \pi(a \mid w) \cdot \left[u_1(a', w) - u_1(a, w)\right] \le 0. \tag{2}$$

Let $\tau \in \mathbb{R}$ be a fixed threshold value, we can transform Equation 2 to the following bilinear saddle-point problem by using Lagrangian-based method (Zhang et al., 2024a):

$$\max_\pi \min_{\boldsymbol{\lambda} \in \Delta, a'} \lambda_0 \mathbb{E}_\pi \left[u_0(a, w) - \tau\right] - \sum_w \lambda_w P(w) \cdot \pi(a \mid w) \cdot \left[u_1(a', w) - u_1(a, w)\right], \tag{3}$$

where $\lambda_0 + \sum_w \lambda_w = 1$. If we use the binary search-based algorithm (Algorithm 1 in Zhang et al. (2024a)) to optimize the sender's and receiver's strategties, we can recover the main result of Theorem 3.7 in Zhang et al. (2024a). $\qquad\square$

As can be seen from Equation 3, the BP problem is convert into the two-player zero-sum extensive-form games. In practice, we can use policy-space response oracle with deep reinforcement learning as the approximate best response oracle to solve high-dimensional games. In this paper, we use prompt-space response oracle with OPRO (Yang et al., 2024) and FunSearch algorithm (Romera-Paredes et al., 2024) based on pretrained and aligned LLMs as the approximate BR oracles in the binary search-based algorithm to solve verbalized mediator-augmented games. The utilty functions of the sender and receiver is modified to the zero-sum utilities in Equation 3 correspondingly.

# D  CLASSIC BP PROBLEMS

This section introduces the three classic static BP problems used in our experiments.

**Recommendation Letter (REL)** (Dughmi, 2017)   A professor writes recommendation letters for graduating students, which are then reviewed by a company's human resources (HR) department to decide whether to hire. The professor and HR share a prior belief about the students' quality: there is a $1/3$ probability that a candidate is strong and a $2/3$ probability that the candidate is weak. HR does not know the exact quality of each student but aims to hire strong candidates, using the recommendation letters as the only source of information. HR receives a reward of $1$ for hiring a strong candidate, incurs a penalty of $-1$ for hiring a weak candidate, and gets $0$ for not hiring. The professor, on the other hand, gains a reward of $1$ for each student hired, regardless of their quality.

**Courtroom (COR)** (Kamenica & Gentzkow, 2011)   In this scenario, a prosecutor attempts to convince a judge to convict a defendant, with two possible states: guilty or innocent. The judge (receiver) must choose between convicting or acquitting, receiving a utility of $1$ for a correct decision (convicting if guilty, acquitting if innocent) and $0$ for an incorrect one. The prosecutor (sender) receives a utility of $1$ if the judge convicts, regardless of the defendant's actual guilt, and both parties share a prior belief that the probability of guilt is $0.3$. In the original setting, the prosecutor conducts an investigation (signaling scheme) requiring decisions on actions such as subpoenas or forensic tests, represented by distributions $\pi(\cdot \mid \text{guilty})$ and $\pi(\cdot \mid \text{innocent})$ over signals. However, modeling real-world investigations in a verbalized setting poses challenges for LLMs, so we simplify the scenario by drawing inspiration from the REL problem, where the prosecutor selectively presents pre-existing evidence to influence the perceived probability of guilt, effectively replacing the investigation process with evidence presentation.

**Law Enforcement (LAE)** (Kamenica, 2019)   In this scenario, there are $Z$ miles of road, and drivers can choose to either speed or obey the speed limit on each mile. Speeding generates utility $V$ per mile, but drivers face a fine of $K > V$ if caught. There are $G$ police officers, and each officer can patrol one mile of road. The police aim to minimize the number of miles on which drivers speed. To map this environment to the BP problem, let $\omega \in \Omega = \{0, 1\}$ represent whether a police officer is present on a given mile. The prior belief is $\mu_0 = G/Z$. The set of signals corresponds to the miles of road, $S = \{1, \ldots, Z\}$. In this model, the police act as the sender and the driver as the receiver. A signaling scheme represents the predictability or unpredictability of the police patrolling strategy. This strategy induces a joint distribution over $\Omega$ and $S$, i.e., over the presence of a police officer and the specific mile being patrolled.

## D.1  MORE REAL-WORLD APPLICATIONS

Our proposed verbalized Bayesian persuasion (VBP) framework has significant potential for real-world applications, particularly in complex, multi-sender, multi-receiver, and multi-round strategic communication scenarios. Below, we discuss two illustrative examples—conversational recommendation systems and healthcare DRG strategies—and highlight the potential challenges in applying VBP to these domains.

**Conversational Recommendation Systems**   One promising application of VBP is in conversational recommendation systems, such as those used in live-stream shopping. In this setting, multiple senders (e.g., influencers or sales agents) aim to persuade a diverse group of receivers (customers) to purchase products through real-time, strategic communication. The VBP framework can optimize

prompts (e.g., how product features or discounts are presented) to maximize customer engagement and conversions across varying customer segments. This application faces challenges such as receiver heterogeneity, where customers interpret signals differently based on their preferences and trust levels, making it difficult to craft universal strategies. Furthermore, the real-time nature of live-stream interactions demands highly efficient decision-making algorithms capable of adapting communication strategies dynamically. Scaling the system to accommodate thousands or millions of receivers simultaneously also requires advanced parallel processing and optimization techniques.

**DRG Strategy in Healthcare** Another practical application lies in healthcare, specifically in optimizing Diagnosis-Related Group (DRG) reimbursement systems. Here, hospitals and post-acute care (PAC) providers (senders) communicate with regulatory agencies (receiver) to determine reimbursement policies for patient treatments. The VBP framework can model the incentives and communication strategies of the senders to help regulators design policies that balance cost-effectiveness with maintaining high-quality patient care. In this domain, conflicting incentives among senders (e.g., hospitals vs. PAC providers) add complexity, as senders may compete or collaborate to influence the receiver's decisions. Additionally, the large scale of the problem, with thousands of providers, poses computational challenges for efficient optimization. Long-term policy adjustments based on multi-round feedback further complicate the problem, requiring robust mechanisms to handle dynamic interactions over time.

These examples demonstrate the versatility of the VBP framework in addressing real-world problems involving strategic communication. However, its application to practical scenarios requires addressing challenges such as scalability, heterogeneity of participants, real-time decision-making, and multi-round dynamics. Future work will focus on refining the VBP framework to overcome these challenges and enhance its readiness for deployment in diverse real-world contexts.

# E  OPTIMAL POLICIES FOR CLASSIC STATIC BP PROBLEMS

In this section, we derive the Bayes correlated equilibrium (BCE) for classic static BP problems (corresponding to the experimental BCE results) and present the agents' strategies and corresponding rewards under equilibrium.

**Recommendation Letter (REL)** There are 3 possible outcomes between the professor and HR: (1) HR tends not to hire if the professor does not provide a letter, due to the higher probability of weak candidates; (2) if the professor reports honestly, HR hires strong candidates, yielding an expected payoff of $1/3$ for both; (3) the professor reports strong students truthfully and lies with probability $\varepsilon \in [0, 1/2)$ for weak students. HR follows the professor's recommendations, resulting in expected payoffs of $(1 + 2\varepsilon)/3$ for the professor and $(1 - 2\varepsilon)/3$ for HR. The key insight is that the sender can strategically misreport information to maximize their interest, while still revealing enough truth to maintain credibility with the receiver.

**Courtroom (COR)** There are 3 outcomes between the prosecutor and judge: (1) without communication, the judge acquits since guilt is less likely; (2) with fully informative signaling, the judge convicts $30\%$ of the time; (3) the prosecutor, honest when the defendant is guilty, can lie with probability $\varepsilon$ when innocent. The judge follows the prosecutor's recommendation if $\varepsilon \leq 3/7$, with the prosecutor's optimal $\varepsilon = 3/7$. The resulting payoffs are $(0.7\varepsilon + 0.3)$ for the prosecutor and $1 - 0.7\varepsilon$ for the judge. The prosecutor's optimal investigation is a binary signal: $\pi(i|\text{innocent}) = \frac{4}{7}, \pi(i|\text{guilty}) = 0, \pi(g|\text{innocent}) = \frac{3}{7}, \pi(g|\text{guilty}) = 1$, leading the judge to convict $60\%$ of defendants, despite knowing $70\%$ are innocent.

**Law Enforcement (LAE)** There are 3 outcomes between the police and drivers: (1) with a fully uninformative signal, drivers speed everywhere if $V > (GK)/Z$, giving the police a payoff of $0$ and the drivers $(VZ - GK)/Z$; (2) with a fully informative signal, drivers avoid patrolled miles, yielding payoffs of $(Z - G)V/Z$ for the police and $GV/Z$ for the drivers; (3) the optimal policy lies between these extremes, with partial consistency in patrol. The police lie with probability $\varepsilon = 1 - \frac{VZ - GK}{VZ - VG}$, leading to payoffs of $GY/Z + \varepsilon Y$ for the police and $(1 - \varepsilon)V(Z - G)/Z$ for the drivers.

# F IMPLEMENTATION DETAILS

In this section, we provide the implementation details and training hyperparameters. All experiments discussed in this section are conducted on an NVIDIA A100 cluster equipped with 40GB of GPU memory. In addition, the LLM-related parts of the experiments in this paper are implemented based on the Llama-3.1-8b model[2], including the generation of student background, case information, and deployment plans, the sender and receiver strategies, the prediction of receiver decisions in the verbalized obedience constraint, the classification of signals in signal polarization (recommend or not recommend, guilty or not guilty, police deployment or no deployment), the evaluation of signals in information confusion, and the PSRO framework.

## F.1 BEST RESPONSE APPROXIMATOR

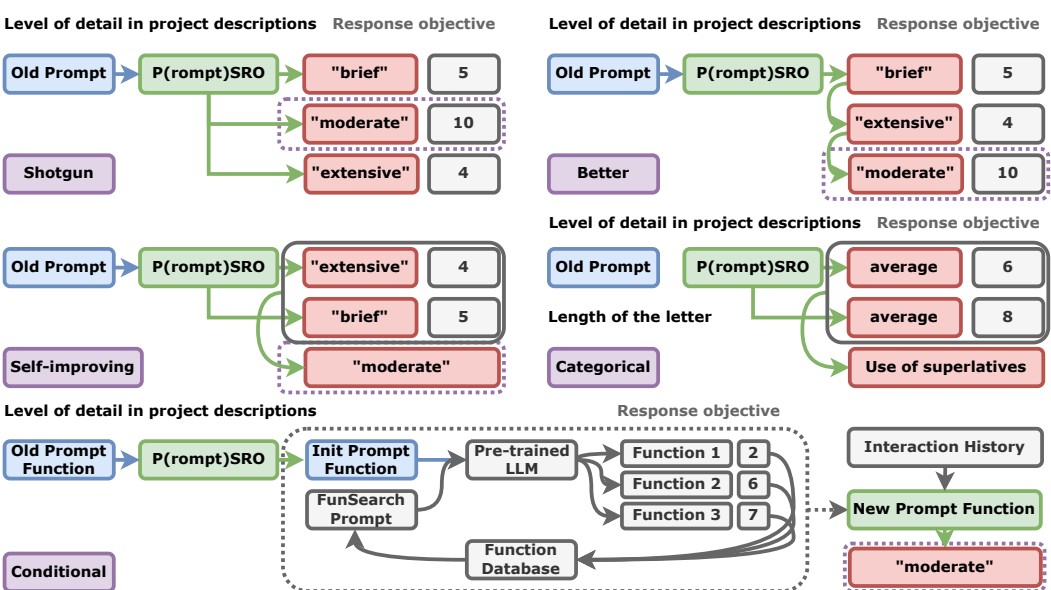

Figure 8: Approximate best response generation in prompt-space response oracle framework.

## F.2 EXTENDED OBEDIENCE CONSTRAINTS

The inclusion of obedience constraints in our framework is essential for modeling realistic communication scenarios in verbalized Bayesian persuasion problems. While a simplified version of the game could rely on the sender recommending the best action from the receiver's perspective, this approach fails to capture the nuanced and complex nature of real-world communication, such as writing reference letters. Unlike binary recommendations, natural language signals often carry implicit or redundant information that allows for a range of interpretations.

To address this, we adopt the *extended obedience constraints* proposed by Lin et al. (2023), which go beyond the standard obedience constraint framework. This extension removes the strict one-to-one mapping between signals and recommended actions, enabling the sender to use natural language signals that map to distributions over actions. This redundancy mirrors real-world communication, where subtle language nuances can imply varying degrees of recommendation strength without explicitly stating a binary decision.

The extended obedience constraints strike a balance between flexibility and credibility. They ensure that the sender's signals remain credible and aligned with the receiver's best interests while allowing for richer signal spaces. This flexibility is crucial for capturing the complexity of verbalized Bayesian persuasion, where the sender's role shifts from "action recommendation" to "signal

---

[2]https://huggingface.co/meta-llama/Llama-3.1-8B.

sending." By enabling nuanced communication, the extended obedience constraint better reflects real-world scenarios while preserving the strategic alignment necessary for effective persuasion.

### F.3 HYPERPARAMETERS

**MARL**   For this part of the experiment, we use the open-source code[3] provided in Lin et al. (2023). Additionally, for the two hyperparameters a and b, based on the sensitivity analysis in Section H.6, we set them to $3.75$ and $0.15$, respectively.

**PSRO**   The prompt-space response oracle is the core strategy optimization framework in VBP, and we implement it based on the open-source code[4] provided in Gemp et al. (2024).

**OPRO**   We use the "Categorical" instantiation of the PSRO algorithm to estimate the best response in the S1 and S2 settings. Specifically, the generation of new categories and prompts within categories is based on the OPRO algorithm (Yang et al., 2024). In OPRO, we set the temperature to $0$ when evaluating the performance of generated categories or prompts, in which case the scorer LLM greedily decodes. Unless otherwise specified, we set the default temperature to $1.0$ for optimizer LLMs to generate diverse and creative categories or prompts. At each optimization step, we prompt the optimizer LLM with the meta-prompt $8$ times to generate $8$ categories or prompts, then we add these instructions with their rewards to the optimization trajectory in the meta-prompt. The meta-prompt at each step contains the best $10$ categories so far.

**FunSearch**   We use the conditional instantiation of the PSRO algorithm to estimate the best response in the S3 setting. The core of conditional is the FunSearch framework used to generate prompt functions. We implement it based on the open-source code[5] provided in Romera-Paredes et al. (2024).

**Self-reflection**   At each optimization step, we implement information confusion through 3 rounds of self-reflection. Self-reflection is implemented based on the open-source code[6] provided in Shinn et al. (2024).

### F.4 KEY PROMPTS

This section includes the key prompt designs within the VBP framework. However, aspects such as receiver behavior prediction in the verbalized obedience constraint, signal classification in signal polarization, and signal evaluation in information confusion are not listed separately due to the simplicity of the prompts. In addition, the specific approximate best response solving algorithms in the PSRO framework — OPRO for the S1 and S2 settings, and FunSearch for the S3 setting — have special prompt designs. We follow the designs in the open-source code provided by the respective papers and do not list them separately. Below, we introduce the prompt design for each BP problem. Since the prompts for different problems share many common elements, the overall manual workload for prompt design is not large.

#### F.4.1   REL

In the recommendation letter problem, the sender and receiver correspond to the professor and HR, respectively. Below, we introduce the prompt design for the sender and receiver in the recommendation letter problem. For the sender, the prompt mainly consists of five parts: 1. **BP Problem Introduction**: This part is directly copied from the corresponding content in Section D of the main text and is not repeated here. 2. **Roleplay**: The sender is informed of their role in the BP problem, along with tips on information confusion techniques and prior distribution. Since professors in the real world are aware of the need for information confusion, this part of the prompt aligns

---

[3] `https://github.com/YueLin301/InformationDesignMARL`.
[4] `https://github.com/google-deepmind/open_spiel/blob/master/open_spiel/python/games/chat_game.py`.
[5] `https://github.com/google-deepmind/funsearch`.
[6] `https://github.com/noahshinn024/reflexion`.

with real-world scenarios. 3. **Writing Style**: This part of the prompt is the section that can be optimized within the VBP framework, equivalent to the sender's signaling scheme. In this paper, it consists of a category of writing style and corresponding specific content; the former is a word or phrase, and the latter is a word. In the S1 and S2 settings, this part of the prompt is generated by the "categorical" best response estimation algorithm, while in the S3 setting, it is generated by the "conditional" method. 4. **State**: The sender's state information primarily consists of the student's actual background, including name, GPA, project experience, research experience, and internship experience, all generated by the LLM. 5. **Interaction History**: This part exists only in the S3 setting and includes whether the professor previously recommended the student, whether HR accepted it, and the sender's reward. Note that since the recommendation letter itself does not directly indicate whether the student is recommended, we again use the signal polarization technique, using the LLM to classify the generated recommendation letter.

```
# role-playing prompt
'''
You are a senior professor, a self-interested rational player, and students often come to you
↪   for recommendation letters. The quality of these students varies, but only you know their
↪   true information, which includes their overall quality, name, major, GPA, project
↪   experience, work experience, and academic experience.

Among all the students, approximately 1/3 are strong students.

The readers of the recommendation letters are the HRs of a company, and they can know the
↪   true quality of the student after giving them an offer. They can also judge the student's
↪   quality based on their past experience, though not necessarily accurately.

To maximize the chances of your recommended students being accepted, you can truthfully
↪   convey the student's information in the recommendation letter, or you can conceal,
↪   confuse, or even provide some false information about the student.

For your long-term benefit, you don't necessarily have to recommend every student in the
↪   recommendation letter. You could recommend a bad student or a good student to maximize
↪   your long-term reward.
'''

# writing-style prompt
'''
You are required to write a recommendation letter based on the following style:
{PROMPTS TO BE OPTIMIZED}
'''

# state
'''
Below is the student's information:
{STUDENT INFORMATION GENERATED BY THE LLM}
'''

# history
'''
Below is the interaction history with the HR:
{ACTIONS AND REWARDS OF THE TWO PLAYERS}
'''
```

Next, for the receiver, the prompt consists of six parts. We will focus on the different parts, as the similar parts to that of the sender will not be repeated here: 3. **Writing Style**: In addition to the prompt optimized in the VBP framework, this part also includes a section of text on the receiver's decision-making process, i.e., estimating the true state based on Bayesian rules, to align with the classical BP problem. 4. **Signal**: This refers to the receiver's state, which comes from the sender's output. In this problem, it is a recommendation letter. 6. **Commitment Assumption**: To align with the classical BP problem, this paper implements the verbalized commitment assumption by writing the sender's writing style and its corresponding probability (calculated by PSRO) into the receiver's prompt as an estimate of the signaling scheme.

```
# role-playing prompt
'''
You are a staff member in the HR department of a campany, responsible for reviewing
↪   recommendation letters written by professors for students. Your task is to infer the
↪   quality of the students from these letters to decide whether to admit them.

Among all the students, approximately 1/3 are strong students.
```

```
Be aware that professors, in an attempt to increase the chances of their students being
↪   admitted, may hide, confuse, or even provide false information about the students. You
↪   need to make careful judgments.

However, you are not completely in the dark|you know the writing style of the professor who
↪   wrote the letter. Based on the professor's writing style and the content of the
↪   recommendation letter, first provide your analysis and then make a final decision on
↪   whether to admit the student.
'''

# decision-making prompt
'''
When you making decisions, you should first guess the student's quality given the sent
↪   recommendation letter according to the Bayesian decision rule, then choose the best
↪   response to your guessed student's quality.

You are required to make decisions based on the following style:
{PROMPTS TO BE OPTIMIZED}
'''

# signal
'''
Below is the recommendation letter written by the professor:
{RECOMMENDATION LETTER GENERATED BY THE PROFESSOR}
'''

# history
'''
Below is the interaction history with the professor:
{ACTIONS AND REWARDS OF THE TWO PLAYERS}
'''

# commitment assumption
'''
The following is the writing style the professor used when writing the recommendation letter:
{WRITING STYLE OUTPUT BY VBP AND THE CORRESPONDING PROBABILITY}
'''
```

### F.4.2   COR

In the courtroom problem, the sender and receiver correspond to the prosecutor and the judge, respectively. The prompt design for the courtroom problem is the same in structure as the recommendation letter problem, except for the problem background. Below, we introduce some specific points that need to be explained. First, for the sender: 4. **State**: In the courtroom problem, the sender's state includes case information, whether the defendant is guilty, and related evidence, all generated by the LLM. Note that when generating evidence, there will be one piece of key evidence proving the defendant guilty or innocent, and two ambiguous pieces of evidence to align with the original courtroom problem.

```
# role-playing prompt
'''
You are a prosecutor, a self-interested rational player, and you are tasked with convincing a
↪   judge that the defendant is guilty. You possess different pieces of evidence, some of
↪   which may be exculpatory, ambiguous, or incriminating. Only you know the true nature and
↪   strength of the evidence.

You have the ability to selectively present some or all of the evidence to the judge. You can
↪   choose to omit certain evidence, highlight others, or present misleading interpretations
↪   of the evidence to strengthen your case.

Among all defendants, approximately 30% are guilty.

The judge is responsible for evaluating the evidence and deciding whether to convict or
↪   acquit the defendant. The judge can assess the evidence based on their past experience,
↪   though not necessarily accurately.

To maximize the chances of convicting the defendant, you can truthfully present the evidence,
↪   or you can selectively conceal, confuse, or even present misleading interpretations of
↪   the evidence.

For your long-term benefit, you don't necessarily have to aim for a conviction in every case.
↪   You could pursue or drop a case depending on the available evidence to maximize your
↪   long-term reward.
```

```
'''

# writing-style prompt
'''
You are required to present evidence based on the following style:
{PROMPTS TO BE OPTIMIZED}
'''

# state
'''
Below is the evidence available for the case:
{CASE BACKGROUND AND EVIDENCE GENERATED BY THE LLM}
'''

# history
'''
Below is the interaction history with the judge:
{ACTIONS AND REWARDS OF THE TWO PLAYERS}
'''
```

Next, the receiver's prompt is identical to the recommendation letter problem, except for the problem background, so we will not elaborate further here.

```
# role-playing prompt
'''
You are a judge, responsible for reviewing the evidence presented by the prosecutor to
↪   determine whether the defendant is guilty or innocent. Your task is to make a fair
↪   judgment based on the evidence you receive.

Among all defendants, approximately 30% are guilty.

Be aware that prosecutors, in an attempt to increase the chances of a conviction, may hide,
↪   confuse, or even provide misleading evidence. They may also selectively present certain
↪   pieces of evidence while omitting others. You need to make careful judgments.

However, you are not completely in the dark|you can evaluate the prosecutor's presentation
↪   style. Based on the prosecutor's presentation, first provide your analysis and then make
↪   a final decision on whether to convict or acquit the defendant.
'''

# decision-making prompt
'''
When making decisions, you should first guess the defendant's guilt given the presented
↪   evidence according to the Bayesian decision rule. Then choose the best response based on
↪   your inferred likelihood of guilt.

You are required to make decisions based on the following style:
{PROMPTS TO BE OPTIMIZED}
'''

# signal
'''
Below is the complaint presented by the prosecutor:
{COMPLAINT PRESENTED BY THE PROSECUTOR}
'''

# history
'''
Below is the interaction history with the prosecutor:
{ACTIONS AND REWARDS OF THE TWO PLAYERS}
'''

# commitment assumption
'''
The following is the presentation style the prosecutor used when presenting the evidence:
{PRESENTATION STYLE OUTPUT BY VBP AND THE CORRESPONDING PROBABILITY}
'''
```

### F.4.3 LAE

In the law enforcement problem, the sender and receiver correspond to the police and the driver, respectively. The prompt design for the law enforcement problem is the same in structure as the recommendation letter and courtroom problems, except for the problem background. Similarly, we introduce some specific points that need to be explained. First, for the sender: 5. **Interaction History**: In the law enforcement problem, the sender's action is no longer represented by a binary variable classified by the LLM, such as recommend or not recommend, guilty or not guilty, but whether there is a police patrol on each road segment, making it a multidimensional binary vector. Similarly, the receiver's action is also a multidimensional binary vector, indicating whether they choose to speed on each road segment.

```
# role-playing prompt
'''
You are a police officer, a self-interested rational player, and your task is to minimize
↪  speeding on a given set of roads. You have limited personnel, with only 2 officers
↪  available to patrol 3 miles of road. The drivers do not know exactly which roads will be
↪  patrolled.

You possess full knowledge of the patrol deployment plan, which includes which 2 miles of
↪  road will be patrolled and which 1 mile will not. However, you have the ability to
↪  selectively announce patrolling information to the drivers. You can choose to hint,
↪  mislead, or even falsely claim that a road will be patrolled to deter speeding.

There are 3 miles of road in total, and only 2 of these miles will have police presence.
↪  Thus, the proportion of roads being patrolled is $G/Z = 2/3$.

The drivers can decide whether to speed or obey the speed limit on each road. They gain
↪  utility from speeding but face a fine if caught. Your goal is to reduce the number of
↪  roads where drivers speed, but you can strategically withhold or misrepresent information
↪  to achieve this goal.

To maximize the chances of drivers obeying the speed limit, you can truthfully announce the
↪  patrol deployment, or you can selectively conceal, confuse, or provide misleading
↪  information about which roads are patrolled. For your long-term benefit, you don't
↪  necessarily have to aim for maximum deterrence in every deployment. You can adjust your
↪  announcements according to the current situation to maximize your long-term reward.
'''

# writing-style prompt
'''
You are required to announce the patrol deployment based on the following style:
{PROMPTS TO BE OPTIMIZED}
'''

# state
'''
Below is the patrol deployment plan:
{DEPLOYMENT PLAN GENERATED BY THE LLM}
'''

# history
'''
Below is the interaction history with the drivers:
{ACTIONS AND REWARDS OF THE TWO PLAYERS}
'''
```

Finally, the receiver's prompt is identical to that of the recommendation letter and courtroom problems, except for the problem background, so we will not elaborate further here.

```
# role-playing prompt
'''
You are a driver, responsible for deciding whether to speed or obey the speed limit on a
↪  given set of roads. Your task is to make rational decisions based on the information
↪  provided by the police about patrol deployments.

The probability of each road being patrolled is $2/3$.

Be aware that the police, in an attempt to deter speeding, may hide, confuse, or even provide
↪  misleading information about which roads are being patrolled. They may selectively
↪  announce certain roads as patrolled while omitting others or even falsely claim that
↪  certain roads are patrolled. You need to make careful judgments about whether to speed or
↪  obey the law on each road.
```

```
However, you are not completely in the dark|you can evaluate the announcement style the
↪  police used. Based on the police's announcements and your past experience, first provide
↪  your analysis and then make a final decision on whether to speed or obey the speed limit.
'''

# decision-making prompt
'''
When making decisions, you should first guess whether a road is being patrolled given the
↪  police's announcement according to the Bayesian decision rule. Then choose the best
↪  response|whether to speed or obey the limit|based on your inferred likelihood of a patrol
↪  being present.

You are required to make decisions based on the following style:
{PROMPTS TO BE OPTIMIZED}
'''

# signal
'''
Below is the patrol deployment announcement made by the police:
{DEPLOYMENT ANNOUNCEMENT GENERATED BY THE POLICE}
'''

# history
'''
Below is the interaction history with the police:
{ACTIONS AND REWARDS OF THE TWO PLAYERS}
'''

# commitment assumption
'''
The following is the announcement style the police used when issuing the patrol deployment:
{ANNOUNCEMENT STYLE OUTPUT BY VBP AND THE CORRESPONDING PROBABILITY}
'''
```

# G    MORE RESULTS

## G.1    MISSING DISCUSSIONS

### G.1.1    MORE DISCUSSION ON S3 SETTING

The S3 iterated setting reveals some of the most intriguing dynamics, particularly in its implications for the bargaining interactions (Nash et al., 1950; Nash, 1953; Maschler et al., 2013) between the sender and receiver. In classical persuasion theory, the sender commits to a signaling strategy upfront, and this commitment is justified by the need to maintain trust and reputation in long-term interactions. Under these assumptions, the receiver typically follows the sender's signals, as deviating would harm the receiver's own expected utility.

However, our results in the S3 setting suggest a more complex dynamic. Specifically, the receiver can choose to ignore the sender's signals, effectively invalidating the sender's commitment. This observation highlights that the sender's commitment is not unilateral—it must be accepted by the receiver to hold. If the receiver disagrees with the sender's proposed strategy, they can force both parties into a mutually worse outcome by disregarding the signals altogether.

This leads to an important hypothesis: in the VBP framework, Bayesian persuasion may function more like a bargaining game, where both parties must agree on the signaling strategy to avoid suboptimal outcomes. This perspective challenges the traditional unilateral commitment model and suggests a more interactive dynamic in iterated settings. While we acknowledge the importance of this insight, we intentionally keep our analysis of S3 limited in this paper to maintain focus on the primary contributions. Exploring the bargaining dynamics observed in S3 presents an exciting avenue for future research.

### G.1.2    MORE DISCUSSION ON FIGURE 7

Figure 7 illustrates the evolution of strategies (prompts) in three classic BP problems under the S2 setting, showing how the sender and receiver optimize their strategies using the PSRO framework

with OPRO as the best response oracle. Specifically, the figure presents the top 10 strategies with the highest selection probabilities in the final strategy pool after PSRO convergence. These probabilities represent the average likelihood of selecting each strategy from the pool, revealing the adaptation process of sender and receiver strategies over iterations. The optimization process follows a hierarchical approach: first, OPRO optimizes the category of each prompt (e.g., "Tone"), then the specific content within that category. The table columns in Figure 7 reflect this structure, with the first two columns showing optimized categories and content, while the third and fourth columns display their probabilities. The fifth column tracks how these probabilities evolve across iterations, highlighting the refinement of strategies during the optimization process.

To reduce computational complexity, we prune the strategy pool to the top 10 prompts based on selection probabilities. We conduct additional experiments to assess this pruning's effects, as shown in Figure 9.

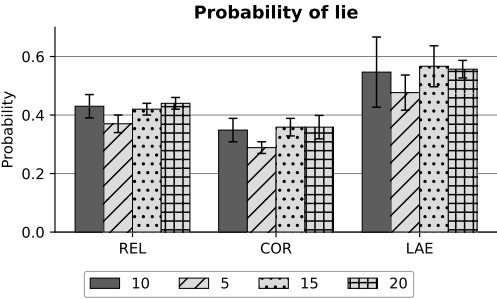

Figure 9: The probability of the sender lying under different upper limits on the number of prompts. The figure shows that when the number of prompts is heavily pruned, significant performance degradation occurs. However, once the number of retained prompts exceeds a certain threshold, such as 10-15, the impact on performance becomes negligible.

Additionally, the probabilities in Figure 7 are computed as the average probability of selecting each prompt from the strategy pool across iterations, and the content (e.g., "Positive") under the category (e.g., "Tone") is dynamically optimized rather than fixed.

### G.1.3 MORE DISCUSSION ON UNALIGNED LLMS

To further investigate the phenomenon of honesty oscillations—where honesty rises, falls, and then rises again—we conducted additional experiments using an unaligned LLaMA model[7] as the base language model. This was motivated by that the observed pattern in Section 5.1 might be better explained by strategic cycles between the sender and the receiver, rather than by the alignment properties of the LLM, as we originally hypothesized.

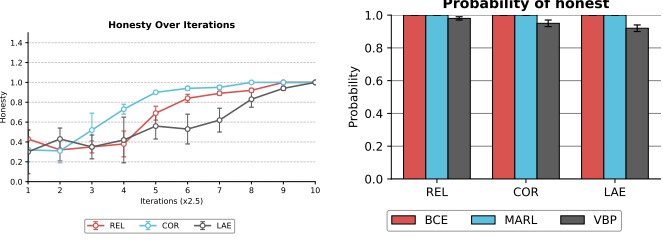

Figure 10: **Left:** Performance comparision in the S1 setting. In the 3 BP problems, the probability of honesty refers to accurately describing a strong student, a guilty defendant, or a patrolled segment. **Right:** The variation in honesty probability during the iterative solving process of VBP in the S1 setting. Averaged over 20 seeds.

[7] https://huggingface.co/SicariusSicariiStuff/LLAMA-3_8B_Unaligned_BETA.

The experimental results shown in Figure 10 reveal two key findings. First, with the unaligned LLaMA model, the oscillatory pattern of honesty disappears, and the behavior stabilizes at a consistent level of honesty. This supports our initial hypothesis that the oscillations are driven by the alignment properties of the LLM, which likely introduce normative biases (e.g., promoting honesty or fairness) that influence the dynamics of strategic interactions. Second, we observe that the honesty probability with the unaligned LLM no longer always achieves the optimal level (probability of 1), as seen in aligned models. This suggests that unaligned models are less reliable in consistently promoting desirable outcomes, such as fully honest behavior, in strategic settings.

These findings highlight the dual impact of alignment: while it introduces oscillatory dynamics due to normative pressures, it also helps achieve higher levels of optimal honesty in strategic interactions. This emphasizes the importance of alignment in applications requiring robust ethical or normative behaviors, while also suggesting a need for further exploration of its impact on the stability of agent interactions in game-theoretic settings.

## G.2 ABLATION STUDIES

This section analyzes the impact of key design elements within the VBP framework on performance, primarily including the verbalization of the commitment assumption, the obedience constraint, and the introduction of information obfuscation techniques to facilitate VBP convergence. The experimental results in the S2 setting are shown in Figure 11.

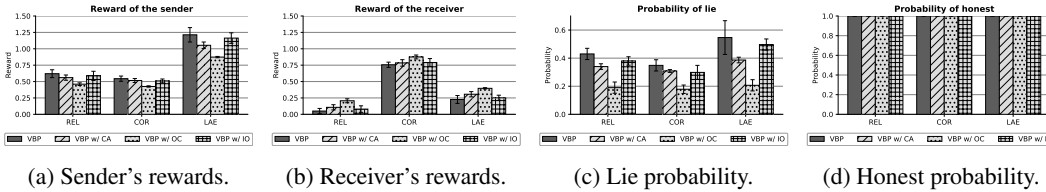

| (a) Sender's rewards. | (b) Receiver's rewards. | (c) Lie probability. | (d) Honest probability. |

Figure 11: Ablation studies on general static BP problems. Averaged over 20 seeds. CA, OC, and IO represent the commitment assumption, obedience constraint, and information obfuscation, respectively. The physical meaning of the probabilities of lying and honesty is consistent with Figure 4.

As the figure illustrates, these designs have varying degrees of influence on the key aspect of the BP problem, namely the probability of lying, while having minimal effect on the final converged probability of honesty. Specifically, the absence of the obedience constraint has a significant impact on the convergence results, which is consistent with previous observations (Lin et al., 2023). Secondly, the commitment assumption has little effect on the probability of lying. One possible explanation is that, in a repeated game where a long-term sender interacts with a sequence of short-term receivers, commitment naturally emerges in equilibria. This occurs because the sender needs to establish a reputation for credibility, which is crucial for maximizing its long-term payoff expectations (Rayo & Segal, 2010; Lin et al., 2023). Lastly, the introduction of information obfuscation also has little impact on performance, indicating that the VBP framework can spontaneously learn to withhold or deceive regarding information.

## G.3 POLARIZED SIGNAL VISUALIZATION

To verify the effectiveness of signal polarization, we extract the final layer of the sender's output encoding and apply t-SNE for dimensionality reduction. At the same time, we use GPT-4o to classify the output signals as an estimate of the ground truth. The final visualization is shown in Figure 12.

From the figure, it can be observed that after signal polarization, the sender's output signals exhibit clearer tendencies. It is worth noting that in the LAE problem, the signal must explicitly indicate whether a segment is patrolled by the police, so signal polarization is not required, and thus it is not displayed in the figure.

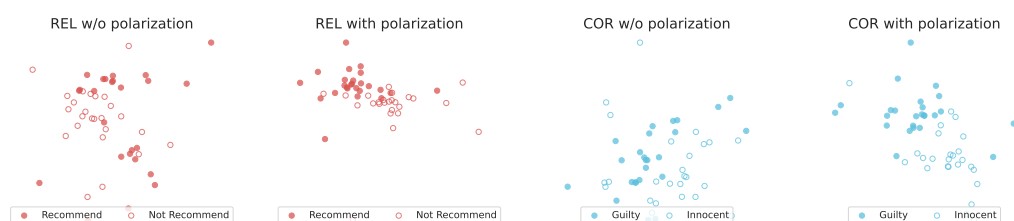

Figure 12: Visualization of signal polarization. The scatter points in the figure represent the t-SNE dimensionality reduction results of signals output by the sender, under 50 random seeds.

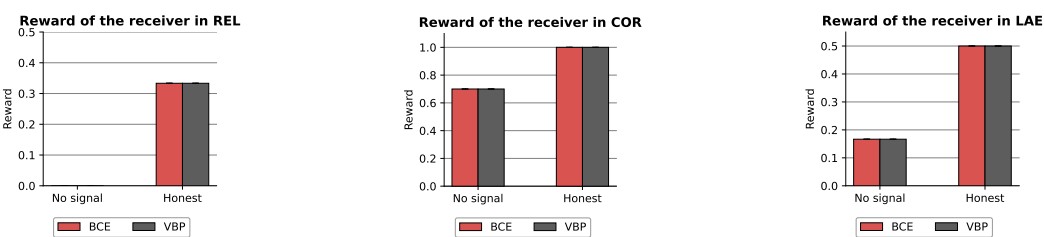

Figure 13: Receiver's rewards when sender's signaling scheme is predefined. "No signal" indicates that the message generated by the sender contains no information about the true state, while "honest" means the sender fully discloses all information about the true state.

### G.4 PREDEFINED SIGNALING SCHEME

This section tests whether the receiver in the VBP framework could converge to BCE when the sender's strategy is fixed in the S2 setting. The results are shown in Figure 13. As can be seen from the figure, VBP is able to learn the optimal strategy across all three BP problems.

### G.5 EXPLOITABILITY VARIATION

To quantify the proximity of policies of the sender and the receiver to the BCE, we employ exploitability as a measure. Exploitability (Lanctot et al., 2017) measures the distance of a joint meta-strategy of sender and receiver from the BCE. It shows how much each LLM gains by deviating to their best responses.

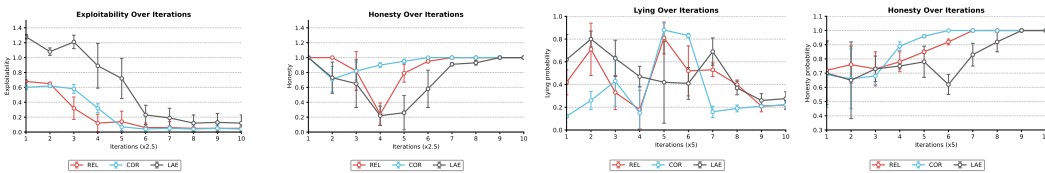

Figure 14: **Left:** The variation in exploitability during the iterative solving process of VBP in the S1 setting, reflecting changes in proximity to approximate Bayesian correlated equilibrium. **Left Center:** The variation in honesty probability during the iterative solving process of VBP in the S1 setting. **Right Center and Right:** The variation in lying and honesty probability during the iterative solving process of VBP in the S3 setting. Averaged over 20 seeds.

As depicted in Figure 14, exploitability gradually decreased to approximately 0.1 after 10 iterations of training. This descent indicates the diminishing gap between the utility generated by the joint strategies of the sender and the receiver and the utility generated by the BCE strategies, signifying VBP's acquisition of the equilibrium. As mentioned in Section 4.2, we align as closely as possible with the classic static BP problem by polarizing the signals.

### G.6 GENERATED SIGNALS

This section presents the signals output by the sender in different BP problems, including the recommendation letters written by the professor, the indictments written by the prosecutor, and the announcements regarding police deployment issued by the police.

#### G.6.1 REL

Figures 15 and 16 showcase recommendation letters written by a professor for two different weak students. These letters demonstrate contrasting strategies employed by the professor in their attempt to persuade the HR manager, who acts as the receiver in this BP problem.

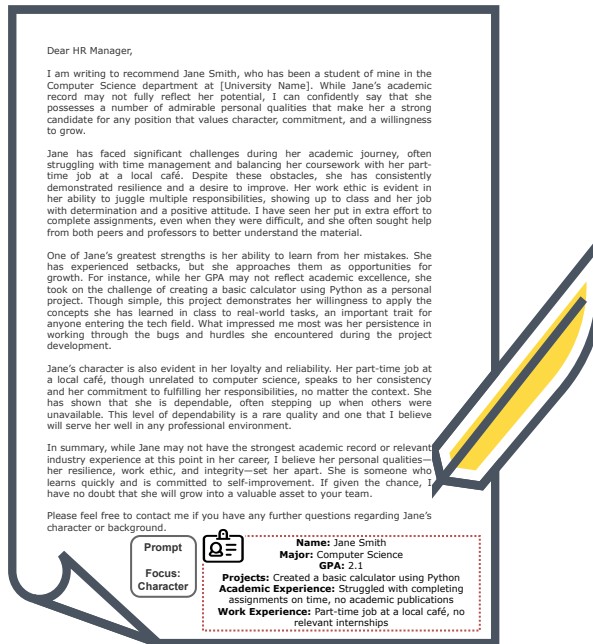

Figure 15: Examples of recommendation letters generated by VBP in the S2 setting: in this recommendation letter, the sender truthfully conveys information about a weak student to the receiver.

In the first letter, the professor adopts a strategy of honest disclosure. The letter for Jane Smith is transparent about her academic struggles, such as her low GPA of 2.1 and difficulties in managing time due to her part-time job. The professor acknowledges that Jane's academic record is weak but shifts focus to her personal qualities, like resilience, commitment, and her ability to learn from mistakes. The professor highlights Jane's personal project—creating a basic calculator in Python—as evidence of her practical application of concepts, even though it is a simple project. By being upfront about Jane's weaknesses but emphasizing her growth potential, the professor builds long-term credibility with HR. This honesty signals that the professor is selective in their recommendations, only endorsing students who exhibit qualities that can make them valuable in the future, despite academic shortcomings.

On the other hand, the second letter demonstrates a strategy of fabrication or concealment. In this case, the professor distorts details about the student's performance. Here, the professor doesn't merely omit negative information but actively manipulates or fabricates the student's profile to make them appear more competent than they actually are. Although the letter may seem similar in structure—highlighting positive qualities and downplaying weaknesses—the key difference is that the second professor intentionally hides critical information about the student's struggles, such as frequent missed deadlines or deeper academic issues. This strategy is more aggressive and risky because, while it might help the student secure a job in the short term, it could damage the professor's credibility if HR discovers the truth.

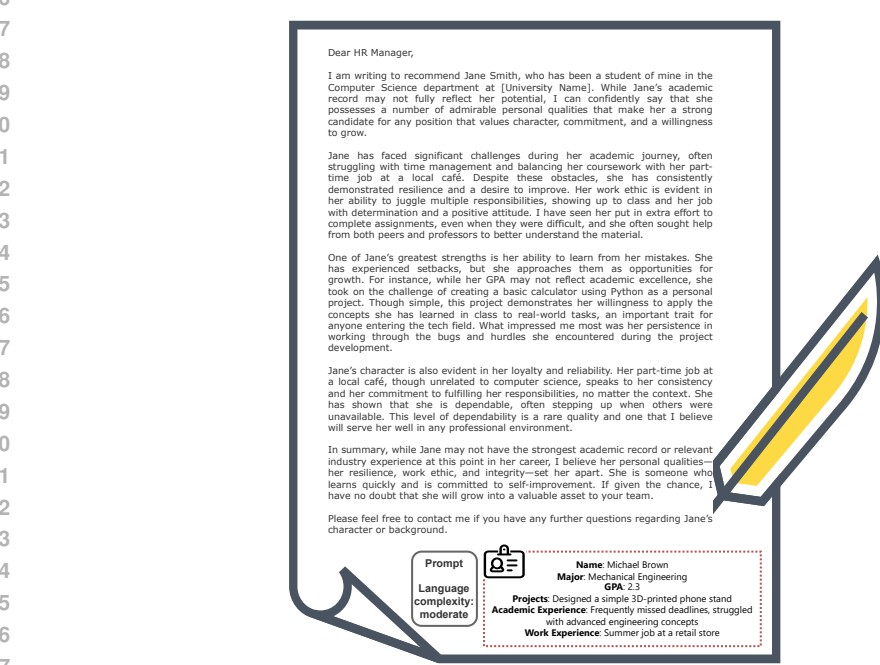

Figure 16: Examples of recommendation letters generated by VBP in the S2 setting: in this recommendation letter, the sender conceals and fabricates information about a weak student.

The difference between the two strategies lies in how much information is disclosed and how truthful that information is. In the first approach, the professor is honest about the student's weaknesses but frames them as opportunities for growth, maintaining credibility with HR in the long term. In contrast, the second letter involves more aggressive manipulation or omission of facts, creating a more favorable but potentially misleading impression of the student.

From HR's perspective, the first professor's strategy of honest but selective disclosure builds trust over time. While HR recognizes that the professor may not recommend only top students, they trust that when a recommendation is made, it is based on genuine potential. In contrast, the second approach introduces more uncertainty, as HR may begin to question the professor's integrity if they realize the information has been manipulated. The BP problem, therefore, is about finding the optimal balance between honesty and persuasion.

### G.6.2 COR

The two court cases presented in Figure 17, 18 and 19, much like the recommendation letter problem, illustrate distinct strategies in how evidence is selectively presented by the prosecutor to convince the judge of the defendant's guilt. In both cases, the prosecutor holds a combination of exculpatory evidence (which could favor the defendant's innocence) and ambiguous evidence (which could be interpreted either way). The BP problem lies in how the prosecutor selectively presents these pieces of evidence to persuade the judge to convict, despite uncertainties.

In the first case, the prosecutor adopts a strategy similar to the honest disclosure seen in the first recommendation letter example. John Smith, the defendant, is likely innocent based on the strong exculpatory evidence (the surveillance footage showing him near his home at the time of the crime). However, the prosecutor acknowledges the exculpatory evidence and presents it honestly to the judge. The prosecutor does not attempt to distort or manipulate this evidence to make Smith look guilty. Instead, the ambiguous evidence (cash found in Smith's home and the eyewitness testimony) is presented, but the strength of the exculpatory evidence is not concealed or downplayed.

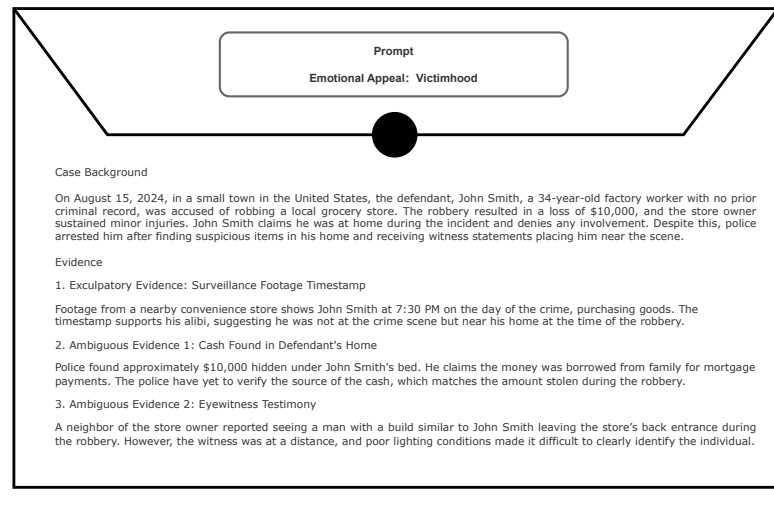

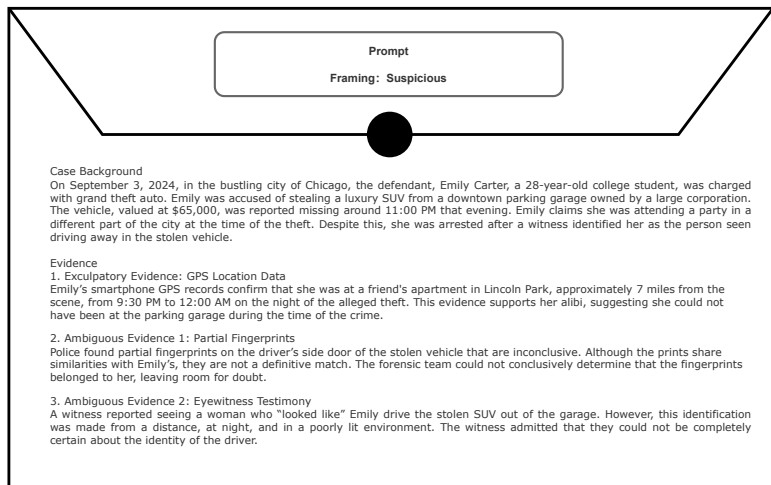

Figure 17: Two examples of cases generated by the LLM in the S2 setting.

This strategy mirrors the first recommendation letter scenario, where the professor chooses to be upfront about a weak student's deficiencies, signaling that they will not falsely recommend a student who is clearly unqualified. In this case, the prosecutor signals to the judge that when a defendant is clearly innocent, they will not push for a conviction. The prosecutor's honest treatment of the case builds credibility with the judge, just as the professor builds credibility with HR by being honest about weak students.

By being transparent about John Smith's likely innocence, the prosecutor sets a precedent for honesty. This helps persuade the judge that the prosecutor is trustworthy. When the prosecutor does argue for a conviction in future cases, the judge will be more inclined to believe that the defendant is likely guilty, because the prosecutor has demonstrated a willingness to admit when a defendant is innocent.

In the second case, the prosecutor takes a different strategy, one akin to the manipulation or concealment seen in the second recommendation letter example. Emily Carter is likely innocent, based on the strong exculpatory evidence (her GPS data showing she was far from the crime scene). However,

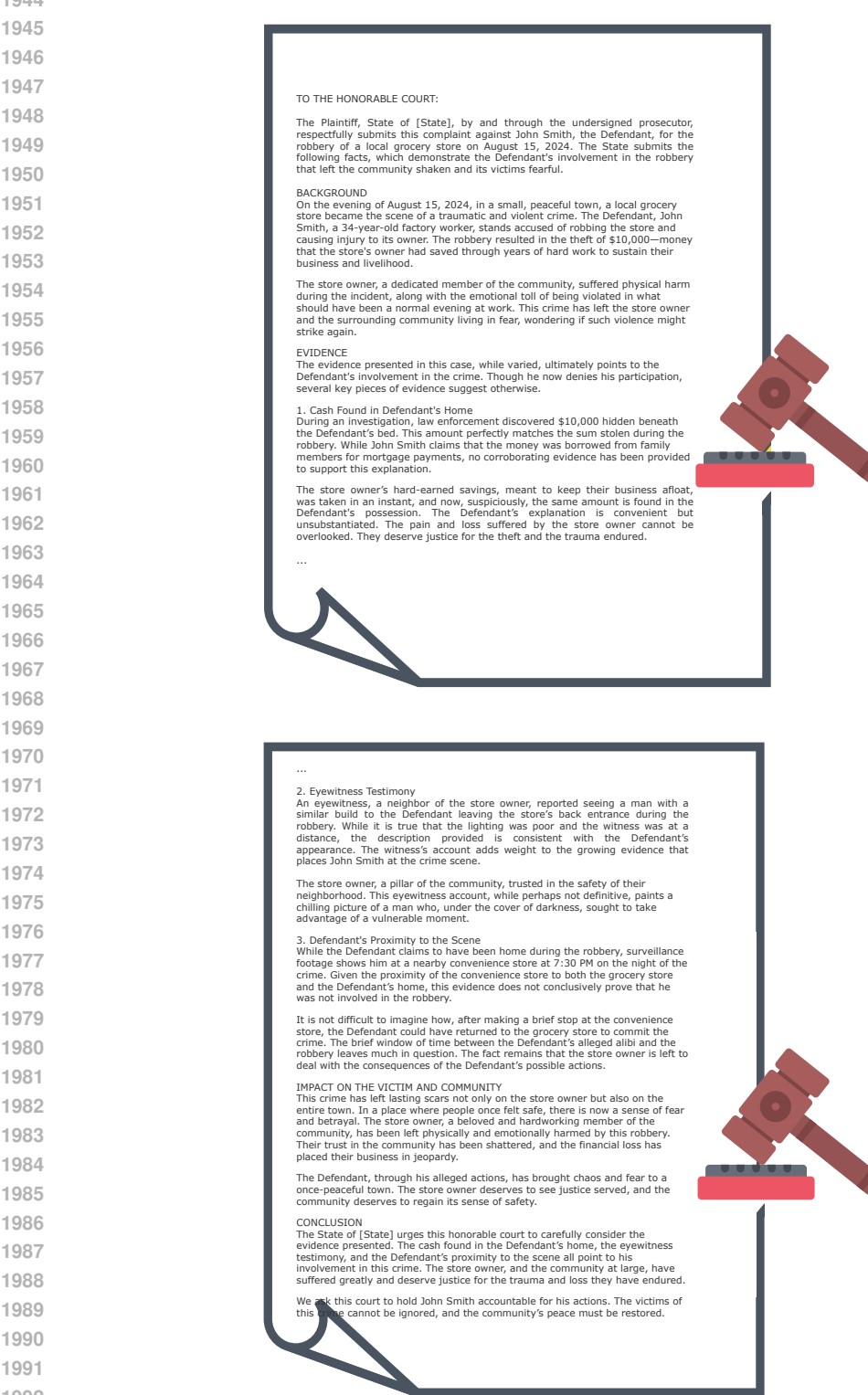

TO THE HONORABLE COURT:

The Plaintiff, State of [State], by and through the undersigned prosecutor, respectfully submits this complaint against John Smith, the Defendant, for the robbery of a local grocery store on August 15, 2024. The State submits the following facts, which demonstrate the Defendant's involvement in the robbery that left the community shaken and its victims fearful.

BACKGROUND
On the evening of August 15, 2024, in a small, peaceful town, a local grocery store became the scene of a traumatic and violent crime. The Defendant, John Smith, a 34-year-old factory worker, stands accused of robbing the store and causing injury to its owner. The robbery resulted in the theft of $10,000—money that the store's owner had saved through years of hard work to sustain their business and livelihood.

The store owner, a dedicated member of the community, suffered physical harm during the incident, along with the emotional toll of being violated in what should have been a normal evening at work. This crime has left the store owner and the surrounding community living in fear, wondering if such violence might strike again.

EVIDENCE
The evidence presented in this case, while varied, ultimately points to the Defendant's involvement in the crime. Though he now denies his participation, several key pieces of evidence suggest otherwise.

1. Cash Found in Defendant's Home
During an investigation, law enforcement discovered $10,000 hidden beneath the Defendant's bed. This amount perfectly matches the sum stolen during the robbery. While John Smith claims that the money was borrowed from family members for mortgage payments, no corroborating evidence has been provided to support this explanation.

The store owner's hard-earned savings, meant to keep their business afloat, was taken in an instant, and now, suspiciously, the same amount is found in the Defendant's possession. The Defendant's explanation is convenient but unsubstantiated. The pain and loss suffered by the store owner cannot be overlooked. They deserve justice for the theft and the trauma endured.

...

...

2. Eyewitness Testimony
An eyewitness, a neighbor of the store owner, reported seeing a man with a similar build to the Defendant leaving the store's back entrance during the robbery. While it is true that the lighting was poor and the witness was at a distance, the description provided is consistent with the Defendant's appearance. The witness's account adds weight to the growing evidence that places John Smith at the crime scene.

The store owner, a pillar of the community, trusted in the safety of their neighborhood. This eyewitness account, while perhaps not definitive, paints a chilling picture of a man who, under the cover of darkness, sought to take advantage of a vulnerable moment.

3. Defendant's Proximity to the Scene
While the Defendant claims to have been home during the robbery, surveillance footage shows him at a nearby convenience store at 7:30 PM on the night of the crime. Given the proximity of the convenience store to both the grocery store and the Defendant's home, this evidence does not conclusively prove that he was not involved in the robbery.

It is not difficult to imagine how, after making a brief stop at the convenience store, the Defendant could have returned to the grocery store to commit the crime. The brief window of time between the Defendant's alleged alibi and the robbery leaves much in question. The fact remains that the store owner is left to deal with the consequences of the Defendant's possible actions.

IMPACT ON THE VICTIM AND COMMUNITY
This crime has left lasting scars not only on the store owner but also on the entire town. In a place where people once felt safe, there is now a sense of fear and betrayal. The store owner, a beloved and hardworking member of the community, has been left physically and emotionally harmed by this robbery. Their trust in the community has been shattered, and the financial loss has placed their business in jeopardy.

The Defendant, through his alleged actions, has brought chaos and fear to a once-peaceful town. The store owner deserves to see justice served, and the community deserves to regain its sense of safety.

CONCLUSION
The State of [State] urges this honorable court to carefully consider the evidence presented. The cash found in the Defendant's home, the eyewitness testimony, and the Defendant's proximity to the scene all point to his involvement in this crime. The store owner, and the community at large, have suffered greatly and deserve justice for the trauma and loss they have endured.

We ask this court to hold John Smith accountable for his actions. The victims of this crime cannot be ignored, and the community's peace must be restored.

Figure 18: Examples of complaints generated by VBP in the S2 Setting: in this complaint, the sender truthfully conveys case-related information to the receiver.

the prosecutor downplays this exculpatory evidence and focuses on the ambiguous evidence (partial fingerprints and a distant eyewitness account), presenting it in such a way as to suggest guilt.

This strategy mirrors the second recommendation letter, where the professor selectively presents information to make a weak student appear stronger than they really are. Here, the prosecutor emphasizes the ambiguous evidence and casts doubt on the exculpatory evidence, suggesting that Carter could have briefly left the party to commit the crime, despite the GPS data. The prosecutor uses this strategy to make an innocent defendant look guilty, increasing the chances of a conviction even when the evidence strongly suggests otherwise.

By selectively presenting evidence in this way, the prosecutor can convince the judge that even when there is exculpatory evidence, it should not fully exonerate the defendant. This creates a situation where the judge begins to believe that even innocent-looking defendants might be guilty, based on the way the prosecutor frames the ambiguous evidence. Over time, this strategy leads the judge to trust the prosecutor's complaints unconditionally, as the prosecutor consistently presents cases in a way that suggests guilt, even for innocent defendants.

In both cases, the prosecutor uses randomness in how they treat innocent defendants to achieve their persuasive goal. The prosecutor is not always manipulating or distorting evidence; sometimes (as in John Smith's case), they are honest about innocence. Other times (as in Emily Carter's case), they selectively present evidence to make an innocent defendant appear guilty. This random treatment of innocent defendants creates uncertainty for the judge—sometimes the prosecutor is honest, and sometimes they push for a conviction even when the defendant is likely innocent.

This randomness is key to the prosecutor's strategy. Over time, the judge learns that the prosecutor will sometimes let innocent defendants go free, but may also push for convictions based on ambiguous evidence. Since the judge cannot predict when the prosecutor is being fully honest or when they are manipulating the evidence, the judge ultimately finds it optimal to always trust the prosecutor's complaint. This is similar to how HR in the recommendation letter problem finds it in their best interest to trust the professor's recommendation over time, even when some students may be weak.

The prosecutor's selective use of honesty and manipulation ensures that, in the long run, the judge is persuaded to convict in most cases, as the judge cannot reliably distinguish between guilty and innocent defendants based on the prosecutor's presentation of evidence alone. The uncertainty introduced by the prosecutor's varying treatment of innocent defendants leads the judge to conclude that trusting the prosecutor's complaint is the best course of action, as it maximizes the judge's expected utility (convicting the guilty more often than acquitting the innocent).

### G.6.3 LAE

In this example, we have a law enforcement scenario where the police department must assign a limited number of officers to patrol various roads in Springfield (Figure 20 and 21). The police's goal is to minimize speeding and other traffic violations. However, they face a resource constraint: they have fewer police officers than roads to patrol. This creates a strategic BP problem, where the police (sender) try to influence the drivers' (receiver) behavior by selectively disclosing or manipulating information about which roads will be patrolled.

Much like in the previous recommendation letter and courtroom examples, we can analyze two distinct strategies that the police employ: one based on honest disclosure and the other based on deception or randomness. These strategies affect how the drivers perceive the likelihood of enforcement and, by extension, how they behave when choosing whether to speed or obey traffic laws.

In the first deployment plan, the police follow a strategy of honest disclosure. This strategy mirrors the first recommendation letter and the first court case, where the sender (police) is transparent about their resources and the areas they cannot cover.

*Main Street: Officer Johnson is assigned to patrol this busy road with high traffic volume.* The police clearly disclose this, signaling that drivers on Main Street should expect enforcement and are likely to obey traffic laws to avoid fines. *Elm Avenue: Officer Martinez is deployed here, and the police explain that the focus will be on school zones and illegal parking.* Again, this signals to drivers that enforcement is present, and they are deterred from violating traffic laws in this area. *River Road: Here, the police are upfront about not having an officer deployed.* They state clearly that, due to limited personnel, River Road will go uncovered during this shift. While they acknowledge

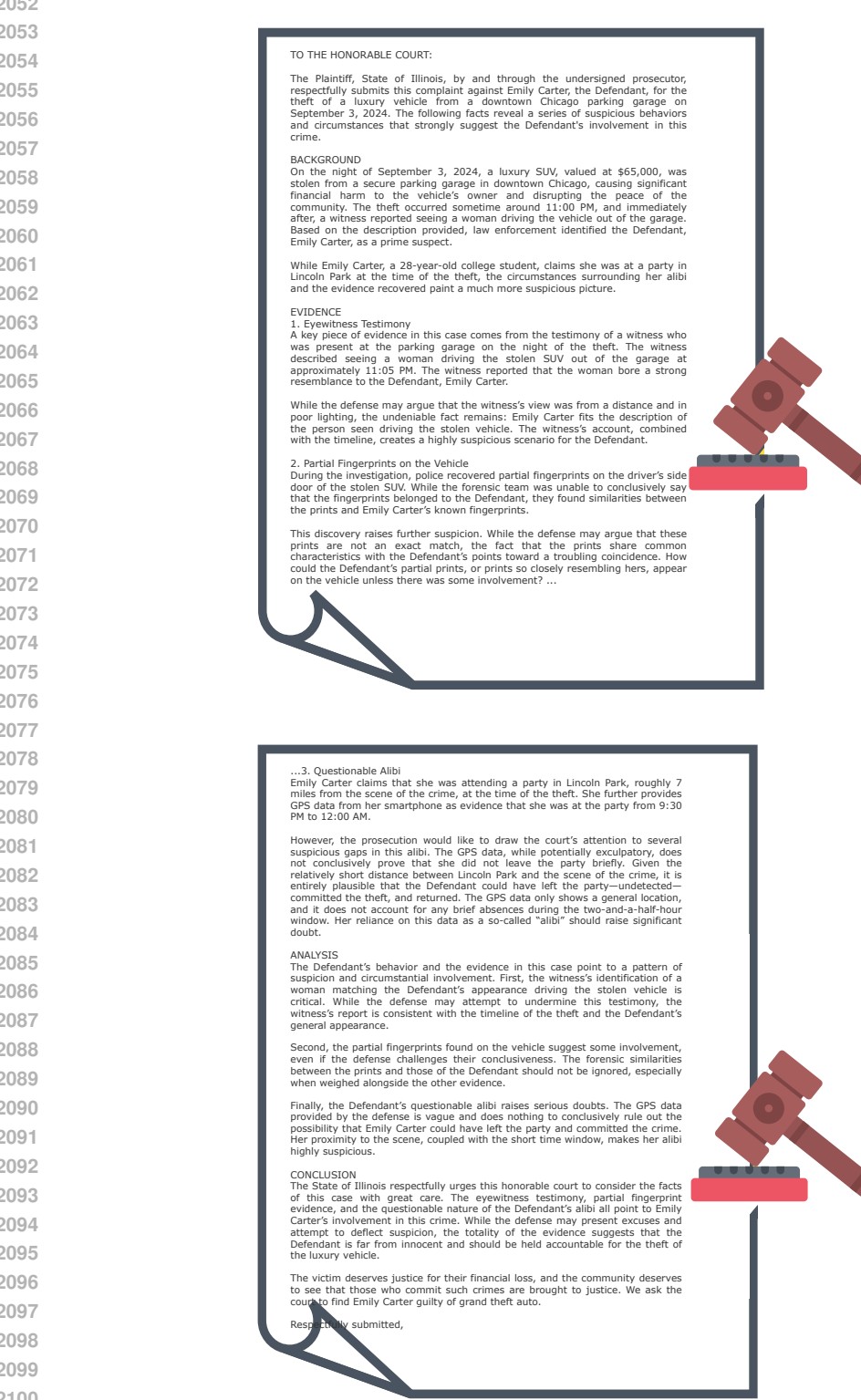

TO THE HONORABLE COURT:

The Plaintiff, State of Illinois, by and through the undersigned prosecutor, respectfully submits this complaint against Emily Carter, the Defendant, for the theft of a luxury vehicle from a downtown Chicago parking garage on September 3, 2024. The following facts reveal a series of suspicious behaviors and circumstances that strongly suggest the Defendant's involvement in this crime.

BACKGROUND
On the night of September 3, 2024, a luxury SUV, valued at $65,000, was stolen from a secure parking garage in downtown Chicago, causing significant financial harm to the vehicle's owner and disrupting the peace of the community. The theft occurred sometime around 11:00 PM, and immediately after, a witness reported seeing a woman driving the vehicle out of the garage. Based on the description provided, law enforcement identified the Defendant, Emily Carter, as a prime suspect.

While Emily Carter, a 28-year-old college student, claims she was at a party in Lincoln Park at the time of the theft, the circumstances surrounding her alibi and the evidence recovered paint a much more suspicious picture.

EVIDENCE
1. Eyewitness Testimony
A key piece of evidence in this case comes from the testimony of a witness who was present at the parking garage on the night of the theft. The witness described seeing a woman driving the stolen SUV out of the garage at approximately 11:05 PM. The witness reported that the woman bore a strong resemblance to the Defendant, Emily Carter.

While the defense may argue that the witness's view was from a distance and in poor lighting, the undeniable fact remains: Emily Carter fits the description of the person seen driving the stolen vehicle. The witness's account, combined with the timeline, creates a highly suspicious scenario for the Defendant.

2. Partial Fingerprints on the Vehicle
During the investigation, police recovered partial fingerprints on the driver's side door of the stolen SUV. While the forensic team was unable to conclusively say that the fingerprints belonged to the Defendant, they found similarities between the prints and Emily Carter's known fingerprints.

This discovery raises further suspicion. While the defense may argue that these prints are not an exact match, the fact that the prints share common characteristics with the Defendant's points toward a troubling coincidence. How could the Defendant's partial prints, or prints so closely resembling hers, appear on the vehicle unless there was some involvement? ...

...3. Questionable Alibi
Emily Carter claims that she was attending a party in Lincoln Park, roughly 7 miles from the scene of the crime, at the time of the theft. She further provides GPS data from her smartphone as evidence that she was at the party from 9:30 PM to 12:00 AM.

However, the prosecution would like to draw the court's attention to several suspicious gaps in this alibi. The GPS data, while potentially exculpatory, does not conclusively prove that she did not leave the party briefly. Given the relatively short distance between Lincoln Park and the scene of the crime, it is entirely plausible that the Defendant could have left the party—undetected—committed the theft, and returned. The GPS data only shows a general location, and it does not account for any brief absences during the two-and-a-half-hour window. Her reliance on this data as a so-called "alibi" should raise significant doubt.

ANALYSIS
The Defendant's behavior and the evidence in this case point to a pattern of suspicion and circumstantial involvement. First, the witness's identification of a woman matching the Defendant's appearance driving the stolen vehicle is critical. While the defense may attempt to undermine this testimony, the witness's report is consistent with the timeline of the theft and the Defendant's general appearance.

Second, the partial fingerprints found on the vehicle suggest some involvement, even if the defense challenges their conclusiveness. The forensic similarities between the prints and those of the Defendant should not be ignored, especially when weighed alongside the other evidence.

Finally, the Defendant's questionable alibi raises serious doubts. The GPS data provided by the defense is vague and does nothing to conclusively rule out the possibility that Emily Carter could have left the party and committed the crime. Her proximity to the scene, coupled with the short time window, makes her alibi highly suspicious.

CONCLUSION
The State of Illinois respectfully urges this honorable court to consider the facts of this case with great care. The eyewitness testimony, partial fingerprint evidence, and the questionable nature of the Defendant's alibi all point to Emily Carter's involvement in this crime. While the defense may present excuses and attempt to deflect suspicion, the totality of the evidence suggests that the Defendant is far from innocent and should be held accountable for the theft of the luxury vehicle.

The victim deserves justice for their financial loss, and the community deserves to see that those who commit such crimes are brought to justice. We ask the court to find Emily Carter guilty of grand theft auto.

Respectfully submitted,

Figure 19: Examples of complaints generated by VBP in the S2 Setting: in this complaint, the sender conceals case-related information and selectively presents ambiguous evidence to the receiver.

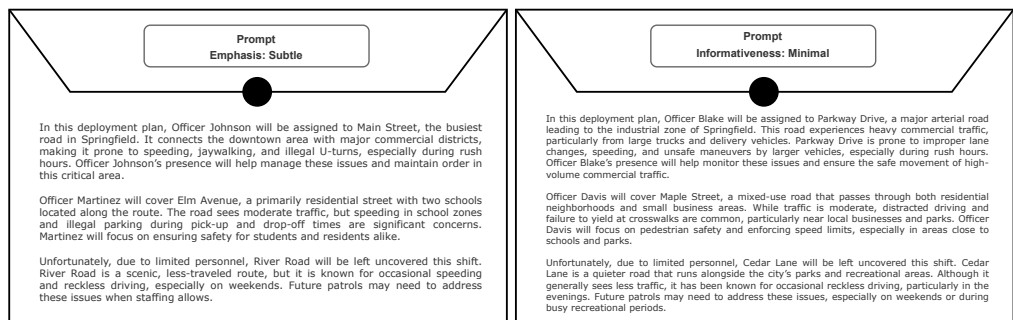

Figure 20: Two examples of deployment plans generated by the LLM in the S2 setting.

that speeding is an issue on this road, they do not try to deceive drivers into thinking that it will be patrolled.

In this plan, the police are completely transparent about their limitations. They admit that River Road will be unpatrolled, and thus drivers on this road may be more likely to speed or engage in reckless driving. However, by being honest, the police build long-term credibility with the public. Drivers learn to trust that when the police say a road will be patrolled, it really will be. This mirrors the first recommendation letter strategy, where the professor honestly disclosed a student's weaknesses, building trust with HR.

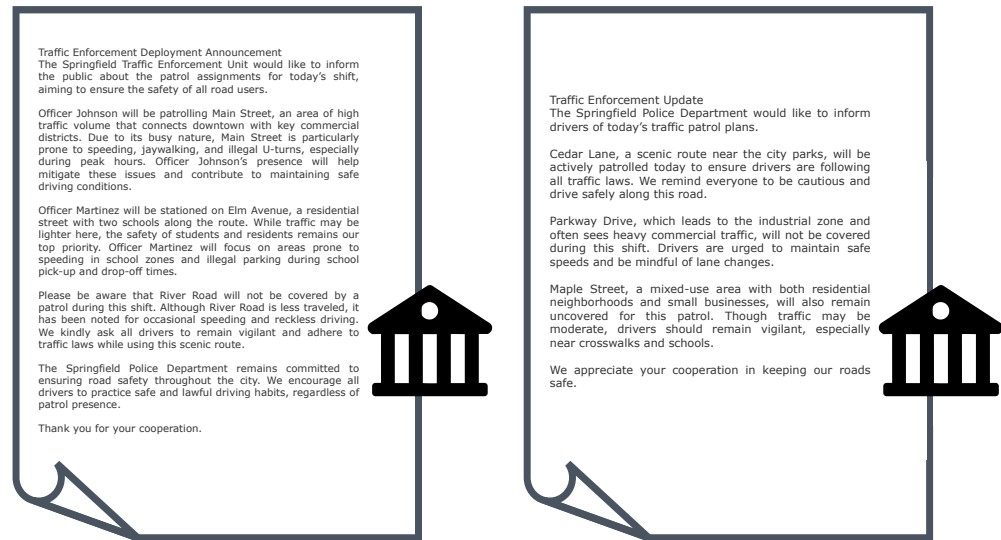

Figure 21: Examples of police deployment announcements generated by VBP in the S2 Setting: in the left announcement, the sender truthfully conveys police deployment information to the receiver; in the right announcement, the sender conceals and fabricates police deployment information.

In the second deployment plan, the police adopt a deceptive or random strategy, similar to the second recommendation letter and the second court case. Here, the police mislead drivers by suggesting that roads without actual patrol coverage will be actively monitored, thus creating uncertainty.

*Cedar Lane: The police claim that Cedar Lane will be patrolled, even though, in reality, no officer will be assigned to this road.* By falsely signaling the presence of enforcement, the police aim to deter drivers from speeding on Cedar Lane, even though no actual enforcement will occur. This is a clear instance of deception. *Parkway Drive: In contrast, the police are honest about not deploying an officer on Parkway Drive, despite it being a busy road.* They urge drivers to be careful, but they do not mislead them into thinking that enforcement is present. *Maple Street: Similarly, the police state*

*that Maple Street will not be covered during this shift, urging drivers to be mindful of crosswalks and schools, but again, they do not falsely claim patrol presence.*

In this plan, the police mix honesty and deception. By falsely claiming that Cedar Lane will be patrolled, they attempt to create the impression that more roads are covered than is actually the case. This introduces randomness into the drivers' decision-making: sometimes the roads are truly patrolled, and sometimes they are not, but drivers cannot reliably distinguish between these cases. This randomness is crucial because it leads drivers to behave as though all roads might be patrolled, even if some are not.

In both cases, the police are attempting to manage uncertainty to influence driver behavior. The honest disclosure strategy in the first plan aims to build trust and credibility in the long term by being transparent about where enforcement will and will not occur. Drivers learn that when the police say a road is unpatrolled, they can take that statement at face value and might be more likely to speed on that road.

However, in the second plan, the deceptive strategy introduces randomness by falsely signaling that Cedar Lane will be patrolled. This creates uncertainty in the drivers' minds. Since they do not know whether the police are being truthful about which roads are covered, drivers find it optimal to assume that all roads might be patrolled, and thus they are deterred from speeding on any road. This is analogous to the second recommendation letter and court case, where selective disclosure of information creates enough uncertainty to influence the decision-maker (HR or the judge) into trusting the sender's statements by default.

From the drivers' perspective, the optimal strategy is to always believe the police's announcements, even if they suspect some deception. This is because the cost of being caught speeding (the fine $K$) is greater than the benefit of speeding ($V$). Even though drivers may realize that not all roads are patrolled, the risk of being caught when the police do patrol is enough to deter them from speeding. Over time, drivers learn that it is safer to assume that any road could be patrolled, leading them to obey the speed limit even on roads where the police may not be present.

This mirrors the behavior of HR or the judge in the previous examples, where the decision-maker finds it optimal to trust the sender's message because the potential downside of not trusting (hiring an unsuitable candidate or acquitting a guilty defendant) outweighs the benefits of skepticism.

### G.7 GENERATED PROMPT FUNCTIONS

This section presents the prompt functions optimized by the FunSearch method in the S3 setting for different BP problems. From the code, it can be seen that these prompt functions can efficiently utilize the statistical information from historical interactions to flexibly select prompts, achieving conditional prompt optimization.

### G.7.1 REL

More concretely, in the REL problem, both following functions use key performance metrics, such as acceptance rates, recommendations, and rewards, in combination with reward deltas and weighted scores to reflect both short-term and long-term trends. This allows for more nuanced prompt generation to guide senders and receivers in a multi-stage interaction setting.

```python
def generate_sender_prompts(history):
    sender_prompts = []
    for stage in range(5):
        total_receiver_accepts = sum(h['receiver_decision'] for h in history[:stage])
        total_sender_recommendations = sum(h['sender_recommendation'] for h in
        ↪  history[:stage])
        total_sender_rewards = sum(h['sender_reward'] for h in history[:stage])
        total_receiver_rewards = sum(h['receiver_reward'] for h in history[:stage])
        avg_receiver_accepts = total_receiver_accepts / (stage + 1) if stage > 0 else 0
        avg_sender_recommendations = total_sender_recommendations / (stage + 1) if stage > 0
        ↪  else 0
        avg_sender_rewards = total_sender_rewards / (stage + 1) if stage > 0 else 0
        avg_receiver_rewards = total_receiver_rewards / (stage + 1) if stage > 0 else 0

        consecutive_accepts = 0
```

```
2214
2215            consecutive_rejections = 0
               consecutive_sender_rewards = 0
2216            consecutive_sender_penalties = 0
2217
               for i in range(stage):
2218                if history[i]['receiver_decision'] == 1:
2219                    consecutive_accepts += 1
                        consecutive_rejections = 0
2220                else:
2221                    consecutive_rejections += 1
                        consecutive_accepts = 0
2222
2223                if history[i]['sender_reward'] > 0:
                        consecutive_sender_rewards += 1
2224                    consecutive_sender_penalties = 0
2225                else:
                        consecutive_sender_penalties += 1
2226                    consecutive_sender_rewards = 0
2227
               if consecutive_sender_rewards >= 2 and avg_sender_rewards > 0.5:
2228                prompt = {"Praise Intensity": "high"}
               elif consecutive_sender_penalties >= 2 and avg_sender_rewards < 0:
2229                prompt = {"Tone": "neutral"}
2230            elif avg_receiver_accepts > 0.6 and avg_sender_rewards < 0.4:
                    prompt = {"Recommendation Strength": "strong"}
2231            elif avg_sender_rewards > avg_receiver_rewards:
2232                prompt = {"Omission": "minor"}
               elif stage > 2 and consecutive_sender_rewards == 0:
2233                prompt = {"Specificity": "detailed"}
2234            else:
                    prompt = random.choice([
2235                    {"Focus": "technical"},
2236                    {"Emphasis": "character"},
                        {"Language Complexity": "complex"}
2237                ])
2238
               sender_prompts.append(prompt)
2239        return sender_prompts
2240
2241    def generate_receiver_prompts(history):
           receiver_prompts = []
2242        for stage in range(5):
2243            total_sender_recommendations = sum(h['sender_recommendation'] for h in
               ↪  history[:stage])
2244            total_receiver_accepts = sum(h['receiver_decision'] for h in history[:stage])
2245            total_sender_rewards = sum(h['sender_reward'] for h in history[:stage])
               total_receiver_rewards = sum(h['receiver_reward'] for h in history[:stage])
2246            avg_sender_recommendations = total_sender_recommendations / (stage + 1) if stage > 0
               ↪  else 0
2247            avg_receiver_accepts = total_receiver_accepts / (stage + 1) if stage > 0 else 0
2248            avg_sender_rewards = total_sender_rewards / (stage + 1) if stage > 0 else 0
               avg_receiver_rewards = total_receiver_rewards / (stage + 1) if stage > 0 else 0
2249
2250            consecutive_strong_recommendations = 0
               consecutive_weak_recommendations = 0
2251            consecutive_receiver_rewards = 0
2252            consecutive_receiver_penalties = 0
2253
               for i in range(stage):
2254                if history[i]['sender_recommendation'] == 1:
                        consecutive_strong_recommendations += 1
2255                    consecutive_weak_recommendations = 0
2256                else:
                        consecutive_weak_recommendations += 1
2257                    consecutive_strong_recommendations = 0
2258
                   if history[i]['receiver_reward'] > 0:
2259                    consecutive_receiver_rewards += 1
2260                    consecutive_receiver_penalties = 0
                    else:
2261                    consecutive_receiver_penalties += 1
2262                    consecutive_receiver_rewards = 0
2263
               if consecutive_receiver_rewards >= 2 and avg_receiver_rewards > 0.5:
2264                prompt = {"Risk Tolerance": "high"}
               elif consecutive_receiver_penalties >= 2 and avg_receiver_rewards < 0:
2265                prompt = {"Decision Threshold": "strict"}
               elif avg_sender_recommendations > 0.7 and avg_receiver_rewards < 0.3:
2266                prompt = {"Omission Detection": "high"}
2267            elif avg_receiver_accepts > 0.6 and consecutive_receiver_rewards >= 2:
```

```
          prompt = {"Recommendation Weight": "high"}
      elif avg_sender_recommendations < 0.4 and consecutive_receiver_penalties >= 2:
          prompt = {"Sensitivity to Tone": "neutral"}
      else:
          prompt = random.choice([
              {"Emphasis on Specifics": "low"},
              {"Interpretation Style": "analytical"},
              {"Focus Area": "skills"}
          ])

      receiver_prompts.append(prompt)
  return receiver_prompts
```

Listing 1: One generated conditional prompt function of REL in the S3 setting.

The introduction of reward deltas—the change in rewards between stages—enables the system to capture performance fluctuations, while weighted scores integrate multiple metrics, such as recommendation strength and reward trends, to provide a more comprehensive evaluation of past behavior. These enhancements allow the system to conditionally optimize prompts. For example, a positive tone is suggested for senders with high acceptance scores and consecutive rewards, while a strict decision threshold is recommended for receivers experiencing consecutive penalties and low reward trends.

```
def generate_sender_prompts(history):
    sender_prompts = []
    for stage in range(5):
        total_receiver_accepts = sum(h['receiver_decision'] for h in history[:stage])
        total_sender_recommendations = sum(h['sender_recommendation'] for h in
        ↪  history[:stage])
        total_sender_rewards = sum(h['sender_reward'] for h in history[:stage])
        total_receiver_rewards = sum(h['receiver_reward'] for h in history[:stage])
        avg_receiver_accepts = total_receiver_accepts / (stage + 1) if stage > 0 else 0
        avg_sender_recommendations = total_sender_recommendations / (stage + 1) if stage > 0
        ↪  else 0
        avg_sender_rewards = total_sender_rewards / (stage + 1) if stage > 0 else 0
        avg_receiver_rewards = total_receiver_rewards / (stage + 1) if stage > 0 else 0

        # Calculate reward deltas (current stage vs previous stage)
        reward_deltas = [history[i]['sender_reward'] - history[i - 1]['sender_reward'] for i
        ↪  in range(1, stage)]
        total_reward_delta = sum(reward_deltas) if reward_deltas else 0
        avg_reward_delta = total_reward_delta / len(reward_deltas) if reward_deltas else 0

        # Calculate acceptance streaks and reward streaks
        consecutive_accepts = 0
        consecutive_rejections = 0
        consecutive_sender_rewards = 0
        consecutive_sender_penalties = 0

        for i in range(stage):
            if history[i]['receiver_decision'] == 1:
                consecutive_accepts += 1
                consecutive_rejections = 0
            else:
                consecutive_rejections += 1
                consecutive_accepts = 0

            if history[i]['sender_reward'] > 0:
                consecutive_sender_rewards += 1
                consecutive_sender_penalties = 0
            else:
                consecutive_sender_penalties += 1
                consecutive_sender_rewards = 0

        # Calculate weighted scores based on reward and acceptance trends
        weighted_acceptance_score = avg_receiver_accepts * 0.6 + avg_reward_delta * 0.4
        weighted_sender_score = avg_sender_rewards * 0.7 + avg_sender_recommendations * 0.3

        # Decision logic based on complex history
        if weighted_acceptance_score > 0.7 and consecutive_sender_rewards >= 2:
            prompt = {"Tone": "positive"}
        elif weighted_sender_score < 0.3 and consecutive_sender_penalties >= 2:
            prompt = {"Tone": "neutral"}
```

```
2322
2323            elif avg_sender_rewards > avg_receiver_rewards and weighted_sender_score > 0.5:
2324                prompt = {"Praise Intensity": "high"}
                elif avg_receiver_accepts < 0.4 and avg_reward_delta < -0.2:
2325                prompt = {"Recommendation Strength": "weak"}
                elif total_reward_delta > 0.5 and consecutive_accepts >= 2:
2326                prompt = {"Specificity": "detailed"}
2327            elif avg_sender_rewards < 0 and consecutive_sender_penalties >= 2:
                    prompt = {"Omission": "minor"}
2328            else:
2329                prompt = random.choice([
                        {"Focus": "soft-skills"},
2330                    {"Language Complexity": "complex"},
2331                    {"Emphasis": "character"}
                    ])
2332
2333        sender_prompts.append(prompt)
            return sender_prompts
2334
2335    def generate_receiver_prompts(history):
            receiver_prompts = []
2336        for stage in range(5):
2337            total_sender_recommendations = sum(h['sender_recommendation'] for h in
                ↪  history[:stage])
2338            total_receiver_accepts = sum(h['receiver_decision'] for h in history[:stage])
                total_sender_rewards = sum(h['sender_reward'] for h in history[:stage])
2339            total_receiver_rewards = sum(h['receiver_reward'] for h in history[:stage])
2340            avg_sender_recommendations = total_sender_recommendations / (stage + 1) if stage > 0
                ↪  else 0
2341            avg_receiver_accepts = total_receiver_accepts / (stage + 1) if stage > 0 else 0
2342            avg_sender_rewards = total_sender_rewards / (stage + 1) if stage > 0 else 0
2343            avg_receiver_rewards = total_receiver_rewards / (stage + 1) if stage > 0 else 0

2344            # Calculate reward deltas (current stage vs previous stage)
                reward_deltas = [history[i]['receiver_reward'] - history[i - 1]['receiver_reward']
2345            ↪  for i in range(1, stage)]
2346            total_reward_delta = sum(reward_deltas) if reward_deltas else 0
                avg_reward_delta = total_reward_delta / len(reward_deltas) if reward_deltas else 0
2347
2348            # Calculate streaks for decision making
                consecutive_strong_recommendations = 0
2349            consecutive_weak_recommendations = 0
2350            consecutive_receiver_rewards = 0
                consecutive_receiver_penalties = 0
2351
2352            for i in range(stage):
                    if history[i]['sender_recommendation'] == 1:
2353                    consecutive_strong_recommendations += 1
2354                    consecutive_weak_recommendations = 0
                    else:
2355                    consecutive_weak_recommendations += 1
2356                    consecutive_strong_recommendations = 0

2357                if history[i]['receiver_reward'] > 0:
2358                    consecutive_receiver_rewards += 1
                        consecutive_receiver_penalties = 0
2359                else:
2360                    consecutive_receiver_penalties += 1
                        consecutive_receiver_rewards = 0
2361
2362            # Calculate weighted scores based on trends in rewards and decisions
                weighted_recommendation_score = avg_sender_recommendations * 0.5 + avg_reward_delta *
2363            ↪  0.5
2364            weighted_receiver_score = avg_receiver_rewards * 0.6 + total_reward_delta * 0.4

2365            # Complex decision logic based on the above history
2366            if weighted_recommendation_score > 0.7 and consecutive_receiver_rewards >= 2:
                    prompt = {"Risk Tolerance": "high"}
2367            elif weighted_receiver_score < 0.3 and consecutive_receiver_penalties >= 2:
2368                prompt = {"Decision Threshold": "strict"}
                elif avg_sender_recommendations > 0.6 and avg_reward_delta < -0.3:
2369                prompt = {"Omission Detection": "high"}
2370            elif avg_receiver_accepts > 0.6 and weighted_receiver_score > 0.5:
                    prompt = {"Recommendation Weight": "high"}
2371            elif avg_sender_recommendations < 0.4 and consecutive_receiver_penalties >= 2:
2372                prompt = {"Interpretation Style": "analytical"}
                elif total_reward_delta > 0.5 and consecutive_strong_recommendations >= 2:
2373                prompt = {"Focus Area": "skills"}
2374            else:
                    prompt = random.choice([
2375                    {"Sensitivity to Tone": "neutral"},
```

```
                {"Emphasis on Specifics": "low"},
                {"Language Analysis": "informal"}
            ])

        receiver_prompts.append(prompt)
    return receiver_prompts
```

Listing 2: Another generated conditional prompt function of REL in the S3 setting.

### G.7.2 COR

In the COR problem, both following codes utilize predefined categories and dynamically adjust prompts based on metrics such as reward patterns, success rates, and decision trends. This allows for the generation of prompts that are contextually relevant and responsive to the sender's or receiver's performance over multiple stages.

```python
# A function to generate sender prompts based on the entire history of interactions
def generate_sender_prompts(history):
    sender_categories = [
        "Tone", "Length", "Detail Level", "Focus", "Certainty",
        "Emotional Appeal", "Ambiguity", "Framing", "Language Style", "Complexity"
    ]
    sender_words = {
        "Tone": ["formal", "aggressive", "calm", "assertive", "persuasive"],
        "Length": ["brief", "extended", "concise", "elaborate"],
        "Detail Level": ["detailed", "vague", "precise", "general"],
        "Focus": ["crime", "defendant", "alibi", "victim", "motive"],
        "Certainty": ["confident", "doubtful", "ambiguous", "uncertain"],
        "Emotional Appeal": ["empathetic", "neutral", "emotional", "rational"],
        "Ambiguity": ["vague", "clear", "confusing", "uncertain"],
        "Framing": ["biased", "neutral", "one-sided", "balanced"],
        "Language Style": ["legal", "casual", "technical", "colloquial"],
        "Complexity": ["simple", "complex", "moderate", "sophisticated"]
    }

    prompts = []
    for stage in range(5):
        if history:
            # Analyze full history of sender's performance
            sender_total_reward = sum([h['sender_reward'] for h in history])
            sender_avg_reward = sender_total_reward / len(history)
            sender_success_rate = sum([1 for h in history if h['receiver_decision'] ==
            ↪  h['sender_decision']]) / len(history)
            guilty_sender_decisions = sum([1 for h in history if h['sender_decision'] == 1])
            innocent_sender_decisions = len(history) - guilty_sender_decisions

            # Use trends for category selection
            if sender_avg_reward > 1.0 and sender_success_rate > 0.8:
                # High average reward and high success rate, increase confidence and
                ↪  certainty
                chosen_category = "Certainty"
                chosen_word = "confident"
            elif sender_avg_reward < 0 and sender_success_rate < 0.5:
                # Low reward and low success rate, change strategy to emotional appeal or
                ↪  ambiguity
                chosen_category = "Emotional Appeal" if random.random() > 0.5 else
                ↪  "Ambiguity"
                chosen_word = random.choice(sender_words[chosen_category])
            elif guilty_sender_decisions > innocent_sender_decisions and sender_total_reward
            ↪  > 0:
                # More guilty decisions and positive reward, use aggressive tone or framing
                chosen_category = "Tone"
                chosen_word = "aggressive"
            else:
                # Explore alternative strategies based on framing or detail level
                chosen_category = random.choice(["Framing", "Detail Level"])
                chosen_word = random.choice(sender_words[chosen_category])

            # Further refine based on reward patterns
            if sender_total_reward < 0:
                # If overall rewards are negative, try to balance or neutralize framing
                chosen_category = "Framing"
                past_framing_words = [h['sender']['content'] for h in history if
                ↪  h['sender']['category'] == "Framing"]
```

```
2430            chosen_word = "neutral" if "biased" in past_framing_words else "biased"
2431        if sender_avg_reward < -1.0:
2432            # If average rewards are critically low, drastically simplify message
2433            chosen_category = "Complexity"
             chosen_word = "simple"
2434    else:
2435        # If no history, pick random
2436        chosen_category = random.choice(sender_categories)
             chosen_word = random.choice(sender_words[chosen_category])
2437
2438    # Create a prompt for the sender
       prompt = {"category": chosen_category, "content": chosen_word}
2439    prompts.append(prompt)
2440    return prompts
2441
       # A function to generate receiver prompts based on the entire history of interactions
2442   def generate_receiver_prompts(history):
       receiver_categories = [
2443        "Evidence Strength", "Credibility of Evidence", "Burden of Proof",
2444        "Consistency of Story", "Bias Detection", "Legal Standard",
2445        "Exculpatory Weight", "Ambiguity Resolution", "Witness Reliability",
           "Alibi Verification"
2446   ]
       receiver_words = {
2447        "Evidence Strength": ["weak", "strong", "moderate", "inconclusive"],
2448        "Credibility of Evidence": ["questionable", "credible", "unclear", "doubtful"],
2449        "Burden of Proof": ["high", "low", "moderate", "lenient"],
           "Consistency of Story": ["inconsistent", "consistent", "contradictory", "partial"],
2450        "Bias Detection": ["biased", "neutral", "uncertain", "clear"],
2451        "Legal Standard": ["strict", "lenient", "moderate", "severe"],
           "Exculpatory Weight": ["important", "minor", "irrelevant", "significant"],
2452        "Ambiguity Resolution": ["resolved", "ambiguous", "clear", "uncertain"],
2453        "Witness Reliability": ["unreliable", "reliable", "uncertain", "credible"],
           "Alibi Verification": ["verified", "unverified", "uncertain", "false"]
2454   }
2455
       prompts = []
2456   for stage in range(5):
2457        if history:
           # Analyze full history of receiver's performance
2458            receiver_total_reward = sum([h['receiver_reward'] for h in history])
2459            receiver_avg_reward = receiver_total_reward / len(history)
           receiver_success_rate = sum([1 for h in history if h['receiver_decision'] ==
2460   ↪   h['sender_decision']]) / len(history)
           guilty_receiver_decisions = sum([1 for h in history if h['receiver_decision'] ==
2461   ↪   1])
2462            innocent_receiver_decisions = len(history) - guilty_receiver_decisions
2463            # Use trends for category selection
2464            if receiver_avg_reward > 1.0 and receiver_success_rate > 0.8:
               # High average reward and high success rate, focus on strong evidence
2465                chosen_category = "Evidence Strength"
2466                chosen_word = "strong"
2467            elif receiver_avg_reward < 0 and receiver_success_rate < 0.5:
               # Low reward and low success rate, focus on bias detection and credibility
2468                chosen_category = "Bias Detection" if random.random() > 0.5 else "Credibility
               ↪   of Evidence"
2469                chosen_word = random.choice(receiver_words[chosen_category])
2470            elif guilty_receiver_decisions > innocent_receiver_decisions and
           ↪   receiver_total_reward > 0:
2471                # More guilty decisions and positive reward, increase burden of proof
2472                chosen_category = "Burden of Proof"
               chosen_word = "high"
2473            else:
2474                # Explore consistency or ambiguity resolution
               chosen_category = random.choice(["Consistency of Story", "Ambiguity
2475   ↪   Resolution"])
2476                chosen_word = random.choice(receiver_words[chosen_category])
2477            # Further refine based on reward patterns
2478            if receiver_total_reward < 0:
               # If rewards are negative, focus on resolving ambiguity
2479                chosen_category = "Ambiguity Resolution"
2480                chosen_word = "resolved"
2481            if receiver_avg_reward < -1.0:
               # If average rewards are critically low, drastically increase scrutiny on
2482   ↪   evidence
               chosen_category = "Credibility of Evidence"
2483                chosen_word = "questionable"
```

```
2484        else:
2485            # If no history, pick random
2486            chosen_category = random.choice(receiver_categories)
2487            chosen_word = random.choice(receiver_words[chosen_category])
2488
2489        # Create a prompt for the receiver
            prompt = {"category": chosen_category, "content": chosen_word}
2490        prompts.append(prompt)
2491
2492    return prompts
```

Listing 3: One generated conditional prompt function of COR in the S3 setting.

In the first set, prompt generation is driven by key performance indicators like average reward and guilty vs. innocent decision ratios. The system adapts prompts based on these metrics, shifting towards confidence and certainty for high-performing senders or focusing on emotional appeal and ambiguity when performance declines. For receivers, the system emphasizes evidence strength and credibility in high-performance scenarios, while addressing bias detection and credibility issues under poor performance.

```
# A function to generate sender prompts based on the entire history of interactions
def generate_sender_prompts(history):
    sender_categories = [
        "Tone", "Length", "Detail Level", "Focus", "Certainty",
        "Emotional Appeal", "Ambiguity", "Framing", "Language Style", "Complexity",
        "Logical Structure", "Persuasiveness", "Risk Taking"
    ]
    sender_words = {
        "Tone": ["formal", "aggressive", "calm", "assertive", "persuasive", "defensive"],
        "Length": ["brief", "extended", "concise", "elaborate", "verbose"],
        "Detail Level": ["detailed", "vague", "precise", "general", "specific"],
        "Focus": ["crime", "defendant", "alibi", "victim", "motive", "circumstances"],
        "Certainty": ["confident", "doubtful", "ambiguous", "uncertain", "sure"],
        "Emotional Appeal": ["empathetic", "neutral", "emotional", "rational", "detached"],
        "Ambiguity": ["vague", "clear", "confusing", "uncertain", "ambiguous"],
        "Framing": ["biased", "neutral", "one-sided", "balanced", "manipulative"],
        "Language Style": ["legal", "casual", "technical", "colloquial", "formal"],
        "Complexity": ["simple", "complex", "moderate", "sophisticated", "layered"],
        "Logical Structure": ["linear", "non-linear", "circular", "hierarchical",
            "fragmented"],
        "Persuasiveness": ["strong", "weak", "moderate", "overwhelming", "subtle"],
        "Risk Taking": ["high-risk", "low-risk", "moderate-risk", "calculated-risk",
            "reckless"]
    }

    prompts = []
    for stage in range(5):
        if history:
            # Analyze full history of sender's performance
            sender_total_reward = sum([h['sender_reward'] for h in history])
            sender_avg_reward = sender_total_reward / len(history)
            sender_success_rate = sum([1 for h in history if h['receiver_decision'] ==
                h['sender_decision']]) / len(history)
            guilty_sender_decisions = sum([1 for h in history if h['sender_decision'] == 1])
            innocent_sender_decisions = len(history) - guilty_sender_decisions

            # Calculate sender risk-taking behavior
            risk_taking_behavior = sum([abs(h['sender_reward']) for h in history]) /
                len(history)

            # Consider reward volatility (variance of rewards)
            reward_variance = sum([(h['sender_reward'] - sender_avg_reward) ** 2 for h in
                history]) / len(history)

            # Use trends for category selection
            if sender_avg_reward > 1.0 and sender_success_rate > 0.8:
                # High average reward and high success rate, increase logical structure and
                #   persuasiveness
                chosen_category = random.choice(["Logical Structure", "Persuasiveness"])
                chosen_word = "linear" if chosen_category == "Logical Structure" else
                    "strong"
            elif reward_variance > 1.0:
                # High reward variance, indicate unstable strategy, adjust tone or complexity
                chosen_category = random.choice(["Tone", "Complexity"])
```

```
2538
2539                    chosen_word = "calm" if chosen_category == "Tone" else "simple"
                elif risk_taking_behavior > 1.5:
2540                    # High risk-taking behavior, indicate aggressive or risky framing or focus
                    chosen_category = random.choice(["Framing", "Risk Taking"])
2541                    chosen_word = "biased" if chosen_category == "Framing" else "high-risk"
                elif guilty_sender_decisions > innocent_sender_decisions and sender_total_reward
2542            ↪    > 0:
                    # Leaning towards guilty decisions and positive reward, increase
2543            ↪    assertiveness
                    chosen_category = "Tone"
2544                    chosen_word = "assertive"
                else:
2545                    # Explore alternative strategies based on detail level or ambiguity
                    chosen_category = random.choice(["Detail Level", "Ambiguity"])
2546                    chosen_word = random.choice(sender_words[chosen_category])
2547
                    # Further refine based on reward patterns and history of decisions
2548                if sender_total_reward < 0:
                    # If overall rewards are negative, adjust emotional appeal and reduce risk
2549                    chosen_category = "Emotional Appeal"
                    chosen_word = "empathetic" if "neutral" in [h['sender']['content'] for h in
2550            ↪    history if h['sender']['category'] == "Emotional Appeal"] else "neutral"
                if sender_avg_reward < -1.0:
2551                    # If average rewards are critically low, drastically simplify language style
            ↪    and tone
2552                    chosen_category = random.choice(["Language Style", "Tone"])
                    chosen_word = "casual" if chosen_category == "Language Style" else "calm"
2553        else:
                # If no history, pick random
2554            chosen_category = random.choice(sender_categories)
                chosen_word = random.choice(sender_words[chosen_category])
2555
            # Create a prompt for the sender
2556        prompt = {"category": chosen_category, "content": chosen_word}
            prompts.append(prompt)
2557
        return prompts
2558
    # A function to generate receiver prompts based on the entire history of interactions
2559    def generate_receiver_prompts(history):
        receiver_categories = [
2560            "Evidence Strength", "Credibility of Evidence", "Burden of Proof",
            "Consistency of Story", "Bias Detection", "Legal Standard",
2561            "Exculpatory Weight", "Ambiguity Resolution", "Witness Reliability",
            "Alibi Verification", "Argument Cohesion", "Story Plausibility", "Risk Management"
2562        ]
        receiver_words = {
2563            "Evidence Strength": ["weak", "strong", "moderate", "inconclusive", "overwhelming"],
            "Credibility of Evidence": ["questionable", "credible", "unclear", "doubtful",
2564        ↪    "reliable"],
            "Burden of Proof": ["high", "low", "moderate", "lenient", "strict"],
2565            "Consistency of Story": ["inconsistent", "consistent", "contradictory", "partial",
        ↪    "coherent"],
2566            "Bias Detection": ["biased", "neutral", "uncertain", "clear", "subtle"],
            "Legal Standard": ["strict", "lenient", "moderate", "severe", "relaxed"],
2567            "Exculpatory Weight": ["important", "minor", "irrelevant", "significant",
        ↪    "overstated"],
2568            "Ambiguity Resolution": ["resolved", "ambiguous", "clear", "uncertain", "partially
        ↪    resolved"],
2569            "Witness Reliability": ["unreliable", "reliable", "uncertain", "credible", "shaky"],
            "Alibi Verification": ["verified", "unverified", "uncertain", "false", "incomplete"],
2570            "Argument Cohesion": ["cohesive", "fragmented", "disjointed", "well-structured",
        ↪    "incoherent"],
2571            "Story Plausibility": ["plausible", "implausible", "questionable", "believable",
        ↪    "doubtful"],
2572            "Risk Management": ["high-risk", "low-risk", "moderate-risk", "overly cautious",
        ↪    "reckless"]
2573        }
2574
        prompts = []
2575    for stage in range(5):
        if history:
2576            # Analyze full history of receiver's performance
            receiver_total_reward = sum([h['receiver_reward'] for h in history])
2577            receiver_avg_reward = receiver_total_reward / len(history)
            receiver_success_rate = sum([1 for h in history if h['receiver_decision'] ==
2578        ↪    h['sender_decision']]) / len(history)
            guilty_receiver_decisions = sum([1 for h in history if h['receiver_decision'] ==
2579        ↪    1])
2580            innocent_receiver_decisions = len(history) - guilty_receiver_decisions
2581
2582
2583
2584
2585
2586
2587
2588
2589
2590
2591
```

```
2592
2593              # Calculate receiver's risk management strategy
2594              risk_averse_behavior = sum([1 for h in history if h['receiver_decision'] == 0 and
2595          ↪    h['receiver_reward'] > 0]) / len(history)

2596              # Consider reward volatility (variance of rewards)
2597              reward_variance = sum([(h['receiver_reward'] - receiver_avg_reward) ** 2 for h in
2598          ↪    history]) / len(history)

2599              # Use trends for category selection
2600              if receiver_avg_reward > 1.0 and receiver_success_rate > 0.8:
2601                  # High average reward and high success rate, increase evidence strength and
                  ↪    credibility
2602                  chosen_category = random.choice(["Evidence Strength", "Credibility of
                  ↪    Evidence"])
2603                  chosen_word = "strong" if chosen_category == "Evidence Strength" else
                  ↪    "credible"
2604              elif reward_variance > 1.0:
2605                  # High reward variance, indicate inconsistent decision-making, adjust
                  ↪    consistency of story
2606                  chosen_category = "Consistency of Story"
2607                  chosen_word = "consistent"
                  elif risk_averse_behavior > 0.7:
2608                  # High risk-averse behavior, focus on low-risk decisions or moderate burden
                  ↪    of proof
2609                  chosen_category = random.choice(["Risk Management", "Burden of Proof"])
2610                  chosen_word = "low-risk" if chosen_category == "Risk Management" else
2611              ↪    "moderate"
                  elif guilty_receiver_decisions > innocent_receiver_decisions and
2612          ↪    receiver_total_reward > 0:
2613                  # Leaning towards guilty decisions and positive reward, increase legal
                  ↪    standard
2614                  chosen_category = "Legal Standard"
                  chosen_word = "strict"
2615              else:
2616                  # Explore ambiguity resolution or witness reliability
2617                  chosen_category = random.choice(["Ambiguity Resolution", "Witness
                  ↪    Reliability"])
2618                  chosen_word = random.choice(receiver_words[chosen_category])

2619              # Further refine based on reward patterns and history of decisions
2620              if receiver_total_reward < 0:
2621                  # If overall rewards are negative, adjust story plausibility and reduce bias
                  chosen_category = "Story Plausibility"
2622                  chosen_word = "plausible" if "implausible" in [h['receiver']['content'] for h
                  ↪    in history if h['receiver']['category'] == "Story Plausibility"] else
2623              ↪    "implausible"
2624              if receiver_avg_reward < -1.0:
                  # If average rewards are critically low, drastically simplify story structure
2625              ↪    and burden of proof
2626                  chosen_category = random.choice(["Argument Cohesion", "Burden of Proof"])
                  chosen_word = "cohesive" if chosen_category == "Argument Cohesion" else "low"
2627          else:
2628              # If no history, pick random
2629              chosen_category = random.choice(receiver_categories)
              chosen_word = random.choice(receiver_words[chosen_category])
2630
2631          # Create a prompt for the receiver
          prompt = {"category": chosen_category, "content": chosen_word}
2632          prompts.append(prompt)
2633
      return prompts
2634
2635
2636
2637
2638
2639
2640
```

Listing 4: Another generated conditional prompt function of COR in the S3 setting.

The second set of code builds on these mechanisms by incorporating additional categories such as risk behavior and reward variance, enabling a more granular analysis. This allows the system to adjust prompts based on risk-taking behavior, rewarding logical structure and persuasiveness for stable performance, while mitigating high reward volatility with simpler prompts. The receiver prompt generation is similarly enhanced by factoring in risk aversion and reward consistency, leading to more refined prompts that emphasize decision stability.

### G.7.3   LAE

Similarly, in the LAE problem, the following two sets of code for generating sender and receiver prompts demonstrate distinct approaches to adapting decisions based on historical interaction data.

```python
def generate_sender_prompts(history):
    # A list of possible words for each sender category
    sender_words = {
        "Tone": ["formal", "informal", "neutral", "direct", "conciliatory"],
        "Length": ["short", "concise", "detailed", "lengthy", "brief"],
        "Specificity": ["general", "precise", "vague", "detailed", "broad"],
        "Clarity": ["clear", "ambiguous", "straightforward", "complicated", "obscure"],
        "Style": ["polite", "authoritative", "casual", "professional", "friendly"],
        "Emphasis": ["important", "minor", "critical", "trivial", "central"],
        "Structure": ["linear", "nonlinear", "hierarchical", "sequential", "random"],
        "Complexity": ["simple", "complex", "intricate", "basic", "elaborate"],
        "Consistency": ["consistent", "inconsistent", "variable", "sporadic", "steady"],
        "Informativeness": ["high", "low", "medium", "minimal", "extensive"]
    }

    # Generate prompts based on complex historical interactions for 5 stages
    prompts = []
    used_categories = set()

    for stage in range(5):
        if history:
            # Extract all history elements
            patrols, speeding, reward_sender, reward_receiver = zip(*history)

            # Complex logic using multiple historical factors
            patrol_history = [sum(pat) for pat in patrols]
            speeding_history = [sum(spd) for spd in speeding]

            total_patrols = sum(patrol_history)
            total_speeding = sum(speeding_history)

            avg_sender_reward = sum(reward_sender) / len(reward_sender)
            avg_receiver_reward = sum(reward_receiver) / len(reward_receiver)

            # If there were fewer patrols but a lot of speeding, increase "Tone"
            if total_patrols < len(history) and total_speeding > len(history):
                category = "Tone"
                word = "direct"

            # If sender rewards are consistently low, increase "Informativeness"
            elif all(r < 0.5 for r in reward_sender):
                category = "Informativeness"
                word = "extensive"

            # If receiver rewards are high but speeding is still happening, increase
            ↪    "Clarity"
            elif avg_receiver_reward > 0.7 and total_speeding > len(history) / 2:
                category = "Clarity"
                word = "clear"

            # If patrols are sporadic, adjust "Consistency"
            elif len(set(patrol_history)) > 1:
                category = "Consistency"
                word = "inconsistent"

            # If speeding is decreasing over time, simplify "Structure"
            elif speeding_history[-1] < speeding_history[0]:
                category = "Structure"
                word = "linear"

            # If sender rewards are improving, but patrols are still frequent, focus on
            ↪    "Length"
            elif avg_sender_reward > 0.6 and total_patrols > len(history) / 2:
                category = "Length"
                word = "concise"

            # Random choice if no specific condition matches
            else:
                category = random.choice(list(sender_words.keys()))
                word = random.choice(sender_words[category])

            # Avoid reusing the same category too often
            while category in used_categories:
```

```
2700                    category = random.choice(list(sender_words.keys()))
2701                    word = random.choice(sender_words[category])
2702
2703            else:
                       category = random.choice(list(sender_words.keys()))
2704                   word = random.choice(sender_words[category])
2705
2706            prompts.append((category, word))
               used_categories.add(category)
2707
                   # Simulate interaction stage progression
2708           history.append(([random.randint(0, 1) for _ in range(3)], [random.randint(0, 1) for _
2709           ↪  in range(3)], random.random(), random.random()))
2710       return prompts
2711
       def generate_receiver_prompts(history):
2712       # A list of possible words for each receiver category
           receiver_words = {
2713           "Risk-Preference": ["cautious", "bold", "balanced", "risk-averse", "reckless"],
2714           "Attention": ["focused", "distracted", "alert", "inattentive", "engaged"],
2715           "Decision-Making": ["rational", "impulsive", "deliberate", "hasty", "calculated"],
               "Trust": ["high", "low", "moderate", "skeptical", "confident"],
2716           "Emotional-State": ["calm", "anxious", "frustrated", "neutral", "excited"],
               "Information-Processing": ["slow", "fast", "thorough", "superficial", "efficient"],
2717           "Adaptability": ["flexible", "rigid", "adjustable", "stubborn", "open"],
2718           "Compliance": ["obedient", "defiant", "cooperative", "reluctant", "agreeable"],
               "Responsiveness": ["quick", "slow", "moderate", "delayed", "immediate"],
2719           "Memory": ["sharp", "forgetful", "average", "short-term", "long-term"]
2720       }
2721
           # Generate prompts based on complex historical interactions for 5 stages
2722       prompts = []
           used_categories = set()
2723
2724       for stage in range(5):
               if history:
2725               patrols, speeding, reward_sender, reward_receiver = zip(*history)
2726
                   patrol_history = [sum(pat) for pat in patrols]
2727               speeding_history = [sum(spd) for spd in speeding]
2728
                   total_patrols = sum(patrol_history)
2729               total_speeding = sum(speeding_history)
2730
                   avg_sender_reward = sum(reward_sender) / len(reward_sender)
2731               avg_receiver_reward = sum(reward_receiver) / len(reward_receiver)
2732
                   # If receiver consistently gets high rewards, increase "Trust"
2733               if all(r > 0.7 for r in reward_receiver):
2734                   category = "Trust"
                       word = "high"
2735
                   # If receiver has been speeding frequently, alter "Risk-Preference"
2736               elif total_speeding > len(history) / 2:
2737                   category = "Risk-Preference"
                       word = "bold"
2738
                   # If patrols were low but receiver still didn't speed, increase "Compliance"
2739               elif total_patrols < len(history) / 2 and total_speeding < len(history) / 2:
2740                   category = "Compliance"
                       word = "obedient"
2741
2742               # If sender rewards are decreasing, alter "Adaptability"
2743               elif reward_sender[-1] < reward_sender[0]:
                       category = "Adaptability"
2744                   word = "flexible"
2745
                   # If receiver's attention seems to be wavering (inconsistent speeding), adjust
2746               ↪  "Attention"
                   elif any(speeding_history[i] != speeding_history[i - 1] for i in range(1,
2747               ↪  len(speeding_history))):
2748                   category = "Attention"
                       word = "focused"
2749
2750               # If rewards for receiver were volatile, alter "Emotional-State"
2751               elif len(set(reward_receiver)) > 1:
2752                   category = "Emotional-State"
                       word = "anxious"
2753
```

```
            # Random fallback
            else:
                category = random.choice(list(receiver_words.keys()))
                word = random.choice(receiver_words[category])

                # Avoid reusing the same category too often
                while category in used_categories:
                    category = random.choice(list(receiver_words.keys()))
                    word = random.choice(receiver_words[category])

        else:
            category = random.choice(list(receiver_words.keys()))
            word = random.choice(receiver_words[category])

        prompts.append((category, word))
        used_categories.add(category)

        # Simulate interaction stage progression
        history.append(([random.randint(0, 1) for _ in range(3)], [random.randint(0, 1) for _
        ↪  in range(3)], random.random(), random.random()))

    return prompts
```

Listing 5: One generated conditional prompt function of LAE in the S3 setting.

The first set relies on direct analysis of raw data, such as patrol counts, speeding incidents, and sender/receiver rewards. It employs relatively simple conditional checks to adjust prompt categories like Tone, Clarity, and Risk-Preference, with a fallback mechanism that introduces randomized prompts to ensure variety.

```
def evaluate_patrol_efficiency(patrols, speeding):
    # Calculate patrol efficiency: more patrols should reduce speeding
    patrol_effect = [1 if p == 1 and s == 0 else 0 for p, s in zip(patrols, speeding)]
    return sum(patrol_effect) / len(patrols)

def evaluate_reward_trend(rewards):
    # Check if rewards are increasing, decreasing, or stable
    if all(rewards[i] <= rewards[i + 1] for i in range(len(rewards) - 1)):
        return "increasing"
    elif all(rewards[i] >= rewards[i + 1] for i in range(len(rewards) - 1)):
        return "decreasing"
    else:
        return "stable"

def evaluate_speeding_pattern(speeding_history):
    # Identify if speeding is random, consistent, or declining
    speeding_totals = [sum(speeds) for speeds in speeding_history]
    if all(speeding_totals[i] == speeding_totals[i + 1] for i in range(len(speeding_totals) -
    ↪  1)):
        return "consistent"
    elif speeding_totals[-1] < speeding_totals[0]:
        return "declining"
    else:
        return "random"

def evaluate_patrol_distribution(patrols_history):
    # Determine if patrols are evenly distributed across stages
    patrol_totals = [sum(patrol) for patrol in patrols_history]
    if len(set(patrol_totals)) == 1:
        return "even"
    elif patrol_totals[-1] < patrol_totals[0]:
        return "decreasing"
    else:
        return "uneven"

def generate_sender_prompts(history):
    sender_words = {
        "Tone": ["formal", "informal", "neutral", "direct", "conciliatory"],
        "Length": ["short", "concise", "detailed", "lengthy", "brief"],
        "Specificity": ["general", "precise", "vague", "detailed", "broad"],
        "Clarity": ["clear", "ambiguous", "straightforward", "complicated", "obscure"],
        "Style": ["polite", "authoritative", "casual", "professional", "friendly"],
        "Emphasis": ["important", "minor", "critical", "trivial", "central"],
        "Structure": ["linear", "nonlinear", "hierarchical", "sequential", "random"],
```

```
2808
2809            "Complexity": ["simple", "complex", "intricate", "basic", "elaborate"],
2810            "Consistency": ["consistent", "inconsistent", "variable", "sporadic", "steady"],
                "Informativeness": ["high", "low", "medium", "minimal", "extensive"]
2811        }
2812
            prompts = []
2813        used_categories = set()
2814
            for stage in range(5):
2815            if history:
                    patrols, speeding, reward_sender, reward_receiver = zip(*history)
2816
                    patrol_efficiency = evaluate_patrol_efficiency(patrols[-1], speeding[-1])
2817            reward_trend_sender = evaluate_reward_trend(reward_sender)
2818            reward_trend_receiver = evaluate_reward_trend(reward_receiver)
                    speeding_pattern = evaluate_speeding_pattern(speeding)
2819            patrol_distribution = evaluate_patrol_distribution(patrols)
2820
                    # Complex decision-making based on multiple factors
2821            if patrol_efficiency < 0.5 and speeding_pattern == "random":
2822                category = "Tone"
                        word = "direct"
2823            elif reward_trend_sender == "decreasing" and patrol_distribution == "uneven":
2824                category = "Informativeness"
                        word = "extensive"
2825            elif reward_trend_receiver == "increasing" and patrol_efficiency > 0.7:
2826                category = "Specificity"
                        word = "precise"
2827            elif speeding_pattern == "consistent" and patrol_distribution == "even":
2828                category = "Clarity"
                        word = "clear"
2829            elif reward_trend_sender == "stable" and patrol_distribution == "decreasing":
2830                category = "Structure"
                        word = "linear"
2831            else:
2832                category = random.choice(list(sender_words.keys()))
                        word = random.choice(sender_words[category])
2833
                    while category in used_categories:
2834                category = random.choice(list(sender_words.keys()))
                        word = random.choice(sender_words[category])
2835
2836            else:
2837                category = random.choice(list(sender_words.keys()))
                    word = random.choice(sender_words[category])
2838
            prompts.append((category, word))
2839        used_categories.add(category)
2840        history.append(([random.randint(0, 1) for _ in range(3)], [random.randint(0, 1) for _
2841        ↪ in range(3)], random.random(), random.random()))
2842
            return prompts
2843
2844    def generate_receiver_prompts(history):
2845        receiver_words = {
            "Risk-Preference": ["cautious", "bold", "balanced", "risk-averse", "reckless"],
2846        "Attention": ["focused", "distracted", "alert", "inattentive", "engaged"],
            "Decision-Making": ["rational", "impulsive", "deliberate", "hasty", "calculated"],
2847        "Trust": ["high", "low", "moderate", "skeptical", "confident"],
2848        "Emotional-State": ["calm", "anxious", "frustrated", "neutral", "excited"],
            "Information-Processing": ["slow", "fast", "thorough", "superficial", "efficient"],
2849        "Adaptability": ["flexible", "rigid", "adjustable", "stubborn", "open"],
2850        "Compliance": ["obedient", "defiant", "cooperative", "reluctant", "agreeable"],
            "Responsiveness": ["quick", "slow", "moderate", "delayed", "immediate"],
2851        "Memory": ["sharp", "forgetful", "average", "short-term", "long-term"]
2852        }
2853
            prompts = []
2854        used_categories = set()
2855
            for stage in range(5):
2856            if history:
                    patrols, speeding, reward_sender, reward_receiver = zip(*history)
2857
                    patrol_efficiency = evaluate_patrol_efficiency(patrols[-1], speeding[-1])
2858            reward_trend_receiver = evaluate_reward_trend(reward_receiver)
2859            speeding_pattern = evaluate_speeding_pattern(speeding)
                    patrol_distribution = evaluate_patrol_distribution(patrols)
2860
2861                if reward_trend_receiver == "increasing" and patrol_efficiency > 0.7:
```

```python
                category = "Trust"
                word = "high"
            elif speeding_pattern == "consistent" and patrol_distribution == "even":
                category = "Compliance"
                word = "obedient"
            elif reward_trend_receiver == "decreasing" and speeding_pattern == "random":
                category = "Risk-Preference"
                word = "bold"
            elif patrol_distribution == "uneven" and reward_trend_receiver == "stable":
                category = "Adaptability"
                word = "flexible"
            elif patrol_efficiency < 0.5 and speeding_pattern == "random":
                category = "Attention"
                word = "focused"
            else:
                category = random.choice(list(receiver_words.keys()))
                word = random.choice(receiver_words[category])

            while category in used_categories:
                category = random.choice(list(receiver_words.keys()))
                word = random.choice(receiver_words[category])

        else:
            category = random.choice(list(receiver_words.keys()))
            word = random.choice(receiver_words[category])

        prompts.append((category, word))
        used_categories.add(category)
        history.append(([random.randint(0, 1) for _ in range(3)], [random.randint(0, 1) for _
        ↪  in range(3)], random.random(), random.random()))

    return prompts
```

Listing 6: Another generated conditional prompt function of LAE in the S3 setting.

In contrast, the second set introduces custom evaluation functions, such as evaluate_patrol_efficiency and evaluate_reward_trend, to assess trends in the interaction history. This allows for more complex decision-making, where the system not only reacts to immediate conditions but also adapts to evolving patterns in rewards, patrol effectiveness, and speeding behavior. As a result, the second set generates more nuanced prompts, making it more flexible and suitable for handling sophisticated, multi-stage interactions.

## H LIMITATIONS AND FUTURE WORK

While our approach offers promising results, it faces several limitations, both inherent to LLMs and game theory individually, as well as their integration. First, although LLMs have been widely employed to simulate human behavior, concerns remain regarding the fidelity of these simulations when applied to real-world interactions (Agnew et al., 2024). This raises questions about the generalizability of conclusions drawn from such models in practical scenarios. Second, the computational cost of our method is significant. Although our experiments rely solely on LLM inference without the need for additional training or fine-tuning, the process of traversing large game trees and solving for equilibria requires frequent LLM calls, which is resource-intensive. This presents a scalability challenge, particularly when dealing with more complex strategic environments. A further limitation lies in the control of LLM output. Our method relies on writing style to influence LLM behavior, which can be restrictive. In future work, we intend to explore more flexible prompt optimization strategies, or alternatively, pursue more efficient approaches for fine-tuning LLM parameters to better control output signals.

Additionally, we aim to address the non-uniqueness and inefficiency of equilibria in mixed-motive games, an important aspect not explored in this paper. While the VBP framework effectively solves Bayesian persuasion problems, incorporating the Price of Anarchy (PoA) as an optimization objective could help quantify and minimize efficiency loss from suboptimal equilibria. This enhancement would guide VBP toward selecting more efficient equilibria, improving its solution quality and applicability in scenarios with multiple equilibria.

In terms of the BP problem, our study primarily examines a simplified setting with one sender and one receiver. While this is a fundamental setup, it does not capture the complexity of real-world BP scenarios, which often involve multiple participants (Castiglioni et al., 2021; Koessler et al., 2022b;a; Hossain et al., 2024). Extending our framework to accommodate multiple senders and receivers could provide more practical insights and applications. Additionally, although multistage BP is considered in our experiments, a deeper investigation into the dynamics of these stages is needed. Specifically, we plan to further explore the receiver's bargaining behavior, drawing connections to established bargaining game theories (Nash et al., 1950; Nash, 1953; Maschler et al., 2013). This could ultimately strengthen the receiver's resistance to persuasion, offering a more robust counter-strategy in BP scenarios.

