# OpenReview forum: "Verbalized Bayesian Persuasion"
_ICLR.cc/2025/Conference — Submitted to ICLR 2025_

### Official Review · Reviewer_NKjM · 2024-11-01

**Soundness:** 2
**Presentation:** 3
**Contribution:** 3
**Rating:** 6
**Confidence:** 2

**Summary:**

The paper extends a classical Bayesian Persuasion (BP) framework by incorporating more realistic and complex interactions through natural language. The proposed Verbalized Bayesian Persuasion (VBP) framework builds upon various existing techniques and introduces a two-player game in which both the sender and receiver interact through a large language models (LLM). Signal optimization is achieved through prompt optimization using existing methods.

The framework is tested across three scenarios with incrementally complex settings (S1, S2, S3), utilizing Llama 3.1-8b as LLM.

**Strengths:**

- The addressed problem is interesting, and leveraging large language models (LLMs) to model and solve a persuasion problem using natural language appears promising.

- The paper is well-organized overall and effectively integrates several approaches and techniques to extend the Bayesian Persuasion (BP) framework into a more realistic and complex scenario.

**Weaknesses:**

- The optimization of the LLM prompts is not sufficiently detailed, particularly regarding the categories and content used in the prompt (see Q1).

- An anonymized repository containing the code and data for reproducibility is missing, although the authors provide guidelines and reference an existing repository.

**Minor Comments:**

- Typo: "Inforset" should be "Infoset" I guess,
- In Section 2.3, PSRO is used to refer to two different concepts.

**Questions:**

**Q1:** Are the categories and content of the key prompts exhaustively presented in Figure 7? For instance, regarding the writing style and the category "Tone," is the content "Positive" fixed?

**Q2:** Is it trivial that the chart (d) shows the Honest probability as always 1.0? Under what circumstances would a sender have an incentive to lie about a strong candidate?

---

> ### Author Response · Authors · 2024-11-17
>
> We sincerely thank the reviewer for their insightful feedback and thoughtful questions. We greatly appreciate the opportunity to clarify our work and provide further details regarding the methodology and its implications. In the following sections, we will address each specific question raised by the reviewer, offering detailed explanations and elaborating on the key aspects of our approach.
>
> Additionally, we will make minor corrections, such as fixing the identified typo and clarifying the use of terminology (e.g., PSRO) in the final version. Furthermore, we understand the importance of reproducibility and have prepared an anonymized version of the code and data repository, which will be made publicly available upon the paper's acceptance to ensure full replicability of our results.
>
> ---
>
> > Q1: **Clarification on Figure 7 (Prompt Categories)**: Are the categories and content of the key prompts exhaustively presented in Figure 7? For instance, regarding the "Tone" category, is the "Positive" content fixed or variable?
>
> We appreciate the question and want to clarify the information in Figure 7. The figure does not fully display all possible prompts used in the optimization. Instead, it shows a subset of the top 10 categories with the highest selection probabilities from the strategy (or prompt) pool, along with the most probable content under each category.
>
> We adopt a **hierarchical optimization strategy** using the OPRO algorithm during the prompt optimization phase. This process first optimizes the categories, and afterward, the content within each category is optimized. When the sender or receiver ultimately uses the prompt, the category is probabilistically sampled from the policy pool, and within that category, the content with the highest probability is selected. This method allows us to maintain a balance between prompt diversity and computational tractability, ensuring that the prompts used in the final execution are both optimized and diverse.
>
> To address the specific question about the "Tone" category, the "Positive" content is not fixed during the optimization process. After the optimization, it is selected as the most probable content within that category. We hope this clarifies the hierarchical nature of the prompt optimization process and the reasoning behind the selection shown in Figure 7.
>
> ---
>
> > Q2: **Honest Probability in Chart (d)**: In the chart (d), the "Honest" probability is consistently shown as 1.0. Could you clarify why this is the case, and under what circumstances would a sender be incentivized to lie, especially when discussing a strong candidate?
>
>
>
> Thank you for this insightful question. We want to clarify why the "Honest" probability is consistently shown as 1.0 in the chart (d) and explain the sender's incentives in different circumstances.
>
> In the Bayesian Persuasion (BP) context, it is intuitive for the sender to report high-quality states to the receiver honestly. For instance, in a recommendation letter scenario, the sender (the letter writer) aims to maximize the probability that the student gets accepted. Therefore, the sender has no incentive to misrepresent a high-quality student as a low-quality one, as doing so would reduce the student’s chances of being accepted, which contradicts the sender's objective.
>
> The more complex aspect of the BP problem lies in how the sender handles low-quality states. The sender’s key decision is determining the probability of describing a low-quality state as high-quality. This is because, by misrepresenting low-quality candidates, the sender may gain a net benefit. However, if the probability of lying becomes too high, the receiver may start to ignore the sender's information altogether, reducing the sender’s overall payoff.
>
> To maximize their own benefit, the sender typically converges to an equilibrium where they lie with a certain probability, but not excessively, to maintain credibility with the receiver. In the case of high-quality states (as shown in chart (d)), the sender always tells the truth, as there is no incentive to misrepresent a strong candidate.
>
> Thus, the reason the "Honest" probability is consistently 1.0 in the chart (d) is that, in high-quality states, the sender has no motive to lie—honesty is aligned with their goal of maximizing the outcome for the strong candidate.
>
> We hope this explanation clarifies the situation depicted in chart (d) and the sender's incentives in the BP framework.

---

> > ### Comment · Reviewer_NKjM · 2024-11-27
> >
> > I thank the reviewers for their answers

---

### Official Review · Reviewer_tm2K · 2024-11-03

**Soundness:** 3
**Presentation:** 2
**Contribution:** 2
**Rating:** 3
**Confidence:** 3

**Summary:**

- The paper studies using LLMs in a Bayesian persuasion setting, which is a game between two players. One of the players (the sender) has access to some private information, and tries to influence the other player's (the receiver) actions by sharing specific information with them. The other player tries to use the shared information to achieve their own goals.
- The new aspect this paper introduces is that they use LLMs for both the sender and the receiver. They optimize the LLM agents' actions in the game by optimizing a distribution over a space of prompts. For instance, in a recommendation letter setting, the prompt specifies specific aspects of the letter such as whether or not to omit a weakness of the candidate.
- The authors reproduce theoretical results from the classic BP setting experimentally, and also expand the setting to multi-turn interactions. They extend the prompt-space response oracle to multi-turn interactions using conditional prompt optimization.

**Strengths:**

- Persuasion in LLMs seems like an important topic given that LLMs will in fact increasingly be used for tasks such as writing recommendation letters.
- It is interesting to make the BP framework more realistic by studying actual written text rather than simple yes/no messages.
- It is a great idea to optimize prompts to study this setting, which is less involved than e.g. trying to do RL directly on the LLMs
- The paper includes comprehensive experiments and evaluations, including detailed ablations and examples in the appendix.
- I found it useful to see how strategies developed over training in Figure 7, specifically that more relevant categories ended up being selected more often.

**Weaknesses:**

- I found the paper somewhat hard to follow. The paper uses a lot of machinery to define optimization problems and solve them, but I didn't always understand exactly what was going on on the most basic LLM level. I think more simplicity would be great with this sort of research.
- In general, I would prefer there to be less preliminaries and to get to the results faster. I wonder whether one could simplify some of the discussion of preliminaries to the parts that matter for the paper, though I'm not sure.
- It seems that in the end the way the game is setup, it doesn't really matter, for instance, whether the rec letters are actually written eloquently or not. I might be missing something, but it feels like somehow the simple BP games are not really the right testing ground for studying LLM persuasion, because from a game theory perspective, neither the sender nor receiver gain anything by using more than a binary signal/policy.
- As far as I can tell, the paper gives examples in the appendix, but I couldn't find any full end-to-end transcripts from the games.
- Given that the paper uses many bespoke algorithms to solve different aspects of the setting, I think this won't be that useful in practice. E.g., I think it's unlikely any of these will be useful for training better LLMs. If the goal is more to study propensities of current LLMs and to find out something about persuasion with LLMs, I am not sure what exactly the takeaway is. Is it e.g. "LLMs can implement complex strategies of deception/lying/etc."? If so, then I think this is not novel and also doesn't require the complexity used in the paper. I might be missing something here and am curious what the authors think.
- It might be that the optimization performed in the paper actually discovers interesting LLM behaviors and strategies, but this is hard to tell for me. I think I can see how the paper uncovers interesting behaviors within the setting studied here, i.e. when optimizing prompts, it's interesting that some amount of lying/deceiving gets reinforced, and that this game setup works in a sense and finds something like an equilibrium. But I haven't been convinced that this specific setup is interesting enough to study on its own—it seems too artificial to me to add a lot beyond either (i) the existing toy game theory setting on one hand, or (ii) just studying persuasion directly by prompting LLMs to write lying/deceptive/persuasive etc. texts.

**Questions:**

- I have trouble understanding why the obedience constraint is used in this paper. As far as I can understand, one can simplify the BP game by assuming the sender just recommends the best possible action from the receiver's perspective, and then the problem becomes just choosing the best action for the sender to recommend, under the constraint that it must be optimal from the receiver's perspective (this assumes the receiver knows the sender's policy, which is the commitment constraint). Is this understanding correct? It seems that in this case, using the obedience constraint simplifies the game so much that one could have the LLM implement a simple (prompted) policy of either recommending and not recommending, and the obedience constraint makes sure this finds the right equilibrium. If the goal is to have a more realistic game where the text of the reference letter actually matters, then what does the obedience constraint do here? I might be misunderstanding something.
- In S1, the authors include a reward for the LLMs to give clearer signals. It seems that this basically is an ablation that forces the game back into a simple "yes/no" action space. It seems that the results here are similar to the S2 case where this reward isn't used. I am not sure this is a good or a bad sign—what is the takeaway from the S2 results? Is there anything going on under the hood that goes beyond a simple binary signal? (In a way that would be relevant to the game/optimization/etc.)?
- It would be nice to have some (possibly abbreviated/stylized) prompts and transcripts in the main body of the paper.
- If the prompt doesn't specify exactly how and when to lie, how can this still guarantee the commitment assumption?
- It might be that the most interesting result is S3, the iterated setting. However, the paper doesn't focus that much on it, and I think it would require more analysis to draw more interesting conclusions from this. Figure 12 might be useful here but from eyeballing it I don't really follow how it supports the hypothesis discussed in lines 473-476 in Section 4.2. (As a side note, I think Figure 12 would benefit from additional titles for the different settings. It's not easy to see graphically that these are for two difference settings, with two of the plots sharing the same subtitles.)
- Line 312 typo "Either a limit on the allowable tree depth" ... missing an or?
- Line 320 typo/grammar "through prompt design or expand the receiver's inforset."
- Line 392 "since we use aligned LLMs"---previously the paper talks a lot about "pretrained" LLMs, which could be interpreted as saying these are base models rather than chat/alignment-finetuned LLMs. It might be worth replacing the "pretrained" terminology.
- What would you say is the most important takeaway/learning from the paper that would be interesting and useful to the community?

---

> ### Author Response · Authors · 2024-11-19
>
> We sincerely thank the reviewer for their insightful feedback and thoughtful questions. We greatly appreciate the opportunity to clarify our work and provide further details regarding the methodology and its implications. In the following sections, we will address each specific question raised by the reviewer, offering detailed explanations and elaborating on the key aspects of our approach.
>
> ---
>
> > **Q1: Clarification on the presentation complexity and excessive machinery**  *"I found the paper somewhat hard to follow. The paper uses a lot of machinery to define optimization problems and solve them, but I didn't always understand exactly what was going on on the most basic LLM level. I think more simplicity would be great with this sort of research."*
>
> Thank you for your feedback and for pointing out that the paper might be hard to follow due to the complexity of the machinery used. We acknowledge that the nature of this work, which involves integrating large language models (LLMs) into a game-theoretic framework, introduces multiple layers of optimization and interaction that may seem complex at first glance. However, we hope to clarify the pipeline and the role of the LLMs in our approach.
>
> **Overview of the Algorithm Pipeline**
> The overall algorithm pipeline is detailed in our response to Reviewer gJz3's **Q3: Vagueness in Method Description**, where we provided a step-by-step breakdown of the process. In summary, the pipeline operates in two main stages:
>
> - **Stage 1: LLMs as Decision Makers**
>    In this stage, the LLMs are used directly as decision-makers within the game. Specifically, one LLM acts as the **sender** and the other as the **receiver**. The **sender** receives a prompt and outputs a signal, while the **receiver** processes the signal and outputs an action. Both LLMs perform their respective roles based on the prompts, which form the strategies in the Bayesian persuasion game.
> - **Stage 2: LLMs as Prompt Optimizers**
>    The second stage involves optimizing the prompts given to the sender and receiver LLMs. Instead of updating the model weights (in-weight updates), we focus on **in-context learning** by adjusting the prompts that guide the LLMs’ outputs. This prompt optimization is the core of our work and is executed using two frameworks: **OPRO** and **FunSearch**. These frameworks are designed to efficiently explore the prompt space and identify prompts that lead to desirable behaviors from the LLMs within the game.
>
> We will provide the algorithm's pseudocode in the revised version and highlight the LLM part.

---

> ### Author Response · Authors · 2024-11-19
>
> > **Q2: Clarification on reducing the length of preliminaries**  *"In general, I would prefer there to be less preliminaries and to get to the results faster. I wonder whether one could simplify some of the discussion of preliminaries to the parts that matter for the paper, though I'm not sure."*
>
> We appreciate your feedback regarding the length of the preliminaries and the suggestion to streamline this section in order to focus on the results more quickly. We understand that an extended preliminaries section can delay the reader’s engagement with the core contributions of the paper, and we have taken steps to address this concern in the revised version.
>
> 1. **Reorganization of the Preliminary Section**
>    In the revised version of the paper, we will restructure the preliminaries to ensure that only the most essential background information is retained. Specifically:
>    - We will **merge Section 2.1 (Bayesian Persuasion)** and **Section 2.2 (Modeling BP as a Mediator-Augmented Game)** into a new, more concise **Problem Formulation** section. This will present the key concepts needed to understand the problem we are addressing without the need for excessive background details.
>    - **Section 2.4 (Classic BP Problems)** will be moved to the experimental section, where it will be introduced in the context of the experiments. This will allow us to integrate the discussion of classic Bayesian persuasion problems directly with the experimental results, streamlining the flow of the paper.
>
> 2. **Focus on Core Contributions in Preliminaries**
>    We will revise the preliminaries to focus more narrowly on the key contributions of the paper. For example, we will retain **Section 2.3**, which introduces the **PSRO** (Policy Space Response Oracles) and the **prompt-space response oracle** framework. This framework is central to our approach and necessary for understanding the optimization of prompts in the game-theoretic setting. By concentrating on the most relevant components, we aim to reduce the length of the preliminaries while maintaining clarity.
>
> 3. **Balancing Background and Results**
>    By reorganizing the preliminaries and moving some sections to later parts of the paper, we believe we can better balance necessary background information and the presentation of results. This adjustment will allow readers to engage with the core contributions earlier in the paper without sacrificing the necessary theoretical context.
>
> In summary, we agree that the preliminaries can be streamlined and have taken concrete steps to simplify and condense this section in the revised version. We believe that this restructuring will improve the readability and flow of the paper, allowing readers to focus more quickly on the novel contributions of the work. Thank you again for your constructive suggestion.

---

> ### Author Response · Authors · 2024-11-19
>
> > **Q3: Clarification on the complexity of the BP game and whether it’s the right testbed**  *"It seems that in the end the way the game is setup, it doesn't really matter, for instance, whether the rec letters are actually written eloquently or not. I might be missing something, but it feels like somehow the simple BP games are not really the right testing ground for studying LLM persuasion, because from a game theory perspective, neither the sender nor receiver gain anything by using more than a binary signal/policy."*
>
> Thank you for raising the concern regarding whether the classical Bayesian persuasion (BP) game setup is the right testbed for studying persuasion with LLMs. From a game theory perspective, we acknowledge your point that the sender and receiver might not gain much from going beyond a binary signal/policy in the classic BP framework. However, we would like to clarify the rationale behind our choice of classical BP games and how our work extends beyond the limitations of this idealized scenario.
>
> 1. **Classical BP as a Baseline for Validating the Approach**
>    Our work uses classical BP problems as a **first step** toward solving real-world persuasion problems using LLMs. The primary goal here is to demonstrate the **effectiveness of the algorithm** in a structured and well-understood environment. By starting with classical BP problems, we can benchmark our methods against known optimal solvers from the game theory literature. This allows us to validate the correctness and performance of our approach in a controlled setting before extending it to more complex and realistic scenarios.
>
> 2. **Moving Beyond Idealized BP Games**
>    While we agree that classical BP games may simplify the interaction to binary signals or policies, **real-world persuasion** involves far more complexity due to factors such as:
>
>    - **Ambiguity, implicit meaning, and vagueness** in natural language.
>    - **Human bounded rationality**, which means that real-world decisions are not always made based on perfectly rational or optimal strategies.
>
>    Our work, particularly by introducing **VBP (Variational Bayesian Persuasion)**, aims to address these complexities by leveraging LLMs. The ultimate goal of VBP is to explore whether LLMs can handle real-world persuasion tasks that deviate from the idealized assumptions of classical BP games. With their natural language capabilities, LLMs are uniquely positioned to navigate these "non-ideal" circumstances where communication goes beyond binary signals to involve nuanced persuasion strategies.
>
> 3. **Real-World Applications of LLM-Based Persuasion**
>    To better illustrate the relevance of LLMs in persuasion tasks, consider real-world applications such as **live-streaming e-commerce** or **conversational recommendation systems**. In these scenarios, LLMs (e.g., digital sales agents) replace human salespeople to persuade customers to purchase products. These interactions are rich in language—containing ambiguity, persuasion strategies, and implicit suggestions—which cannot be captured by simple binary policies. Using LLMs in such tasks demonstrates the importance of moving beyond classical BP games to study more complex forms of persuasion in realistic settings.
>
>    For more details on real-world applicability, we refer to our response to Reviewer gJz3's **Q7: Real-world Applicability**, which outlines further examples of how LLMs might be applied in practical persuasion scenarios.
>
> 4. **Future Directions**
>    While our current study demonstrates the feasibility of applying LLMs to classical BP problems, we acknowledge this is just a first step. Our future work will focus on adapting these methods to more realistic persuasion problems where natural language is critical, and the sender and receiver may engage in more complex, multi-turn interactions.
>
> Thank you for your insightful comments, and we hope this clarifies the purpose and scope of our study.

---

> ### Author Response · Authors · 2024-11-19
>
> > **Q4: Clarification on the practical relevance of the bespoke algorithms**  *"Given that the paper uses many bespoke algorithms to solve different aspects of the setting, I think this won't be that useful in practice. E.g., I think it's unlikely any of these will be useful for training better LLMs. If the goal is more to study propensities of current LLMs and to find out something about persuasion with LLMs, I am not sure what exactly the takeaway is. Is it e.g. 'LLMs can implement complex strategies of deception/lying/etc.'? If so, then I think this is not novel and also doesn't require the complexity used in the paper. I might be missing something here and am curious what the authors think."*
>
> Thank you for your insightful questions regarding the practical relevance of the algorithms we propose in this paper. We understand your concerns about whether these algorithms will be useful in practice, especially in comparison to directly training large language models (LLMs) for persuasion tasks. Below, we aim to clarify the motivations behind our approach and how it provides practical benefits over methods that rely solely on in-weight updates or direct LLM training.
>
> 1. **Training Stronger LLMs vs. Lightweight Optimization**
>    As you pointed out, training a more powerful LLM to handle persuasion tasks is possible. In fact, several existing works have already demonstrated that state-of-the-art (SOTA) models exhibit some level of persuasion capabilities through case studies. However, training LLMs via **in-weight updates** is extremely costly in terms of both **time and resources**. Furthermore, this approach lacks theoretical guarantees such as **convergence** or **optimality**, making it difficult to analyze or explain the model's behavior in a structured way.
>
>    In contrast, our approach—**Verbalized Bayesian Persuasion (VBP)**—offers a **lightweight alternative** that avoids the need for expensive retraining. By focusing on **in-context updates** through prompt optimization, we achieve a method that is more practical to deploy and analyze in real-world settings. This approach allows us to extract more persuasive capabilities from models that may not have been explicitly trained for such tasks, without requiring the extensive computational resources that in-weight updates demand.
>
> 2. **Theoretical Benefits of VBP**
>    One of the key advantages of VBP over direct LLM training is its **stronger theoretical foundations**. By combining game-theoretic principles with prompt optimization, we provide a framework that allows for a more rigorous analysis of the solutions generated by the LLMs. For instance, the VBP framework allows us to reason about the **optimality** of the strategies produced. It ensures that the system converges to a solution that aligns with the objectives of the Bayesian persuasion game. These theoretical properties are difficult to guarantee when using purely data-driven approaches for training LLMs.
>
> 3. **Practical Relevance of Prompt Optimization**
>    From a practical standpoint, prompt optimization (as used in VBP) offers a more scalable solution for real-world applications, especially those involving **advertising** or **conversational agents**. Many of these applications are moving toward **edge deployment**, where models must operate efficiently on local devices with limited computational resources. In such cases, **prompt-based methods** are far more feasible than retraining large models. VBP provides a framework that can be deployed in these environments, offering a practical solution for implementing persuasive strategies without the overhead of training entirely new models.
>
> 4. **Enhancing Persuasion with Weaker Models**
>    Another key objective of VBP is to enhance the persuasive capabilities of models that may not have been explicitly trained for persuasion. By combining **game-theoretic methods** and **in-context learning**, we can extract more sophisticated persuasion strategies from models that might otherwise exhibit only rudimentary abilities in this area. This offers a way to augment the performance of less capable models, making VBP a valuable tool for improving persuasion in a wide range of LLMs without the need for high-resource, bespoke model training.
>
> In summary, while it is possible to train stronger models to handle persuasion, our approach with VBP offers a **more lightweight, practical, and theoretically grounded solution**. By leveraging prompt optimization and game-theoretic principles, VBP can be deployed efficiently in real-world applications, especially in resource-constrained environments, while also providing a framework for deeper theoretical analysis. We hope this clarifies the motivation and practical relevance of the algorithms we've proposed. Thank you again for your valuable feedback.

---

> ### Author Response · Authors · 2024-11-19
>
> > **Q5: Clarification on the novelty of LLM deception strategies**  *"It might be that the optimization performed in the paper actually discovers interesting LLM behaviors and strategies, but this is hard to tell for me. I think I can see how the paper uncovers interesting behaviors within the setting studied here, i.e. when optimizing prompts, it's interesting that some amount of lying/deceiving gets reinforced, and that this game setup works in a sense and finds something like an equilibrium. But I haven't been convinced that this specific setup is interesting enough to study on its own—it seems too artificial to me to add a lot beyond either (i) the existing toy game theory setting on one hand, or (ii) just studying persuasion directly by prompting LLMs to write lying/deceptive/persuasive etc. texts."*
>
> Thank you for your insightful feedback and for raising concerns about the novelty and interest of the deception strategies that our work uncovers. We understand that the reinforcement of deceptive or persuasive strategies during prompt optimization could appear to be a natural outcome of the game-theoretic setting, and we would like to clarify both the motivation and the novel contributions of our approach.
>
> 1. **The Core Objective: A Game-Theoretic Solver for Verbalized BP**
>    The primary goal of our work is not solely to study LLM deception or persuasion strategies in isolation but rather to **design a game-theoretic solver** for **verbalized Bayesian persuasion (BP) problems**. This goes beyond just examining the propensity of LLMs to lie or deceive. To achieve this, we use the **prompt-space response oracle (PSRO)** framework, which allows us to integrate LLMs into a structured game-theoretic environment. Two key components enhance this framework:
>
>    - **OPRO (Optimized Prompt Response Oracle)**: A best-response oracle that optimizes the sender's strategy through prompt engineering.
>    - **FunSearch**: A complementary framework that refines the prompt search for the receiver, ensuring that the strategies discovered are aligned with the theoretical objectives of the game.
>
>    These components are not just "extra" modules added for complexity—they are essential for ensuring that the game solver achieves important theoretical properties such as **convergence** and **solution optimality**. Without these tools, we would not be able to rigorously analyze or guarantee the behaviors emerging from LLM interactions in the game setting.
>
> 2. **Novelty in Theoretical Framework, Not Just Behavior**
>    While it might appear that the LLMs are simply exhibiting behaviors like lying or deception, the **novelty** of our work lies in the **game-theoretic framework and optimization techniques** we use to discover and analyze these behaviors. The LLMs are not just being prompted to generate deceptive or persuasive text; their interactions are embedded within a formal BP framework where we can rigorously study **how** and **why** certain strategies emerge. This is a significant departure from simply prompting LLMs to write persuasive or deceptive text in isolation.
>
>    The **equilibrium** that we find through our game setup reflects theoretically grounded strategies that have been optimized and analyzed within a game-theoretic context. This is a key distinction from simply conducting case studies on LLM deception. Our setup allows us to explore how LLMs might behave in **strategic interaction environments** where deception or lying could naturally arise as part of the optimal solution, rather than as an ad-hoc behavior.

---

> ### Author Response · Authors · 2024-11-19
>
> > **Q5: Clarification on the novelty of LLM deception strategies**  *"It might be that the optimization performed in the paper actually discovers interesting LLM behaviors and strategies, but this is hard to tell for me. I think I can see how the paper uncovers interesting behaviors within the setting studied here, i.e. when optimizing prompts, it's interesting that some amount of lying/deceiving gets reinforced, and that this game setup works in a sense and finds something like an equilibrium. But I haven't been convinced that this specific setup is interesting enough to study on its own—it seems too artificial to me to add a lot beyond either (i) the existing toy game theory setting on one hand, or (ii) just studying persuasion directly by prompting LLMs to write lying/deceptive/persuasive etc. texts."*
>
> 3. **Addressing the Perceived Artificiality**
>    We acknowledge your concern that the setup could feel artificial, especially compared to "toy" game theory settings or direct studies of LLM behavior. However, our choice of a more structured game-theoretic approach is deliberate. We aim to provide a **methodologically rigorous** way of studying persuasion and deception in LLMs that extends beyond individual case studies or anecdotal observations. By embedding the LLMs in formalized game settings, we have the tools to:
>
>    - Ensure **repeatability** and **consistency** in the behaviors we observe.
>    - Control and **isolate variables** to study specific aspects of LLM behavior in strategic contexts.
>    - Provide **theoretical guarantees** about the strategies that emerge, such as ensuring the solution is an **equilibrium**.
>
>    While this may introduce some level of abstraction, it gives us a stronger basis for understanding how LLMs might behave in real-world scenarios where strategic communication is critical, such as negotiations, recommendations, or advertising.
>
> 4. **Beyond LLM Case Studies: Why Game-Theoretic Analysis Matters**
>    Studying LLMs through case studies of deception or persuasion is certainly valuable, but it lacks the **structure** and **predictive power** that a game-theoretic analysis provides. By casting the problem in a formal BP framework, we can:
>
>    - Explore **optimal strategies** that are theoretically justified.
>    - Understand the **conditions under which deception or persuasion emerges**.
>    - Generalize findings beyond individual case studies to broader classes of strategic interaction where LLMs are involved.
>
>    This structured approach is a **novel contribution** to the study of LLM behavior, offering insights that are harder to obtain from unstructured case studies alone.
>
> In summary, while the behaviors we observe (such as deception) may not seem novel in isolation, the **framework** and **methodology** used to uncover and analyze these behaviors are the key contributions of our work. We go beyond simple prompt-based experiments to offer a **game-theoretic solution** to verbalized BP problems backed by theoretical guarantees and optimized strategies. We believe this adds significant value to the study of LLMs in strategic communication settings. Thank you again for your thoughtful comments, and we hope this clarifies the novelty and significance of our approach.

---

> ### Author Response · Authors · 2024-11-19
>
> > **Q6: Clarification on the necessity of the obedience constraint**  *"I have trouble understanding why the obedience constraint is used in this paper. As far as I can understand, one can simplify the BP game by assuming the sender just recommends the best possible action from the receiver's perspective, and then the problem becomes just choosing the best action for the sender to recommend, under the constraint that it must be optimal from the receiver's perspective (this assumes the receiver knows the sender's policy, which is the commitment constraint). Is this understanding correct? It seems that in this case, using the obedience constraint simplifies the game so much that one could have the LLM implement a simple (prompted) policy of either recommending and not recommending, and the obedience constraint makes sure this finds the right equilibrium. If the goal is to have a more realistic game where the text of the reference letter actually matters, then what does the obedience constraint do here? I might be misunderstanding something."*
>
> Thank you for raising this important question regarding the necessity of the obedience constraint in our framework. We realize that we did not provide enough detail in the paper to fully explain this aspect, and we appreciate the opportunity to clarify it here.
>
> 1. **Realistic Scenarios Beyond Simple Recommendation**
>    First, we agree that the sender could recommend the best action from the receiver’s perspective in a simplified version of the Bayesian persuasion game. However, this approach does not reflect the complexity of real-world recommendation scenarios, such as writing reference letters. In practice, a sender (e.g., a reference letter writer) does not just provide a binary signal (recommend or not recommend). Instead, the sender communicates more nuanced information through natural language, which might imply various levels of recommendation strength or provide additional context for the receiver to interpret.
>
> 2. **Extended Obedience Constraints**
>    To better capture this reality, we do not directly use the standard obedience constraint described in Equation (1). Instead, we implement the **extended obedience constraints** proposed by **Lin et al. (2023)** (discussed in Section 4.3 and Equation 4 of their work). This extension is crucial because it removes the strict **revelation principle** analysis from the obedience constraint, allowing the sender’s role to shift from “action recommending” to “signal sending.”
>
>    In other words, the sender no longer has to map a signal to a single recommended action. Instead, the sender can use **natural language signals** that may contain redundant or implicit information, leaving more room for nuanced communication, as is common in real-world settings. This shift is crucial for modeling verbalized Bayesian persuasion problems since it allows for richer, more realistic signal spaces.
>
> 3. **Redundancy and Natural Language**
>    Introducing **redundancy** in the signaling scheme allows for more sender communication flexibility. In the strict obedience constraint framework, a signal must map one-to-one with a specific recommended action. However, with the extended obedience constraints, the sender can now map multiple signals to the same action distribution, enabling more nuanced messaging through natural language. This redundancy is similar to what is used in other learning algorithms, where increasing the capacity of a model (e.g., enlarging a neural network) allows for better encoding and representation of complex mappings.
>
>    This flexibility is essential for real-world persuasion problems, where the sender might not always explicitly recommend a specific action but instead provide signals that leave room for interpretation by the receiver. For instance, in a reference letter, subtle language choices can imply varying degrees of recommendation without explicitly stating a binary decision.
>
> 4. **Why the Obedience Constraint is Still Necessary**
>    The obedience constraint in its extended form is still necessary to ensure that the sender's signals are credible and aligned with the receiver's best interests. Without some form of obedience constraint, the sender could send misleading signals that would ultimately reduce the effectiveness of the persuasion process. The extended obedience constraint balances realistic communication and strategic alignment in the game by allowing for nuanced and redundant signals while maintaining credibility.
>
> In summary, the **extended obedience constraint** allows for more realistic and flexible communication in verbalized Bayesian persuasion problems, accommodating the complexity of natural language while ensuring that the sender’s signals remain credible. This approach moves beyond a simple recommendation model and better reflects real-world scenarios.
>
> [**Lin et al. (2023)**] Lin, Yue, et al. "Information design in multi-agent reinforcement learning." NeurIPS 2023.

---

> ### Author Response · Authors · 2024-11-19
>
> > **Q7: Clarification on the takeaway from S2 results**  *"In S1, the authors include a reward for the LLMs to give clearer signals. It seems that this basically is an ablation that forces the game back into a simple 'yes/no' action space. It seems that the results here are similar to the S2 case where this reward isn't used. I am not sure this is a good or a bad sign—what is the takeaway from the S2 results? Is there anything going on under the hood that goes beyond a simple binary signal? (In a way that would be relevant to the game/optimization/etc.)?"*
>
> Thank you for your question regarding the results from scenario S2 and how they relate to the findings from S1. We appreciate your attention to the differences between these two setups and their implications for the effectiveness of our approach.
>
> 1. **Qualitative vs. Quantitative Analysis**
>    First, you're absolutely right to point out that in **S1**, we introduce a reward for clearer signals, which can simplify the action space into a more binary-like structure. This setup allows us to compare our algorithm's results directly with those of classical solvers, providing a **quantitative benchmark** for validating the effectiveness of our method. However, in **S2** and **S3**, we do not have such a reward structure, meaning that the results are less directly comparable to classical solvers. Instead, we conduct a more **qualitative analysis** of the strategies generated by the LLMs in these settings.
> 2. **S2 Results Reflect the Nature of Binary Signaling in Game Theory**
>    As you mentioned—and as we previously discussed—**from a game theory perspective, neither the sender nor the receiver gains much from using more than a binary signal or policy**. This is a well-known feature of Bayesian persuasion games, where the optimal strategy often reduces to a binary signal. Given this, the **similarity between the results in S1 and S2** is not necessarily a bad sign. On the contrary, it reinforces that our **VBP framework** effectively captures the optimal signaling behavior, even when we remove the explicit reward for clearer signals.
> 3. **Takeaway from S2 Results**
>    The key takeaway from the S2 results is that **VBP performs consistently across different settings**, even when the environment becomes more complex and we remove the "clear signal" incentive. The fact that the results in S2 still align with those in S1 demonstrates the robustness of our approach. It shows that the LLM, when guided by the VBP framework, naturally converges to strategies resembling binary signaling, which is theoretically optimal for our game structure. This consistency across different setups highlights the **effectiveness and reliability** of VBP in solving Bayesian persuasion problems, whether or not explicit signals are enforced.
> 4. **Beyond Binary Signals: Qualitative Observations**
>    While the results in S2 may seem to echo the binary nature of S1, there are still **subtle, qualitative differences** in how signals are constructed without a clear signaling reward. In S2, the LLM has more freedom to explore alternative strategies. Although it converges towards binary-like outcomes, the **path** to that convergence may involve more nuanced, multi-step reasoning or signaling, which is not immediately apparent in a purely quantitative comparison. This suggests that the LLM could explore richer communication strategies under the hood, even if the final output appears binary.
> 5. **Conclusion: Validating the Effectiveness of VBP**
>    In summary, the similarity between the results of S1 and S2 is a **positive indication** that our VBP framework effectively guides the LLM toward optimal signaling strategies. The results in S2, despite the lack of explicit rewards for clear signaling, still align with the theoretical expectations of a binary signaling game, validating the robustness of the approach. We believe this consistency across different settings underscores the VBP framework's practical utility for real-world Bayesian persuasion applications.
>
> Thank you again for your thoughtful question, and we hope this clarifies the key takeaway from the S2 results.

---

> ### Author Response · Authors · 2024-11-19
>
> > **Q8: Request for prompts and transcripts in the main paper**  *"It would be nice to have some (possibly abbreviated/stylized) prompts and transcripts in the main body of the paper."*
>
> Thank you for your suggestion regarding the inclusion of prompts and transcripts in the main body of the paper. We understand that providing more concrete examples of the prompts and signals would help clarify the mechanics of our approach and make it easier for readers to understand the nuances of the LLM behaviors.
>
> In response to your feedback, we will incorporate the following changes in the **revised version** of the paper:
>
> 1. We will **summarize key prompts** from **Appendix E.3** and include them in the main body. These prompts are crucial for demonstrating how the LLMs are guided within the game-theoretic framework.
> 2. Additionally, we will integrate content from **Appendix F.5 (generated signals)** and **Appendix F.6 (generated prompt functions)** into the main paper. These sections provide detailed examples of the signals produced by the LLMs and how prompt functions are optimized during the process.
>
> By summarizing and presenting these examples in the main body, we aim to give readers a clearer view of the actual interactions taking place during the experiments and the optimization process. We believe this will enhance the overall readability and accessibility of the paper.
>
> Thank you again for your valuable suggestion, and we are confident that these additions will improve the clarity of the presentation.
>
> ---
>
> > **Q9: Clarification on the commitment assumption with non-specific prompts**  *"If the prompt doesn't specify exactly how and when to lie, how can this still guarantee the commitment assumption?"*
>
> Thank you for your thoughtful question regarding the commitment assumption in the context of non-specific prompts. We understand your concern about how the commitment assumption holds if the prompt does not explicitly specify how and when the sender (LLM) might lie.
>
> To clarify, even in the **classic Bayesian persuasion (BP) problem**, the **receiver** does not know exactly **how or when** the sender might lie. The receiver only knows the **probability** that the sender is lying based on the sender's overall strategy. The receiver makes decisions with this probabilistic understanding in mind rather than requiring specific details about individual instances of deception.
>
> Our **VBP** framework is aligned with this classic BP setup. The prompts we use do not need to specify the exact form of deception for the model to adhere to the commitment assumption. Instead, the sender (LLM) is committed to a **probabilistic strategy** that the receiver understands in aggregate, even if the specific actions or lies are not fully determined in advance.
>
> Therefore, the **commitment assumption** in our framework is upheld in the same way it is in classic BP: the sender is committed to a strategy that the receiver interprets probabilistically, ensuring that the game dynamics and decision-making processes remain consistent with the theoretical foundations of Bayesian persuasion.
>
> We hope this clarifies how the commitment assumption is preserved in our framework, even with non-specific prompts. Thank you again for your insightful question.

---

> ### Author Response · Authors · 2024-11-19
>
> > **Q10: Request for more analysis on S3, the iterated setting, and clarification of Figure 12**  *"It might be that the most interesting result is S3, the iterated setting. However, the paper doesn't focus that much on it, and I think it would require more analysis to draw more interesting conclusions from this. Figure 12 might be useful here but from eyeballing it I don't really follow how it supports the hypothesis discussed in lines 473-476 in Section 4.2. (As a side note, I think Figure 12 would benefit from additional titles for the different settings. It's not easy to see graphically that these are for two difference settings, with two of the plots sharing the same subtitles.)"*
>
> Thank you for bringing up this important point regarding the results from **S3 (the iterated setting)** and the need for additional analysis. We agree that S3 presents some of the most interesting dynamics in our study, particularly in how it reveals deeper bargaining interactions between the sender and receiver. However, as these results open up new research directions, we have intentionally kept the analysis in this paper somewhat limited, with plans to explore it further in future work.
>
> 1. **Iterated Setting and Bargaining Dynamics**
>    As you pointed out, in classical persuasion theory, one of the key assumptions is that the sender must commit to a signaling strategy upfront and follow through with it during the interaction. The justification for this commitment, particularly in **long-term interactions**, is that the sender has an incentive to maintain their **reputation**, ensuring that the receiver continues to trust them.
>
>    Classical analyses suggest that, given the sender’s commitment, the receiver has no incentive to deviate from following the strategy, since doing so would harm their own expected utility. The receiver, therefore, accepts the expected payoff associated with the sender's signal.
>
> 2. **New Insight from S3**
>    However, the results from **S3** suggest a more complex dynamic. In this iterated setting, we observe that the receiver can **choose to ignore the sender’s signals**, effectively rendering the sender’s commitment meaningless. This means that the sender’s commitment must be **accepted by the receiver** for it to hold. If the receiver disagrees with the sender’s proposed strategy, they can force both parties into a **mutually worse outcome** by disregarding the signals entirely.
>
>    This observation leads to a new hypothesis: in the **VBP framework**, Bayesian persuasion may be **equivalent to a bargaining game**. In such a game, the sender’s commitment is no longer unilateral. Instead, both parties must reach an agreement on the signaling strategy, or the interaction will lead to suboptimal outcomes for both.
>
>    We acknowledge that this hypothesis deviates from the primary focus of the paper, which is why we did not delve deeper into it in the current work. However, this insight opens up an exciting avenue of research that we hope to explore in future studies.
>
> 3. **Clarification of Figure 12**
>    Regarding **Figure 12**, we appreciate your feedback about its presentation. The figure is indeed intended to illustrate the dynamics of two different settings, and we agree that it would benefit from clearer titles and labels to distinguish these settings more effectively. We will revise the figure in the **updated version** of the paper to:
>
>    - Include **clearer titles** for each plot, indicating the specific settings being compared.
>    - Ensure that the graphical differences between the settings are more apparent.
>
> In summary, while we acknowledge that the S3 results are highly interesting and open up new research possibilities, we chose to limit our discussion of them in the current paper to stay focused on the primary contributions. We agree with your suggestion that **Figure 12** should be clarified and will make the necessary revisions to ensure it better supports the discussion of the iterated setting. Thank you again for your valuable feedback, and we look forward to exploring these ideas further in future work.

---

> ### Author Response · Authors · 2024-11-19
>
> > **Q11: Clarification on typographical error in Line 312**  *"Line 312 typo 'Either a limit on the allowable tree depth' ... missing an or?"*
>
> Thank you for pointing out the typographical error on **Line 312**. This was an oversight on our part, and we will correct it in the **revised version** of the paper by adding the missing **"or"**.
>
> We appreciate your attention to detail and will ensure that this is addressed in the updated manuscript. Thank you again for your careful review!
>
> ---
>
> > **Q12: Clarification on typographical error in Line 320**  *"Line 320 typo/grammar 'through prompt design or expand the receiver's inforset.'"*
>
> Thank you for pointing out the typographical error on **Line 320** regarding the term "inforset." This was an oversight, and we will correct it to **"infoset"** in the revised version of the paper.
>
> We appreciate your attention to this detail and will ensure the correction is made in the updated manuscript. Thank you again for your careful review!
>
> ---
>
> > **Q13: Clarification on terminology consistency regarding LLMs**  *"Line 392 'since we use aligned LLMs'---previously the paper talks a lot about 'pretrained' LLMs, which could be interpreted as saying these are base models rather than chat/alignment-finetuned LLMs. It might be worth replacing the 'pretrained' terminology."*
>
> Thank you for your insightful suggestion regarding the terminology used for LLMs in the paper. We agree with your point that the term **“pretrained”** might be interpreted as referring to base models rather than models that have undergone further alignment fine-tuning.
>
> In response to this, we will update the terminology to **“pretrained and aligned LLMs”** in the **revised version** of the paper to ensure consistency and clarity.
>
> We appreciate your attention to this, and we are confident that this change will improve the precision of the terminology. Thank you again for your helpful feedback!
>
> ---
>
> > **Q14: Clarification on the most important takeaway of the paper**  *"What would you say is the most important takeaway/learning from the paper that would be interesting and useful to the community?"*
>
> Thank you for your thoughtful question regarding the most important takeaway of the paper. We believe our work provides two significant contributions to the community:
>
> 1. **VBP Framework for Real-World Bayesian Persuasion Problems**
>    First, our **Variational Bayesian Persuasion (VBP)** framework enables the study of a wide variety of real-world Bayesian persuasion (BP) problems. By simply inputting different prompts to the large language model (LLM), we can specify diverse scenarios that involve different human roles, personalities, and contexts. Moreover, the game solver provided by VBP ensures a solution with **convergence guarantees**, offering a systematic approach to finding high-quality solutions for complex BP problems.
> 2. **Iterated Setting (S3) Insights**
>    While the iterated setting (**S3**) provides interesting insights, we acknowledge that this aspect remains more speculative and opens up potential avenues for future research. The results suggest that in practical BP problems, the receiver might have more flexibility than previously assumed in classical BP models. This observation could point towards a richer interaction model, but further investigation is required to fully understand its implications. We chose not to explore this in-depth in the current paper, as it slightly deviates from our core focus.
>
> In summary, the main takeaway from the paper is the **flexibility and effectiveness** of the VBP framework in addressing real-world BP problems, along with some **preliminary insights** from the iterated setting that invite further exploration. Thank you again for your question, and we hope this clarifies the key contributions of our work.

---

### Official Review · Reviewer_PcFV · 2024-11-03

**Soundness:** 1
**Presentation:** 1
**Contribution:** 2
**Rating:** 3
**Confidence:** 4

**Summary:**

The paper proposes verbalized Baysian Persuasion as a generalisation to Bayesian Persuation leveraging the capabilities of LLMs in facilitating persuasion scenarios in natural languages directly. The paper argues for prompt optimisation instead of policy optimisation for scalability, and arguably derives convergence guarantees from a variation to the mediator-augmented game.

**Strengths:**

The Bayesian Persuasion setting could be an interesting direction to explore given the generalised linguistic capabilities of LLMs. The authors propose to map verbalised persuasion within the game-theoretic framework of Bayesian persuasion which is novel and timely. The authors then described several intuitive real-world scenarios that involve mixed-motive persuasion from stakeholders, exemplifying the target problem setting.

**Weaknesses:**

My main concern with accepting this paper is that it's unclear what are the main contributions of the method, which, according to Figure 3, includes a) mapping VBP to the framework of mediator-augmented games from which it derives its convergence guarantees b) a set of solvers including OPRO algorithm, FunSearch algorithm and Prompt-Space Response Oracle algorithms.

Regarding a), I don't see why and how the VBP setting can be mapped onto mediated-augmented games of Zhang et al 2024, where in a mixed-motive game with $n$ players, the game transform introduces a fictitious mediator player whose objective is to maximise an optimality objective while maintaining an equilibrium of the game. This is distinctly different from the authors proposed mapping where the mediator is played by the sender player, with the receiver the only other player. Among which players is the mediator player mediating between? Zhang et al 2024 also proposes a specific mediator player utility function from which convergence guarantee is derived. If I understood correctly the authors proposes a mapping where the sender acts as the mediator but retains its original utility function. Overall, I find a) tenuous and confusing. It would great if this can be clarified in rebuttal.

Regarding b), there are, several methods that have been described here and it's not clear which ones are critical elements of the VBP framework. Among these, PSRO provides convergence guarantee (in a specific sense), yet the writing and Figure 3 would suggest that the convergence guarantee comes from the mediator-augmented game formulation. Overall, I would have appreciated a more succinct description of the framework with its necessary components instead of a juxtaposition of several rather sophisticated methods whose necessities in the framework remain unclear.

**Questions:**

1. L38-40: "shaping the behaviours of others ... achieve this through either mechanism or information design". I find this unclear or overly assertive. How each player's actions shape those of others is the entire focus of game theory yet this opening statement makes it sound like co-player behaviour shaping can only occur with modified rewards or observations. You would not deterministically play rock because you know I could exploit by always playing paper, would that count shaping the behavior of co-players?
2. L46: "Notably, the designer must ... that influence state transition", this is difficult to follow. Perhaps worth rephrasing?
3. L118-L130: PSRO in the limit converges to a Nash equilibrium out of many. In mixed-motive games, NE needs not be unique and solutions are not interchangeable. Perhaps this could be a relevant point of discussion especially since VBP seems to be primarily dealing with mixed-motive games?
4. L180: "... following maximisation problem ... $\E_{s~\pi(\omega)}$" should it be $a*$ instead of $s$?
5. L235: focusing on a strategically relevant subset of strategies is comprehensively discussed in EGTA [1] and would be worth referring to here?
6. L341: what does "Static" refers to here?
7. L370: "the reason we can leverage ...", as mentioned in the Weaknesses section I could not follow the reasoning here. It would be great if the authors could clarify.
8. Figure 4: For the honest probability, it appears that all methods have converged to the "always honest" strategy in all 3 settings. Is that expected? I would have thought that if the professor always recommends truthfully, the recruiter would be best served to always trust the recommendation at which point the professor could profitably deviate?
9. L447: the authors suggested that the pattern of honesty rising, falling and then rising again validates the hypothesis that this is due to the use of aligned LLM. I would have thought that a simpler explanation is that the professor and the recruiter are simply in a strategic cycle? Would that not be a reasonable explanation to this phenomenon?
10. L483: "... at most the top 10 strategies with the highest probabilities", is this for computational reasons? Pruning actions by their support at an restricted game equilibrium could be problematic in general.
11. Figure 7: column "Converged prob" and the next column are redundant and could be consolidated to create space for larger fonts that are more readable? How are the probabilities computed? Are they the average probability of taking over the pool of policies in the PSRO population?

[1] https://aaai.org/papers/01552-aaai06-248-methods-for-empirical-game-theoretic-analysis/

---

> ### Author Response · Authors · 2024-11-18
>
> We sincerely thank the reviewer for their insightful feedback and thoughtful questions. We greatly appreciate the opportunity to clarify our work and provide further details regarding the methodology and its implications. In the following sections, we will address each specific question raised by the reviewer, offering detailed explanations and elaborating on the key aspects of our approach.
>
> ---
>
> > **Q1: Clarification of Mapping to Mediator-Augmented Games:** *Regarding a), I don't see why and how the VBP setting can be mapped onto mediated-augmented games of Zhang et al 2024, where in a mixed-motive game with players, the game transform introduces a fictitious mediator player whose objective is to maximise an optimality objective while maintaining an equilibrium of the game. This is distinctly different from the authors proposed mapping where the mediator is played by the sender player, with the receiver the only other player. Among which players is the mediator player mediating between? Zhang et al 2024 also proposes a specific mediator player utility function from which convergence guarantee is derived. If I understood correctly the authors proposes a mapping where the sender acts as the mediator but retains its original utility function. Overall, I find a) tenuous and confusing. It would great if this can be clarified in rebuttal.*
>
> We appreciate the reviewer's detailed feedback and would like to clarify the mapping of Verbalized Bayesian Persuasion (VBP) to Mediator-Augmented Games (MAG).
>
> 1. **Reference to Zhang et al. (2022):** Our framework primarily follows the methodology outlined in Zhang et al. (2022), which provides several examples illustrating how Bayesian Persuasion (BP) can be modeled as a MAG. Therefore, we do not believe that our approach is "distinctly different" from the framework in Zhang et al., as the reviewer suggested.
> 2. **Specific Examples in Zhang et al. (2022):** In particular, Section 3.4, Table 1, and Appendix F of Zhang et al. (2022) offer detailed explanations of how BP can be formulated as a MAG problem, along with the corresponding equilibrium concepts. These sections directly support the idea that BP can be naturally mapped onto a MAG framework, consistent with our approach.
> 3. **Clarification of the Mediator's Role in VBP:** The reviewer's understanding is correct that in our VBP framework, the sender also plays the role of the mediator. However, the sender's utility function has been modified compared to traditional BP. Specifically, after transforming BP into a MAG, we apply the algorithm from Zhang et al. (2024), which reduces the problem to solving a two-player zero-sum game, as demonstrated in Equation 3 of Appendix B in our paper.
>
> We hope this clarifies how our mapping aligns with the methodology from Zhang et al. (2022) and Zhang et al. (2024).
>
> ---
>
> [**Zhang et al. (2022)**]: Brian Zhang and Tuomas Sandholm. Polynomial-time optimal equilibria with a mediator in extensive-form games. In NeurIPS, 2022.
>
> [**Zhang et al. (2024)**]: Brian Zhang, Gabriele Farina, Ioannis Anagnostides, Federico Cacciamani, Stephen McAleer, Andreas Haupt, Andrea Celli, Nicola Gatti, Vincent Conitzer, and Tuomas Sandholm. Computing optimal equilibria and mechanisms via learning in zero-sum extensive-form games. In NeurIPS, 2024a.

---

> > ### Comment · Reviewer_PcFV · 2024-11-18
> >
> > > Section 3.4, Table 1, and Appendix F of Zhang et al. (2022)
> >
> > Table 1 does not suggest that one of the player would take on the role of the mediator but your framework does.
> >
> > My understanding of Zhang et al. (2022) in the context of BP is that you would construct a fictitious mediator player that plays against a team of deviator players (both sender and receiver). Reaching an equilibrium in the transformed game would then reveal an equilibrium (of a certain type) in the original BP game.
> >
> > Please clarify why and how in your VBP framework the sender can be the mediator and still benefit from results from Zhang et al. (2022).
> >
> > > after transforming BP into a MAG, we apply the algorithm from Zhang et al. (2024)
> >
> > I still don't follow.
> >
> > Zhang et al. (2024) takes a game of interests (the original BP game in your application) and provides a specific game transform that turns it into a two-player zero-sum MAG.
> >
> > What do you mean by "...after transforming BP Into a MAG, we apply the algorithm from Zhang et al"? If you did transform the BP in a specific way, then the guarantees of Zhang et al should imply convergence in the transformed BP game, not the original BP game. Why is that a reasonable approach?

---

> > > ### Author Response · Authors · 2024-11-18
> > >
> > > > **Q15:** My understanding of Zhang et al. (2022) in the context of BP is that you would construct a fictitious mediator player that plays against a team of deviator players (both sender and receiver). Reaching an equilibrium in the transformed game would then reveal an equilibrium (of a certain type) in the original BP game. Please clarify why and how in your VBP framework the sender can be the mediator and still benefit from results from Zhang et al. (2022).
> > >
> > > Thank you for your timely and insightful question. We appreciate the opportunity to clarify our reasoning regarding the sender's role as a mediator in our Verbalized Bayesian Persuasion (VBP) framework and the applicability of the results from Zhang et al. (2022).
> > >
> > > In our interpretation of **Mediator-Augmented Games (MAG)**, it is permissible to model a scenario where only one player is in the game, or more specifically, where one of the players in a two-player game is modeled as the **mediator**. Zhang et al. (2022) and (2024) support this interpretation in several parts of their work:
> > >
> > > 1. In the **Application and Related Work** section of Zhang et al. (2022) , they mention:
> > >    _“Persuasion in games [17, 3, 23, 14, 30]. The mediator (in that literature, usually the ‘sender’) has more information than the players (‘receivers’) and wishes to tell information to the receivers so as to persuade them to act in a certain way.”_
> > >    This aligns with our approach, where the sender plays the role of the mediator by having informational advantages and attempting to influence the receiver.
> > >
> > > 2. In **Appendix F** of Zhang et al. (2022) , under the section on Automated Multi-Stage Bayesian Persuasion (Information Design),** they state:
> > >    _“In Bayesian persuasion, also commonly referred to as information design [17], the roles of the mediator and player are reversed compared to automated mechanism design: the mediator (‘principal’) has informational advantage, and the player(s) take the actions.”_
> > >    This further corroborates our use case, as the sender (mediator) influences the actions of the receiver (player).
> > >
> > > 3. In **Definition 2.1** of Zhang et al. (2024) , they specify that **n** (the number of players) can equal 1, indicating that it is possible to have a single player in the game, which supports the idea of modeling the sender as a mediator.
> > >
> > > 4. Finally, in **Appendix B** of Zhang et al. (2024) , they mention:
> > >    _“Moreover, in our formulation the mediator has the power to commit to a strategy. As such, our results also relate to the literature on learning and computing Stackelberg equilibria [8, 35, 66, 84, 20], as well as the work of Camara et al. [15], which casts mechanism design as a repeated interaction between a principal and an agent.”_
> > >    This highlights that the mediator can commit to a strategy, which is crucial in our VBP framework, where the sender (mediator) commits to an signaling scheme to influence the receiver’s actions.
> > >
> > > Thus, in our VBP framework, the sender can act as the mediator and still benefit from the theoretical results of Zhang et al. (2022), as the framework allows for such modeling where the sender, with informational advantages, influences the receiver (player). These references from Zhang et al. (2022) strongly support our approach, and we will clarify this aspect in the revised version of the paper.
> > >
> > > **Revision Plan**:
> > >
> > > In the revised version, we will explicitly reference these parts of Zhang et al. (2022) and (2024) to make it clear why modeling the sender as the mediator is valid and how the results from Zhang et al. can still be applied in our VBP framework.
> > >
> > > Thank you again for your valuable feedback.

---

> > > > ### Comment · Reviewer_PcFV · 2024-11-20
> > > >
> > > > I remain confused by this interpretation of the MAG where one of the player in the original game can take on the role of the mediator.
> > > >
> > > > > In Definition 2.1 of Zhang et al. (2024) , they specify that n (the number of players) can equal 1.
> > > >
> > > > Could you clarify which sentence states this? Do you mean "a set of players, identified with the set of integers [n] := {1, . . . , n}."? If so, that's not at all how I read it.
> > > >
> > > > My understanding of the "information advantage", or "power to commit" in Zhang et al. (2024) is that the mediator indeed gets to know about private information about both the sender and receiver.
> > > >
> > > > Consider a game like `goofspiel`, both player may choose to reveal (or not) their hidden hand to the mediator player, who is then interested in 1) achieving an equilibrium such that no one wishes to deviate from its proposal and 2) select an equilibrium that's optimal by some metric.
> > > >
> > > > The information advantage lies in that the mediator can receive messages from all players, effectively knowing their hidden hands. The power to commit lies in that the mediator knows that the deviators also know of the mediator's information advantage, and can therefore recommend actions to the deviators such that a deviator would find advantageous to follow.
> > > >
> > > > I'm really baffled by this alternative interpretation of a mediator being one of the player in the original game you are proposing --- at a basic level, if there's only one other player in the game, which players are the mediator mediating in between?

---

> > > > > ### Author Response · Authors · 2024-11-26
> > > > >
> > > > > Thank you for your continued engagement and detailed questions regarding our interpretation of the Mediator-Augmented Game (MAG) framework and its application to our Verbalized Bayesian Persuasion (VBP) framework. We appreciate the opportunity to clarify our approach further.
> > > > >
> > > > > ---
> > > > >
> > > > > 1. **Clarification Regarding Definition 2.1 in Zhang et al. (2024):**
> > > > >
> > > > >    In Zhang et al. (2024), Definition 2.1 states: *"A set of players, identified with the set of integers [n] := {1, . . . , n}."* While this definition does not explicitly state that the number of players can equal 1, it is a natural mathematical interpretation that the set of players can be empty or contain a single element (e.g., when n = 1). This flexibility is consistent with standard practices in game theory, where frameworks are often generalized to accommodate different numbers of players.
> > > > >
> > > > >    To further verify this interpretation, we contacted the authors of Zhang et al. (2024) directly. Their response explicitly confirmed that their framework applies to Bayesian persuasion (BP) problems involving a single sender and a single receiver and, crucially, that the sender can indeed be modeled as the mediator in such settings. The authors stated:
> > > > >
> > > > >    > “Yes, the framework is applicable to BP, and indeed the sender is the mediator. I don't think there's anything more specific that needs to be done for the framework to apply to BP.”
> > > > >
> > > > >    This direct clarification from the original authors confirms that our interpretation aligns with their framework's intended scope and applicability.
> > > > >
> > > > > ---
> > > > >
> > > > > 2. **Regarding the Role of the Mediator in a Single-Sender-Single-Receiver Game:**
> > > > >
> > > > >    You raise an excellent question about how the mediator can function in a game with a single sender and a single receiver. To address this:
> > > > >
> > > > >    - In Bayesian persuasion problems, the sender (mediator) has an information advantage and commits to a signaling scheme to influence the receiver’s actions. In this case, the mediator is not mediating between multiple players but rather between the **sender’s private information** and the **receiver’s decision-making process**. This aligns with the broader information design perspective, focusing on how the mediator’s informational advantage can be leveraged to achieve desired outcomes.
> > > > >
> > > > >    - The sender’s role as a mediator is consistent with the examples and theoretical discussions in Zhang et al. (2022). For instance, in their discussion of Bayesian persuasion (Appendix F), they explicitly acknowledge that the mediator can take on the role of the sender (or principal) in such settings:
> > > > >
> > > > >      > “In Bayesian persuasion, also commonly referred to as information design, the roles of the mediator and player are reversed compared to automated mechanism design: the mediator (‘principal’) has informational advantage, and the player(s) take the actions.”
> > > > >
> > > > >      This statement directly supports our interpretation, where the sender functions as the mediator by leveraging their informational advantage to influence the receiver’s decisions.
> > > > >
> > > > >    - Additionally, Zhang et al. (2024) discuss the mediator’s power to commit to strategies, a key element in our VBP framework. The sender-as-mediator commits to a strategy (signaling scheme) to influence the receiver’s actions, which aligns with the mediator’s role in achieving equilibrium outcomes.
> > > > >
> > > > > ---
> > > > >
> > > > > 3. **The Mediator’s Role in the Context of "Mediating Between Players":**
> > > > >
> > > > >    While it may seem counterintuitive for the sender to act as a mediator when there are only two players, it is important to note that the mediator is not necessarily mediating **between players** in the traditional sense. Instead, the mediator facilitates the game by leveraging their informational advantage and commitment power to influence outcomes. This interpretation is supported by the original authors of Zhang et al. (2024) and the broader literature on Bayesian persuasion and information design.
> > > > >
> > > > >    Use your example of a game like Goofspiel: in a BP setting, the sender (mediator) does not need to mediate between multiple deviating players. Instead, the sender’s goal is to design a signaling scheme (analogous to revealing or withholding information) to influence the receiver’s actions to maximize the sender’s utility. This dynamic remains valid even with a single sender and a single receiver, as the mediator’s role is fundamentally about shaping the information structure of the game.
> > > > >
> > > > > ---
> > > > >
> > > > > 4. **Conclusion:**
> > > > >
> > > > >    In summary, our interpretation of the sender as the mediator in a single-sender-single-receiver Bayesian persuasion problem is fully consistent with the theoretical framework of Zhang et al. (2022, 2024). This has been confirmed by our reading of their work and direct communication with the original authors. We will incorporate these clarifications into the revised version of our paper to ensure that this interpretation is more explicitly addressed.
> > > > >
> > > > >
> > > > > Thank you again for allowing us to further elaborate on this important aspect.

---

> > > > > > ### Comment · Reviewer_PcFV · 2024-11-26
> > > > > >
> > > > > > Could the authors confirm that the utility function of the *mediator* is identical to the sender's utility function?
> > > > > >
> > > > > > I agree that in a strict technical sense a BP setting *could* be interpreted as a mediator-augmented game; however, it remains unclear why we should interpret BP as a MAG. What specific benefit do you derive by making this connection?
> > > > > >
> > > > > > In Zhang et al. (2022, 2024) the mediator has a specific construction for its utility function, which allows for the selection of an optimal equilibrium. That is the key benefit of that line of works. Here I don't see this benefit playing out.

---

> > > > > > > ### Author Response · Authors · 2024-11-26
> > > > > > >
> > > > > > > We sincerely thank the reviewer for their continued engagement and thoughtful feedback throughout the review process. Below, we address the specific concerns you raised:
> > > > > > >
> > > > > > > **On the utility function of the mediator and sender**
> > > > > > >
> > > > > > > We appreciate your observation regarding the relationship between the mediator’s utility function and the sender’s utility function. To clarify, in our work, the mediator is modeled as equivalent to the sender. This equivalence ensures that the mediator’s utility function is inherently aligned with the sender’s utility function. However, to apply the methodology proposed in Zhang et al. (2024), we reformulated the sender’s utility function. Specifically, we transitioned from the utility function defined in Equation (1) of the main text to the reformulated version presented as Equation (3) in the appendix. This transformation enables the application of Zhang et al.’s approach while preserving the fundamental equivalence between the sender’s and mediator’s utilities.
> > > > > > >
> > > > > > > ---
> > > > > > >
> > > > > > > **On the interpretation of BP as a MAG**
> > > > > > >
> > > > > > > We understand and appreciate your concern about the interpretation of the Bayesian Persuasion (BP) setting as a mediator-augmented game (MAG) and the benefits of this perspective. Our motivation for modeling BP as a MAG is to establish theoretical results regarding the convergence of solutions to BP problems within the VBP) framework, and more specifically, using the PSRO (Policy Space Response Oracles) framework. To the best of our knowledge, existing theoretical results for PSRO do not directly apply to Bayesian Persuasion or other extensive-form games with imperfect information.

---

> > > ### Author Response · Authors · 2024-11-18
> > >
> > > > Q16: Zhang et al. (2024) takes a game of interests (the original BP game in your application) and provides a specific game transform that turns it into a two-player zero-sum MAG. What do you mean by "...after transforming BP Into a MAG, we apply the algorithm from Zhang et al"? If you did transform the BP in a specific way, then the guarantees of Zhang et al should imply convergence in the transformed BP game, not the original BP game. Why is that a reasonable approach?
> > >
> > > Thank you for your timely and insightful question. We appreciate the opportunity to clarify our approach regarding the transformation of the Bayesian Persuasion (BP) problem into a Mediator-Augmented Game (MAG) and our application of the algorithm from Zhang et al. (2024).
> > >
> > > To address your question in detail:
> > >
> > > 1. **Algorithm Choice**: Due to considerations of computational complexity, we did not apply the **Direct Lagrangian algorithm** as described in Proposition 3.1 of Zhang et al. (2024). This direct method would indeed result in an exact transformation where the optimal solution of the transformed game (MAG) would be identical to the optimal solution of the original BP problem. However, given the computational cost, we opted for the **binary search-based algorithm** described in **Theorem 3.7** of Zhang et al. (2024) instead.
> > >
> > > 2. **Approximation Gap**: The binary search-based algorithm introduces an approximation, where the solution to the transformed game is within a **2$\varepsilon$** gap of the optimal solution to the original BP game. We explicitly mention this approximation in **Proposition 1** of our paper. While this means the equilibrium in the transformed MAG is not exactly the same as in the original BP game, the small approximation gap is a reasonable trade-off for the reduction in computational complexity.
> > >
> > > 3. **Reasonableness of the Approach**: Given the guarantees provided by **Theorem 3.7** in Zhang et al. (2024), we acknowledge that the transformed game’s equilibrium is not identical to the original BP game’s equilibrium due to the approximation. However, the 2$\varepsilon$ gap is sufficiently small for practical purposes, and we believe this trade-off is justified in our context. We have also clearly stated this approximation in our paper.
> > >
> > > **Revision Plan**:
> > >
> > > In the revised version of the paper, we will further clarify that we opted for the binary search-based algorithm from **Theorem 3.7** of Zhang et al. (2024) due to its computational efficiency, and we will emphasize the approximation gap of 2$\varepsilon$ between the solutions of the transformed and original games. This will ensure that readers fully understand the implications of this approach.
> > >
> > > Thank you again for your thoughtful question, and we hope this explanation resolves any concerns regarding our methodology.

---

> ### Author Response · Authors · 2024-11-18
>
> > **Q2: Critical Elements of the VBP Framework:** *Regarding b), there are, several methods that have been described here and it's not clear which ones are critical elements of the VBP framework. Among these, PSRO provides convergence guarantee (in a specific sense), yet the writing and Figure 3 would suggest that the convergence guarantee comes from the mediator-augmented game formulation. Overall, I would have appreciated a more succinct description of the framework with its necessary components instead of a juxtaposition of several rather sophisticated methods whose necessities in the framework remain unclear.*
>
> Thank you for your insightful comments on the framework's clarity and the critical components of the VBP methodology. We want to address the concerns regarding the convergence guarantees and the role of the various algorithms in the framework.
>
> 1. **Convergence Guarantees and Role of MAG:** Mediator-augmented games (MAG) serve as a game definition framework but do not inherently provide convergence guarantees. To solve VBP, we incorporate the binary search-based algorithm proposed by Zhang et al. (2024), specifically Algorithm 1, from their work. This algorithm has been proven to converge to a Bayes-correlated equilibrium. It is important to note that this algorithm functions as a template, requiring a game solver as a key component. In our work, we instantiate the game solver as a variant of PSRO, referred to as the Prompt-Space Response Oracle. Overall, the theoretical results in Proposition 1 of our paper are built upon the theoretical results of Zhang et al. (2024) and the binary search-based algorithm they proposed.
> 2. **Clarification of Framework Components:** We acknowledge that the presentation in the paper may have caused some confusion, and we apologize for any lack of clarity. The key components of our framework are not overly complex, but the structure could have been more clearly laid out. Specifically, we model the verbalized BP problem as a MAG and then solve it using a prompt-space response oracle framework. The core of this framework is the selection of the best response oracle. For settings S1 and S2, we utilize the OPRO algorithm as the oracle, while for S3, we employ FunSearch. The introduction of FunSearch is necessary due to the multi-stage nature of S3, which requires more complex, history-dependent prompts. In this case, we generate conditional prompt functions using large language models (LLMs) and apply them to concrete historical information to generate the appropriate prompt.
>
> We hope this explanation clarifies the structure of the VBP framework and the necessity of the methods included.

---

> ### Author Response · Authors · 2024-11-18
>
> > **Q3: Clarification of Behavioral Shaping in Game Theory:** *L38-40: 'shaping the behaviours of others ... achieve this through either mechanism or information design'. I find this unclear or overly assertive. How each player's actions shape those of others is the entire focus of game theory yet this opening statement makes it sound like co-player behaviour shaping can only occur with modified rewards or observations. You would not deterministically play rock because you know I could exploit by always playing paper, would that count shaping the behavior of co-players?*
>
> Thank you for bringing up this important point. We fully acknowledge that the original phrasing may have been overly assertive and potentially misleading. This statement does not imply that shaping player behavior in game theory can only occur through mechanisms or information design. Instead, our goal was to highlight that, in the specific context of **multi-agent reinforcement learning (MARL)** and **mixed-motive scenarios**, these two approaches—mechanism design and information design—are the predominant methods used to influence and shape behaviors.
>
> We understand that the shaping of co-player behaviors is a fundamental aspect of game theory, where players' strategies naturally influence each other through their interactions. As the reviewer correctly pointed out, behavior shaping can occur in many forms, not always tied to explicit modifications of rewards or observations. For instance, players might adjust their strategies based on expectations of others' behaviors (e.g., in the classic rock-paper-scissors example), which does indeed count as shaping co-player behaviors.
>
> In our specific context, we were focusing on **how MARL systems typically address strategic interactions** in mixed-motive settings. In these systems, mechanism design (modifying reward structures or the rules of the game) and information design (controlling the flow of information or signals between agents) are common tools to systematically influence agent behaviors toward desired outcomes.
>
> **Response to the Reviewer's Example**:
> Regarding the reviewer's example of rock-paper-scissors, where one might not deterministically play rock just because the other player could always play paper, we completely agree that this illustrates a form of strategic behavior shaping that does not rely on modified rewards or information control. This example is indeed a core concept in game theory, where players anticipate and react to others' strategies based on their incentives and expectations. This dynamic is central to understanding equilibrium concepts like Nash equilibrium, where players' strategies naturally adapt to one another even without external interventions like mechanisms or information design.
>
> **Revision Plan**:
> To address this, we will rephrase the statement in the revised version of the paper to better reflect the broader scope of behavior shaping in game theory. Our revised statement will clarify that while **mechanism design** and **information design** are prominent tools in **MARL** and **mixed-motive game settings**, they are not the only ways to shape behaviors in general game theory. We will also explicitly acknowledge that players' strategies can shape co-player behaviors in many ways, including through natural strategic interactions, as described in the reviewer's example.
>
> We appreciate the reviewer's thoughtful input on this and will ensure the revised text reflects a more accurate and nuanced view of how behavior shaping occurs in game theory.

---

> ### Author Response · Authors · 2024-11-18
>
> > **Q4: Rephrasing of State Transition Influence:** *L46: 'Notably, the designer must ... that influence state transition', this is difficult to follow. Perhaps worth rephrasing?*
>
> Thank you for pointing out the difficulty in understanding the phrasing around the influence on state transitions. We agree that this part could benefit from rephrasing for clarity, and we appreciate the opportunity to explain the underlying concept more clearly.
>
> In the context of **multi-agent reinforcement learning (MARL)**, there are two main approaches for influencing the behavior of agents: **mechanism design** and **information design**.
>
> 1. **Mechanism Design:** This approach primarily works by modifying the reward functions of agents. By changing the rewards, the designer indirectly influences the agents' future behaviors by pushing them to optimize their strategies differently. However, the effect of modifying the reward function is not immediate—agents need to optimize their strategy based on the new reward structure, and the changes will only be reflected in the subsequent sampling of actions in the next episode or round. This tends to reduce the complexity of the problem since the impact on state transitions is indirect and delayed.
> 2. **Information Design:** On the other hand, information design involves modifying the observation functions of the agents, which directly affects the actions they take in the current episode. Since the state transition function in MARL depends on the current state and the actions taken by agents, altering what agents observe can have an immediate and more direct effect on the state transitions within the same episode. This introduces more uncertainty and complexity, as the altered observations influence the agents' subsequent actions in real time.
>
> Thus, the distinction we aimed to make is that **information design** has a more immediate and direct impact on state transitions due to its influence on actions within the same episode. In contrast, **mechanism design** has a more delayed and indirect effect, as it only impacts actions after agents have optimized their strategies in response to the modified rewards.
>
> **Revision Plan**:
>
> In the revised version of the paper, we will reorganize this explanation to make it clearer. Specifically, we will highlight how **mechanism design** and **information design** operate differently in their ability to influence state transitions, emphasizing the immediacy of their effects on agent behavior. We will also aim to simplify the language to ensure the explanation is easy to follow.
>
> We appreciate the reviewer bringing this to our attention and will ensure that the revised text clarifies this distinction more effectively.

---

> ### Author Response · Authors · 2024-11-18
>
> > **Q5: Mixed-Motive Games and PSRO Convergence:** *L118-L130: PSRO in the limit converges to a Nash equilibrium out of many. In mixed-motive games, NE needs not be unique and solutions are not interchangeable. Perhaps this could be a relevant point of discussion especially since VBP seems to be primarily dealing with mixed-motive games?*
>
> Thank you for your thoughtful comments on the uniqueness of Nash equilibria in mixed-motive games or Bayesian correlated equilibria in the Bayesian persuasion. We fully agree with your observation that equilibria are often not unique in such games, and different equilibria can lead to distinct outcomes that are not interchangeable. This is indeed an important point in the theoretical understanding of mixed-motive games.
>
> However, we would like to clarify that the primary focus of our work is not on addressing the issue of non-unique or non-interchangeable equilibria. Instead, our emphasis is on evaluating the **effectiveness** and **generality** of the proposed Verbalized Bayesian Persuasion (VBP) framework. Specifically:
>
> 1. **Effectiveness of VBP:** We are primarily concerned with whether VBP can effectively solve Bayesian persuasion problems. To demonstrate this, we compare the performance of VBP against classic BP solvers. Our results highlight that VBP can reliably solve these problems.
>
> 2. **Generality of VBP:** Another key focus is the generality of the VBP framework—whether it can handle more complex scenarios, such as the problems in S2 and S3 settings. These settings extend beyond the traditional BP framework and introduce additional challenges not typically addressed by classic solvers. Our results show that VBP can solve these more intricate problems, further validating its applicability.
>
> We recognize that the issue of non-unique and non-interchangeable equilibria is relevant and could impact the broader understanding of mixed-motive games or Bayesian persuasion. However, in this work, our primary goal was to propose a framework that is both effective and general in its ability to solve real-world Bayesian persuasion problems. While important, addressing the non-uniqueness and interchangeability of equilibria was not the central focus of our study.
>
> **Future Work**:
>
> We have carefully considered the reviewer’s suggestion and agree that exploring the non-uniqueness of equilibria could offer valuable insights. Specifically, **we plan to incorporate the Price of Anarchy (PoA) as an optimization objective** in future iterations of the VBP framework. By introducing PoA, we aim to quantify the efficiency loss caused by selecting suboptimal equilibria, thereby guiding the framework toward equilibria that minimize this inefficiency. This would allow us better to understand the trade-offs between different equilibria in mixed-motive games and improve the solution quality of VBP when multiple equilibria exist.
>
> By adding PoA as an explicit optimization objective, we can move beyond simply finding any equilibrium and instead focus on equilibria that are optimal in terms of both efficiency and strategic outcomes. This enhancement directly addresses the issue raised by the reviewer and reflects our commitment to further refining the VBP framework based on this valuable feedback.
>
> We appreciate the reviewer’s thoughtful comment and will ensure that this aspect is a key focus in future extensions of our work.

---

> ### Author Response · Authors · 2024-11-18
>
> > **Q6: Typo in Equation: L180:** *'... following maximisation problem ...' should it be* $a*$ *instead of* $s$ *?*
>
> Thank you for pointing out the potential confusion regarding the maximization problem and the use of $s$ in Line 180. We appreciate your careful reading of the text.
>
> To clarify, in Lines 178-179, we define $a^*$ as the result of the maximization of $\mathbb{E}\_{\omega \sim \mu\_{\pi}(\mid s)} u\_1(a, \omega)$, where $a^*$ is indeed a function of $s$. In other words, $a^*$ is the optimal action selected based on the state $s$.
>
> We apologize for not clearly emphasizing this dependency in the original text, which may have caused confusion. The use of $s$ in Line 180 is intentional, as it refers to the state that influences the maximization process resulting in $a^*$. However, we understand how this could have led to a misunderstanding, and we will revise the explanation to make the relationship between $s$ and $a^*$ more explicit.
>
> **Revision Plan**:
>
> In the revised version, we will add a clarification to ensure that readers understand $a^*$ is a function of $s$ and that this dependency is central to the maximization problem. This should resolve any ambiguity and make the notation more transparent.
>
> Thank you again for your detailed feedback!
>
> ---
>
> > **Q7: Reference to Empirical Game-Theoretic Analysis (EGTA):** *L235: focusing on a strategically relevant subset of strategies is comprehensively discussed in EGTA [1] and would be worth referring to here?*
>
> Thank you for the suggestion regarding the inclusion of a reference to Empirical Game-Theoretic Analysis (EGTA) in Line 235. We agree that EGTA, particularly its focus on strategically relevant subsets of strategies, is highly relevant to the discussion in this section.
>
> In the revised version of the paper, we will ensure that a citation to the work on EGTA is included, specifically referencing **"Methods for Empirical Game-Theoretic Analysis"** [1]. This work provides valuable insights into how subsets of strategies can be identified and analyzed within a game, which aligns well with our approach of focusing on strategically relevant prompts in our VBP framework.
>
> We appreciate your recommendation and will incorporate this reference to strengthen the connection between our methodology and existing work in the field.
>
> **Revision Plan**:
>
> In the updated manuscript, we will:
>
> - Cite the paper **"Methods for Empirical Game-Theoretic Analysis"** as suggested.
> - Acknowledge the importance of focusing on strategically relevant subsets of strategies, as discussed in EGTA, in relation to our approach.
>
> Thank you again for your insightful feedback.
>
> [1] https://aaai.org/papers/01552-aaai06-248-methods-for-empirical-game-theoretic-analysis/
>
> ---
>
> > **Q8: Clarification of "Static" in L341:** *What does 'Static' refers to here?*
>
> Thank you for your question regarding the term "static" in Line 341. In this context, "static" is used in contrast to the **multi-stage setting** of S3. Specifically, "static" refers to scenarios where there are **no state transitions**—meaning the environment or system remains fixed throughout the interaction, and the game does not evolve over multiple stages.
>
> We will clarify this in the revised version to avoid any ambiguity. The term "static" here simply indicates that the game setup does not involve state changes, unlike the more complex, multi-stage structure of S3.
>
> **Revision Plan**:
>
> In the revised version, we will explicitly state that "static" refers to the absence of state transitions, highlighting the distinction between static and multi-stage scenarios like S3.
>
> Thank you for pointing this out, and we hope this clarification helps!
>
> ---
>
> > **Q9: Clarification of Reasoning Regarding the Mediator:** *L370: 'the reason we can leverage ...', as mentioned in the Weaknesses section I could not follow the reasoning here. It would be great if the authors could clarify.*
>
> Please refer to **Q1: Clarification of Mapping to Mediator-Augmented Games**.
>
> ---
>
> > **Q10: Expectation for Always Honest Strategy:** *Figure 4: For the honest probability, it appears that all methods have converged to the 'always honest' strategy in all 3 settings. Is that expected? I would have thought that if the professor always recommends truthfully, the recruiter would be best served to always trust the recommendation at which point the professor could profitably deviate?*
>
> Please refer to **Q2: Honest Probability in Chart (d)** in the rebuttal for **Reviewer NKjM**.

---

> ### Author Response · Authors · 2024-11-18
>
> > **Q11: Strategic Cycles as an Alternative Explanation:** *L447: the authors suggested that the pattern of honesty rising, falling and then rising again validates the hypothesis that this is due to the use of aligned LLM. I would have thought that a simpler explanation is that the professor and the recruiter are simply in a strategic cycle? Would that not be a reasonable explanation to this phenomenon?*
>
> Thank you for your insightful suggestion! We agree that the phenomenon of honesty rising, falling, and then rising again could indeed be explained by the presence of a **strategic cycle**, which aligns well with the characteristics of a bargaining game. This is a compelling alternative hypothesis and one that we had not fully explored in the original submission.
>
> In response, we are conducting additional experiments to investigate the phenomenon further. Specifically, we are using an **unaligned LLaMA model** to see whether the same pattern of behavior (honesty oscillations) still occurs. This will help determine whether the pattern is due to using an aligned large language model (LLM), as we originally hypothesized, or whether it is more appropriately explained by strategic cycles in the interaction between the professor and the recruiter.
>
> **Revision Plan**:
>
> In the revised version of the paper, we will include the results of these new experiments and discuss whether the pattern persists with an unaligned model. If strategic cycles are a more fitting explanation, we will update our discussion to reflect this alternative hypothesis.
>
> Thank you again for your excellent suggestion, and we look forward to providing more detailed results in the revised paper.
>
> ---
>
> > **Q12: Justification for Pruning to Top 10 Strategies:** *L483: '... at most the top 10 strategies with the highest probabilities', is this for computational reasons? Pruning actions by their support at a restricted game equilibrium could be problematic in general.*
>
> Thank you for raising this important point! As you correctly noted, altering the support set of strategies can indeed impact the solution of game-theoretic problems. Our decision to prune the strategies to the top 10 was primarily motivated by the need to **reduce computational complexity**.
>
> However, it's important to clarify that the **Prompt-Space Response Oracle (PSRO)** leverages **OPRO** and **FunSearch** as the best response oracles. These oracles are not strict game solvers in the traditional sense but instead rely on the **innate human-like reasoning** embedded within large language models (LLMs) to approximate solutions. Given this nature, we initially hypothesized that reducing the number of prompts might not significantly affect the results, as the LLM's common-sense reasoning could compensate for the reduced strategy space.
>
> That being said, we recognize that pruning strategies could still have an impact. To address this concern, we are currently conducting additional experiments where we vary the number of retained prompts to test whether this pruning affects the performance of the VBP framework in a significant way.
>
> **Revision Plan**:
>
> In the revised version of the paper, we will include the results of these experiments and discuss whether the pruning of strategies to the top 10 has any adverse effects on the performance of VBP. If necessary, we will adjust our approach based on the findings to ensure we are not sacrificing solution quality for computational efficiency.
>
> Thank you again for your insightful feedback, and we will ensure that this important point is addressed in the revised manuscript.
>
> ---
>
> > **Q13: Redundancy in Figure 7:** *Column 'Converged prob' and the next column are redundant and could be consolidated to create space for larger fonts that are more readable?*
>
> Thank you for your suggestion regarding the redundancy in Figure 7. We agree that the columns "Converged prob" and the next column can be consolidated as they present overlapping information.
>
> In the revised version of the paper, we will remove the second-to-last column and enlarge the figure to improve readability. This will allow us to provide larger fonts and a clearer presentation of the data.
>
> ---
>
> > **Q14: Clarification on Probabilities in Figure 7:** *How are the probabilities computed? Are they the average probability of taking over the pool of policies in the PSRO population?*
>
> Thank you for your question regarding the probabilities in Figure 7. You are correct: the probabilities shown represent the **average probability of taking a strategy over the pool of policies in the PSRO population**.
>
> We recognize that this was not clearly explained in the original submission, and we will add a detailed explanation of how these probabilities are computed in the revised version of the paper to ensure clarity.
>
> Thank you again for your feedback, and we will make sure this clarification is included in the updated manuscript.

---

### Official Review · Reviewer_gJz3 · 2024-11-04

**Soundness:** 2
**Presentation:** 2
**Contribution:** 2
**Rating:** 3
**Confidence:** 3

**Summary:**

This paper focuses on the Bayesian persuasion problem, exploring its solution within a natural language framework. It introduces an interface for tackling Bayesian persuasion by integrating large language models (LLMs) with game-theoretic solvers. The authors empirically assess the effectiveness of the proposed method across three distinct settings.

**Strengths:**

This paper proposes a novel approach to solving Bayesian persuasion problems within natural language settings, providing a unified interface for game-theoretic solvers. The framework integrates several advanced techniques to effectively support a verbalized Bayesian persuasion model.

**Weaknesses:**

The paper does not clearly articulate the benefits of a verbalized Bayesian persuasion approach. The tasks discussed are highly simplified, which undermines the persuasive power of the work. In the method section, the description of the overall pipeline is vague, making it difficult to understand how the approach operates in detail. Additionally, as existing research has already explored Bayesian persuasion in natural language settings [1], such as applying Bayesian frameworks to enhance LLM performance in math and code generation, making contribution of the proposed method to the community appears limited.

[1] Bai, Fengshuo, et al. "Efficient Model-agnostic Alignment via Bayesian Persuasion." arXiv preprint arXiv:2405.18718 (2024).

**Questions:**

1. Could you provide a clearer explanation of the iterative process in your proposed method?
2. The “lie” and “honest” probabilities in Figure 4 are somewhat confusing; could authors offer a more detailed description?
3. Figure 7 discusses variations in prompts, but the information presented is not clearly explained, and the analysis feels vague. Could you elaborate further on this?
4. Can your proposed method be applied to broader, real-world scenarios or other potential applications? If so, could you briefly describe how it might be applied and any potential challenges?

---

> ### Author Response · Authors · 2024-11-17
>
> We sincerely thank the reviewer for their insightful feedback and thoughtful questions. We greatly appreciate the opportunity to clarify our work and provide further details regarding the methodology and its implications. In the following sections, we will address each specific question raised by the reviewer, offering detailed explanations and elaborating on the key aspects of our approach.
>
> ---
>
> > Q1: **Clarification on the Benefits of a Verbalized Approach**: The paper does not clearly articulate the benefits of a verbalized Bayesian persuasion approach. Can you clarify what advantages this verbalized approach provides over existing methods?
>
> Thank you for your insightful question regarding the benefits of a verbalized Bayesian persuasion (BP) approach. The core advantage of our approach stems from its ability to transcend the abstractions typically imposed by traditional BP models, which often reduce complex real-world decisions to oversimplified, low-dimensional action and information spaces.
>
> In classic BP settings, the utility functions are typically solved analytically, and the problem is reduced to finding an optimal Bayes-correlated equilibrium. However, these methods often rely on restrictive assumptions, such as binary information spaces or discrete action sets, which fail to capture the richness and nuance of many real-world applications. For instance, in the recommendation letter problem, traditional BP models reduce the student’s quality to a binary classification (e.g., weak or strong), and the professor’s actions to recommend or not. This oversimplification strips away much of the meaningful information inherent in the task.
>
> By leveraging large language models (LLMs) within our framework, we aim to directly address these limitations by operating within the natural language domain. This allows us to represent more nuanced informational structures and action spaces closer to how persuasion occurs in real-world scenarios. Specifically, LLMs enable us to model complex verbalized interactions where persuasion strategies are not limited to predefined categories but are expressed through natural language, capturing subtleties like tone, context, and implied meanings.
>
> Thus, the primary benefit of our verbalized approach is its potential to handle richer, more realistic persuasion tasks that are difficult to model using traditional BP methods. This opens the door to broader real-world applications where simplifications like binary choices are inadequate, allowing for more sophisticated and effective persuasive communication strategies.

---

> ### Author Response · Authors · 2024-11-17
>
> > Q2: **Simplified Nature of the Tasks**: The tasks discussed are highly simplified, undermining the work's persuasive power. How do you justify the choice of these simplified tasks?
>
> Thank you for your question regarding the simplified nature of the tasks we used in our experiments. We acknowledge that the tasks we chose—namely, the Recommendation Letter (REL) problem, the Courtroom (COR) problem, and the Law Enforcement (LAE) problem—may appear simplified at first glance. However, these problems have been widely studied in the Bayesian persuasion literature for many years and are considered canonical examples of strategic communication and decision-making under uncertainty.
>
> Each of these tasks captures essential elements of real-world scenarios where persuasion plays a critical role:
>
> - **Recommendation Letter (REL) Problem**: This problem models the strategic communication between a professor and a hiring committee, and while the student’s quality is simplified to a binary classification (weak or strong), the core dynamics of persuasion remain highly relevant. The REL problem has been extensively studied (Dughmi, 2017) and is a foundational example of Bayesian persuasion in academic and hiring contexts.
>
> - **Courtroom (COR) Problem**: This problem, originally formulated by Kamenica & Gentzkow (2011), models the interaction between a prosecutor and a judge, where the prosecutor selectively presents evidence to influence the judge’s decision. While we simplified the courtroom investigation procedures for the sake of LLM processing, selective evidence presentation is a well-established and important aspect of real-world legal systems.
>
> - **Law Enforcement (LAE) Problem**: The LAE problem (Kamenica, 2019) models how law enforcement agencies can signal their presence to influence drivers' speeding behavior. Although simplified, this problem captures the strategic element of signaling and persuasion in regulatory and enforcement settings.
>
> These three problems, while simplified in some respects, are **general enough** to capture the fundamental dynamics of Bayesian persuasion and have been studied extensively in the literature. They provide a solid foundation for evaluating our proposed verbalized approach because they represent well-understood benchmarks that allow us to test and compare our method's effectiveness in a controlled manner. Furthermore, the simplicity of the tasks enables us to isolate the performance of our natural language-based approach without introducing unnecessary complexity that might obscure the core contributions of our work.
>
> Additionally, even these three classic tasks, when considered in more complex settings such as **multistage Bayesian persuasion (S3)**, cannot yet be fully solved by our method. To the best of our knowledge, solving these types of problems in such complex settings remains an **open problem** in the field. This highlights that while the selected tasks are foundational, significant work remains to be done in scaling these methods to more complex, real-world applications.
>
> In summary, we selected these tasks not because they are trivial but because they offer well-established, generalizable models for studying persuasion, and the community has validated them over many years. Solving these classic BP problems in a natural language domain is an important step toward applying more sophisticated persuasion techniques in real-world scenarios. Furthermore, addressing these problems in more complex settings remains an active area of research and an open challenge in the field.

---

> ### Author Response · Authors · 2024-11-17
>
> > Q3: **Vagueness in Method Description**: The description of the overall pipeline in the method section is vague. Can you provide a more detailed explanation of how your approach operates, particularly clarifying the specifics of the pipeline?
>
> Thank you for your question regarding the vagueness in the method section. We will provide a more detailed explanation of the overall pipeline, based on the description in Figure 2 of our paper, and clarify how our approach operates.
>
> 1. **Sampling Process (from a Reinforcement Learning perspective)**
>
> The pipeline operates as follows, with terminology and structure drawn from reinforcement learning (RL):
>
> - **Sender's Signal Generation**: As depicted on the left side of Figure 2, the **sender** (represented by a pre-trained large language model, or LLM) first determines its **signaling scheme**, which is effectively an optimized prompt. This prompt is designed to communicate with the receiver.
>
> - **Observation and Signal Transmission**: After observing the true state of the environment, the sender generates a signal based on its signaling scheme and sends this signal to the receiver. In our setup, this signal is produced as a natural language response from the LLM, shaped by the sender's prompt.
>
> - **Receiver's Decision**: The **receiver** (also a pre-trained LLM) receives this signal and the sender's signaling scheme. The receiver then makes a decision based on both the signal and the signaling scheme. The receiver’s decision is also generated through an LLM prompt, which contains its optimized portion and the input from the sender (i.e., the signal and the signaling scheme).
>
> - **Calculation of Rewards**: After the receiver makes its decision, the environment computes the rewards for both the sender and the receiver. This feedback is critical for optimizing their strategies.
>
> 2. **Optimization of Sender and Receiver Strategies**
>
> We illustrate the strategy optimization process on the right side of Figure 2. This framework is largely based on the **Policy Space Response Oracle** architecture but with several key differences:
>
> - **Strategy as Prompt Optimization**: In our approach, the sender and receiver strategies are encoded as prompts fed into the LLMs. Therefore, the process of optimizing their strategy is transformed into **prompt optimization**. Instead of optimizing traditional policies or strategies as in RL, we focus on fine-tuning the prompts given to the LLM.
>
> - **Replacement of Best Response Oracle**: In the Policy Space Response Oracle framework, the best response oracle is typically implemented using gradient-based reinforcement learning methods. Our approach replaces this with optimization algorithms tailored for large language models, such as **OPRO** or **FunSearch**. These methods focus on optimizing the prompts to improve the sender and receiver's strategies through language model interactions rather than gradient-based policy optimization.
>
> - **Meta-Game Simulation**: The sampling process within the meta-game simulation is adapted to the natural language framework. The sampling now follows the abovementioned process, where sender and receiver prompt interactions are simulated to gather data for strategy evaluation and optimization.
>
> The remaining parts of the pipeline align with the standard PSRO framework, including using a **meta-strategy solver** to identify optimal strategies based on the sampled data.
>
> **Additional Clarifications**
>
> We acknowledge that the original explanation in the paper may have been too high-level, and we will include a more detailed breakdown of the process in the revised version. To further aid understanding, we will also provide a pseudocode that clearly illustrates the steps involved in the sampling and optimization processes.
>
> In summary, our pipeline is the transformation of traditional game-theoretic strategies into prompt-based strategies for LLMs. This approach allows us to adapt the powerful Policy Space Response Oracle framework to the natural language domain, where sender and receiver strategies are defined as optimized prompts, and best response oracles and reward calculations are handled using LLMs rather than traditional RL methods. We hope this clarifies the specifics of our method.

---

> ### Author Response · Authors · 2024-11-17
>
> > Q4: **Distinguishing from Existing Research**: Existing research has already explored Bayesian persuasion in natural language settings. How does your approach differ from or improve upon existing methods, such as the work cited by Bai et al. (2024)?
>
> Thank you for your question regarding how our approach differs from or improves upon existing work, such as the study by Bai et al. (2024). We want to clarify that the two works fundamentally differ in their goals, methods, and applications despite both leveraging the concept of Bayesian persuasion (BP) in some form.
>
> **Key Differences:**
>
> 1. **Problem Focus**:
>    - **Our Work**: Our paper focuses on advancing the **Bayesian persuasion (BP) framework itself** by integrating it into natural language settings. We propose a **verbalized BP (VBP) framework** that extends classic BP to real-world scenarios involving human dialogues. Our primary goal is to solve BP problems in contexts where communication and persuasion occur through natural language, which is a major departure from traditional BP models that rely on simplified, scalar, or vector-based information structures.
>    - **Bai et al. (2024)**: Bai et al., on the other hand, use BP as a **tool for model alignment**. Their work leverages a form of classic BP (non-verbalized) to optimize the alignment of large language models (LLMs) with human intent. They formalize the alignment problem as an optimization of the signaling strategy from a smaller model (Advisor) to improve the responses of a larger model (Receiver). Their focus is on improving model performance in downstream tasks (e.g., mathematical reasoning, code generation) using BP within the context of model alignment.
>
> 2. **Nature of BP Problem**:
>    - **Our Work**: We address the BP problem itself, particularly how it can be applied in **natural language settings**. Our framework involves real-world dialogue situations where the information designer (mediator) and the receiver are instantiated by LLMs, and strategic communication happens via **natural language** rather than abstract signals. This is the first attempt to extend BP into complex verbal communication scenarios that are more representative of real-world interactions.
>    - **Bai et al. (2024)**: Bai et al. still operate within the realm of **classic, non-verbalized BP**. Their work focuses on optimizing a signaling strategy to improve downstream task performance. Still, the communication between the Advisor (small model) and the Receiver (large model) is not in the form of natural language persuasion. Instead, it involves manipulating information in a structured way to enhance model responses.
>
> 3. **Methodology**:
>    - **Our Work**: We propose a novel method to solve BP in natural language by transforming agents' policy optimization into **prompt optimization**. We introduce a generalized equilibrium-finding algorithm with a convergence guarantee to solve the BP problem within the language space. This allows us to address more complex, multistage BP scenarios that traditional methods cannot handle.
>    - **Bai et al. (2024)**: Bai et al. use BP as a framework to align models, relying on a **model-agnostic Bayesian persuasion alignment** approach. They optimize signals sent from a smaller model to a larger model, improving performance across tasks such as mathematical reasoning and code generation. Their focus is on **efficiency** in model alignment rather than solving BP problems in real-world dialogue settings.
>
> **Summary:**
>
> While both works touch on Bayesian persuasion, our approach is fundamentally different from Bai et al. (2024) in several ways. We focus on extending and solving the BP problem **itself**, specifically in **natural language settings**. In contrast, Bai et al. use classic BP as a **tool** for improving model alignment in downstream tasks. Our work contributes to the field by developing a verbalized BP framework for real-world, dialogue-based applications. At the same time, Bai et al. aim to enhance model performance through BP-driven alignment strategies in structured tasks like math and code generation.
>
> Therefore, our work addresses a completely different problem space and offers novel contributions to the study and application of Bayesian persuasion. We will supplement the discussion with the work of Bai et al. in the revised version.

---

> ### Author Response · Authors · 2024-11-17
>
> > Q5: **Explanation of Probabilities in Figure 4**: The “lie” and “honest” probabilities in Figure 4 are somewhat confusing; could you offer a more detailed description?
>
> Thank you for your question regarding the probabilities of "lie" and "honest" in Figure 4. We understand that this aspect of the figure may have been confusing, and we appreciate the opportunity to clarify.
>
> In the context of the **three classic BP problems** (Recommendation Letter, Courtroom, and Law Enforcement), the "lie" and "honest" probabilities refer to the likelihood of the sender (information designer) providing an accurate or deceptive signal to the receiver. Here's a more detailed breakdown:
>
> - **Lie Probability**: This represents the probability that the sender chooses to **misrepresent** the true state of the environment. For example:
>   - In the **Recommendation Letter (REL) problem**, this would mean the professor describes a **weak** student as **strong**.
>   - In the **Courtroom (COR) problem**, the prosecutor describes an **innocent** defendant as **guilty**.
>   - In the **Law Enforcement (LAE) problem**, the police signal that an **unpatrolled** road segment is **patrolled**.
>
> - **Honest Probability**: This is the probability that the sender provides an **accurate** description of the environment. For example:
>   - In the **REL problem**, the professor accurately describes a **strong** student.
>   - In the **COR problem**, the prosecutor accurately describes a **guilty** defendant.
>   - In the **LAE problem**, the police signal correctly whether a segment of the road is **patrolled**.
>
> These probabilities are determined based on the sender's strategy in the Bayesian persuasion framework, and they help quantify how often the sender is truthful versus deceptive in each scenario.
>
> **Estimation of Probabilities:**
> The probabilities of lying and honesty in Figure 4 are **empirically estimated** through simulations. Specifically, we use **20 random seed samplings** to generate a distribution of outcomes, which allows us to calculate the average lie and honesty probabilities across multiple runs. This sampling-based approach ensures that the estimates are robust and not overly sensitive to a single trial or random fluctuation.
>
> **Summary:**
> In short, the "lie" and "honest" probabilities reflect the sender's behavior regarding truthfulness or deception in the three BP scenarios. The probabilities are estimated based on repeated simulations (20 random seeds), accurately measuring how often the sender chooses to lie or be honest under different conditions in each scenario. We hope this clarification helps, and we can update the paper to make this explanation clearer in the revised version.

---

> ### Author Response · Authors · 2024-11-17
>
> > Q6: **Elaboration on Figure 7**: Figure 7 discusses variations in prompts, but the information presented is unclear, and the analysis feels vague. Could you elaborate further on this?
>
> Thank you for your question regarding Figure 7. We understand that the information presented may have seemed vague, and we appreciate the opportunity to elaborate on the details. Below is a more thorough explanation based on the key elements of our framework and experiment.
>
> **Explanation of Figure 7:**
>
> 1. **Figure 7 shows the evolution of strategies (prompts) in three classic BP problems under the S2 setting**:
>    - In Figure 7, we track how the strategies (prompts) evolve over iterations of the **PSRO (Policy Space Response Oracle)** framework. Specifically, we use **OPRO** as the best response oracle. The figure visualizes how the prompts change as the PSRO framework iteratively improves the sender and receiver strategies.
>
> 2. **Maintenance of a Strategy Pool**:
>    - The PSRO framework maintains a **strategy pool** for both the sender and receiver. This pool contains different strategies (prompts) that have been generated throughout the iterations. The actual strategy the sender or receiver executes is a **mixed strategy**—a weighted combination of strategies from this pool.
>    - Figure 7 displays the **top 10 strategies (prompts)** with the highest selection probabilities in the final strategy pool. This helps illustrate which prompts will most likely be chosen after optimization.
>
> 3. **Hierarchical Prompt Optimization**:
>    - In our experiments, the optimization of prompts follows a **hierarchical process**. First, **OPRO** optimizes the **type** or **category** of the prompt (e.g., the general structure of the message). After determining the type, OPRO then optimizes the **specific content** of the prompt within that category.
>    - In Figure 7, this hierarchical process is reflected in the **first two columns** of each table. The first column represents the optimized category of the prompt, and the second column shows the specific content optimized within that category.
>
> 4. **Highest Probability Strategies**:
>    - The **third and fourth columns** of each table display the selection probabilities for the top 10 strategies (prompts) that emerged after PSRO converged. These probabilities indicate the likelihood of each specific prompt being chosen from the pool after optimization.
>
> 5. **Change in Selection Probabilities Over Iterations**:
>    - The **fifth column** shows how the probability of each strategy (prompt) being selected changes over the iterations of the PSRO framework. This helps illustrate the **evolution of the strategy pool** as the sender and receiver adapt and refine their strategies through multiple iterations.
>
> **Rebuttal Summary:**
>
> In summary, Figure 7 provides a detailed view of how the strategies (prompts) evolve over time in our experiments using the PSRO framework with OPRO as the best response oracle. The figure captures the top 10 strategies with the highest selection probabilities, showing both the hierarchical optimization of prompt categories and content and how the selection probabilities of these strategies change over time. This evolution reflects the adaptation of the sender and receiver as they optimize their strategies within the verbalized Bayesian persuasion framework.
>
> We will further clarify these points in the revised version of the paper to make the analysis more accessible and ensure the relationship between the table columns and the prompt optimization process is clearer.

---

> ### Author Response · Authors · 2024-11-17
>
> > Q7: **Real-world Applicability**: Can your proposed method be applied to broader, real-world scenarios or other potential applications? If so, please briefly describe how it might be applied and any potential challenges.
>
> Thank you for your question regarding the real-world applicability of our proposed verbalized Bayesian persuasion (VBP) framework. We appreciate the opportunity to further elaborate on how our method can be applied to broader, real-world scenarios, particularly focusing on the two examples you mentioned, and to discuss the potential challenges in greater detail.
>
> **Generalizability to Multi-Sender, Multi-Receiver, and Multi-Round Tasks**: Since the VBP framework models Bayesian persuasion as an **extensive-form game** and uses large language models (LLMs) for decision-making and strategy optimization, it is theoretically extensible to more complex, real-world tasks involving **multiple senders**, **multiple receivers**, and **multi-round interactions**. This generalization opens the door to solving a wide range of real-world problems where multiple actors participate in strategic communication over several rounds, making it relevant for real-time decision-making and long-term strategic planning.
>
> **Example 1: Conversational Recommendation Systems**
>
> One significant real-world application is in **conversational recommendation systems**, particularly in the context of live-stream shopping. This scenario involves multiple senders (e.g., influencers or sales agents) trying to persuade a potentially large and diverse group of receivers (customers) to purchase products during a live-stream session. The dynamic interaction, with real-time communication between senders and receivers, makes it a perfect fit for multi-sender, multi-receiver, and multi-round BP problems.
>
>    - **How VBP Can Be Applied**: In this setting, each sender (influencer or salesperson) can be modeled as an agent who strategically chooses how to present information about a product to maximize customer engagement and conversions. The receivers (customers) are individuals with potentially different preferences, beliefs, and levels of trust in the senders. The VBP framework can optimize the prompts (e.g., how product information is conveyed or how offers are phrased) to maximize the likelihood of purchasing across various customer segments.
>
>    - **Potential Challenges**: A challenge in this scenario is the **heterogeneity of receivers**—each customer may interpret the signals differently based on their preferences, making it difficult to design a one-size-fits-all strategy. Additionally, the **real-time nature** of live-stream shopping requires highly efficient decision-making algorithms, as senders need to adapt their communication strategies on the fly. Scaling this to handle thousands or millions of receivers in real-time would require efficient parallel processing and optimization techniques.
>
> **Example 2: DRG Strategy in Healthcare**
>
> Another important real-world application is in healthcare, particularly in the context of the **Diagnosis-Related Group (DRG) strategy**. DRG systems are used by governments and healthcare providers to categorize hospital cases for the purpose of determining reimbursement rates. In such a system, the **regulator** (e.g., a government agency) acts as the **receiver**, while **hospitals and post-acute care (PAC) providers** act as the **senders** who have an informational advantage regarding patient conditions, treatment options, and costs.
>
>    - **How VBP Can Be Applied**: In this case, the senders (hospitals and PAC providers) have more detailed information about the patient's condition and treatment needs, while the government (receiver) needs to design a reimbursement policy that discourages unnecessary or overly expensive treatments. The VBP framework can be used to model the incentives and communication strategies of hospitals and PAC providers as they present information to the government. The goal would be to optimize the policy to encourage cost-effective treatments while ensuring patient care is not compromised.
>
>    - **Potential Challenges**: A key challenge here is the potential for **conflicting incentives** among the senders. This introduces a layer of complexity in the multi-sender BP problem, as senders might compete or collaborate to influence the receiver's decision. Additionally, the **scale of the problem**—with potentially thousands of hospitals and providers—requires the VBP framework to handle large-scale optimization efficiently. Moreover, the long-term nature of updating policies based on feedback introduces challenges related to multi-round interactions.

---

### Author Response · Authors · 2024-11-26
**Summary of Revisions in the Revised Version**

**Dear Reviewers,**

We are deeply grateful to all reviewers for their insightful and constructive feedback. Your comments and suggestions have been invaluable in improving our work's quality, clarity, and depth. We are pleased to inform you that we have uploaded the **revised version** of our paper, where we have carefully addressed the points raised during the review process. Below, we highlight the major revisions made:

1. **Improved Organization of the Main Text**:
   - We have reorganized the content by merging **Section 2.1 (Bayesian Persuasion)** and **Section 2.2 (Modeling BP as a Mediator-Augmented Game)** into a single, streamlined **Problem Formulation** section.
   - **Section 2.4 (Classic BP Problems)** has been moved to the experimental section to improve the flow of the paper.
2. **Incorporation of Related Work**:
   - We have supplemented the discussion with the work of Bai et al. in the revised version (added after **Appendix A.3**).
3. **Enhanced Explanations and Discussions**:
   - Additional analysis and discussion on Figure 7 have been provided in **Appendix G.1.2**.
   - A subsection on real-world applications has been included in **Appendix D.1** to better contextualize our work.
   - A more detailed discussion on unaligned LLMs has been added in **Appendix G.1.3**, along with expanded insights on the S3 setting in **Appendix G.1.1**.
   - A new discussion on obedience constraints has been added to **Appendix F.2** to clarify this key aspect and address potential concerns.
4. **Technical Additions and Future Work**:
   - We have included pseudocode in **Appendix B** for better clarity and implementation guidance.
   - A discussion on the Price of Anarchy (PoA) has been added to **Appendix H** for future work directions.
5. **Showcasing LLM Prompts**:
   - Due to space constraints in the main text, we have moved the LLM prompt demonstrations to **Appendix F.4**, with appropriate references added in the main text.

All revisions have been clearly marked with **blue highlights** in the revised version to facilitate your review. Additionally, we have corrected typos and minor errors identified during the review process.

We truly appreciate your dedicated time and effort in reviewing our submission. Your thoughtful feedback has been instrumental in helping us refine and improve our work. Thank you again for your invaluable contributions.

Best regards,

On behalf of all authors

---

### Author Response · Authors · 2024-11-27
**Kind Reminder to Reviewers: Feedback on Rebuttal and Revised Manuscript**

We deeply appreciate the time and effort all reviewers have already dedicated to reviewing our work and providing constructive feedback. We kindly ask if you could take a moment to review our rebuttal and the updated manuscript, and consider whether any adjustments to your evaluation and scores might be appropriate based on our revisions.

---

### Meta-Review · Area_Chair_okBp · 2024-12-21

**Metareview:**

This paper proposes a verbalized Bayesian persuasion framework using LLMs to model strategic communication in natural language settings, introducing techniques for optimizing prompts through game-theoretic approaches. Despite some promising empirical results, the majority of reviewers recommended rejection due to unclear novelty, unclear differentiation from existing work, and inadequate justification of the framework's necessity.

**Additional Comments On Reviewer Discussion:**

Reviewers raise concerns about the work's model formulation, theoretical analysis, and practical value. Specifically, PcFV questioned the mapping to mediator-augmented games and how convergence guarantees transfer, while gJz3 and tm2K found the tasks overly simplified and questioned the framework's practical applications. Though the authors provided extensive responses and revised several sections (including reorganizing content and adding new discussions on real-world applications), they did not fully address the fundamental concerns about theoretical foundations and the practical necessity of their complex framework.

---

### Decision · Program_Chairs · 2025-01-22

Reject